# Striatal hub of dynamic and stabilized prediction coding in forebrain networks for olfactory reinforcement learning

Laurens Winkelmeier [1], Carla Filosa[2], Renée Hartig [2], Max Scheller[2], Markus Sack[1], Jonathan R. Reinwald [1], Robert Becker[1], David Wolf[1], Martin Fungisai Gerchen [1], Alexander Sartorius [1], Andreas Meyer-Lindenberg[1], Wolfgang Weber-Fahr[1], Christian Clemm von Hohenberg [1,3], Eleonora Russo [1,2,3] & Wolfgang Kelsch [1,2,3 ✉]

Identifying the circuits responsible for cognition and understanding their embedded computations is a challenge for neuroscience. We establish here a hierarchical cross-scale approach, from behavioral modeling and fMRI in task-performing mice to cellular recordings, in order to disentangle local network contributions to olfactory reinforcement learning. At mesoscale, fMRI identifies a functional olfactory-striatal network interacting dynamically with higher-order cortices. While primary olfactory cortices respectively contribute only some value components, the downstream olfactory tubercle of the ventral striatum expresses comprehensively reward prediction, its dynamic updating, and prediction error components. In the tubercle, recordings reveal two underlying neuronal populations with non-redundant reward prediction coding schemes. One population collectively produces stabilized predictions as distributed activity across neurons; in the other, neurons encode value individually and dynamically integrate the recent history of uncertain outcomes. These findings validate a cross-scale approach to mechanistic investigations of higher cognitive functions in rodents.

[1] Central Institute of Mental Health, Medical Faculty Mannheim, Heidelberg University, 68159 Mannheim, Germany. [2] Department of Psychiatry and Psychotherapy, University Medical Center, Johannes Gutenberg University, 55131 Mainz, Germany. [3]These authors contributed equally: Christian Clemm von Hohenberg, Eleonora Russo, Wolfgang Kelsch. ✉email: wokelsch@uni-mainz.de

Complex functions like association learning involve a chain of cognitive processes comprising the recognition of unexpected outcomes and the updating of reward predictions (RP)[1,2]. In the framework of reinforcement learning, the RP associated with a conditioned stimulus (CS) is updated by a prediction error at the unconditioned stimulus (US). The prediction error computes the mismatch between the predicted value and the actual outcome[2–8]. Functional magnetic resonance imaging (fMRI) in humans suggests that forebrain regions contribute to a different extent to the computation of the RP and prediction error[9–12]. In translational rodent and primate models, deeper mechanistic insights have emerged from the study of few cell types and brain regions, most notably dopamine neurons in the ventral tegmental area[3,4,6,8,13–16]. Neuronal correlates underlying stimulus-outcome learning are examined in rodents frequently with odor cues. In analogy to human olfactory learning[17–21], rodent olfactory and higher-order cortices, as well as striatal regions appear to differ in their contribution to stimulus value and reward outcome coding[8,13,22–32]. Comparisons between the region-specific contributions to the associative process are however limited, since only one or few regions are respectively studied under the same task conditions and, often, outside of an explicit reinforcement learning framework.

To identify the forebrain regions involved in these computations, a possible solution would be fMRI in task-performing rodents combined with behavioral parametrization. This would provide a translational discovery approach to leverage animal models amenable to mechanistic dissection. Recent developments demonstrate the potential of fMRI for describing reward and fear circuits in awake, passive mice[33–35] or connectivity during resting state in mouse mutants[36]. Even though fMRI has become a standard tool to assess task-related brain activity in humans, to date, only very few fMRI studies exist in rodents during task performance[37], and none examine whether BOLD correlates reflect the task information of the local cellular coding.

To address these issues, we develop here a hierarchical cross-scale approach, from behavioral modeling and fMRI to cellular recordings, for an olfactory stimulus-outcome learning task with different reward probabilities. As a first step, we model the learning to formulate specific hypotheses about RP and prediction error signals; on such behavioral correlates, we then regress the fMRI data from mice performing the task in the scanner. We aim here to identify a functional network of forebrain regions and their respective contributions to RP and prediction error computations. We then validate with single-unit recordings whether the BOLD signal reflects the task-related information of the local neuronal code. Finally, we dissect coding mechanisms underlying these RP signals in olfactory reinforcement learning.

## Results

**Reward prediction and its updating by the recent outcome history.** To disentangle the regions involved in distributed computations of reward prediction and prediction error in the mouse forebrain, a cohort of 23 animals, examined later with fMRI, was habituated to a head-fixed setup and trained on stimulus-outcome pair associations. Mice were conditioned to odor stimuli that signaled different probabilities of upcoming reward (100%—geranium, 50%—ylang–ylang, or 0%—rose; hereafter, these trial types are labeled CS100, CS50, or CS0, respectively) (Fig. 1a). Across sessions, odor and reward contingency pairs were kept unchanged. The odor stimulus was presented for 1 s followed by a 1.7 s wait interval before the water drop was delivered with the given reward probability (Fig. 1a). The water port was positioned so that mice had to actively lick to sense and retrieve the reward. After training, mice performed above criterion and licked in more

than 80% of CS100 and CS50 trials, but not in the CS0 trials (Supplementary Fig. 1a). Lick intensities in the waiting window correlated with the probability of reward to that odor cue (Fig. 1b, c and Supplementary Table 1 for statistical test). In ten trained mice, we monitored pupil responses as a second proxy for the animals' reward expectation (Fig. 1d and Supplementary Fig. 1b, c). Pupil dilation equally reflected the stimulus-specific RP (Fig. 1e). Thus, mice learned to predict reward outcomes upon CS presentation.

We used reinforcement-learning modeling to parametrize the RP value associated at CS with each odor ('$V(CS)$'), and the prediction error at US ('$\delta(US)$'). We built a temporal difference (TD) model with the learning rate ('$\alpha$') set as a free parameter and optimized on the pupil responses of the behavioral sessions. The average of the learning rates was used to build a TD model applied throughout the study. Exploiting the generative power of the TD model, we simulated 100 sessions and observed that the average of $V(CS)$ across sessions approached the true reward frequencies associated with each CS (Fig. 2a). As expected, the absolute value of $\delta$ at the time of potential reward became small in CS100 and CS0, but remained large for CS50 trials (Fig. 2b). In CS50 trials, the average reward probability was at chance level throughout the session. Yet, the RP value was dynamically updated at each trial to integrate local fluctuations on the outcome probabilities. According to the TD model, the RP associated with a CS at each time instance $t$ results from the integration of the whole history of rewards paired (or not) with each instance of CS. Each obtained reward will increase the RP associated with CS50 while the absence of reward will diminish it. Consistently, when dividing CS50 trials according to the outcome of the preceding CS50 trial, the recent outcome history was reflected in the state values of the TD model (Fig. 2c, d) and confirmed behaviorally by the pupil responses during the waiting interval (Fig. 2e). Thus, animals learned to predict reward outcome at CS modulating RP by the recent outcome history of uncertain rewards as predicted by the TD model.

**Regional specializations for olfactory reward prediction in forebrain networks.** We employed fMRI to localize forebrain regions involved in olfactory RP, modulation by recent outcome history, and the prediction error. A Bruker 9.4 T rodent MR scanner was used to image mice during task performance. A mouse MRI cradle was designed with odor and lick ports, connectors for head bars, and a cover over the back of the mouse (Fig. 3a). Mice were habituated to the cradle and head fixation in mock scanners, progressively adding the task paradigm, and then increasing levels of MRI sound recorded during an fMRI session. After completion of this training (Supplementary Fig. 1d), the cohort of 23 mice underwent fMRI. No anesthesia or sedation was used at any stage of the fMRI experiment, including habituation and image acquisition. To reduce stress and assure task performance, mice performed 20–30 preparatory trials before commencing the BOLD imaging sequence. In order to maintain comparable levels of satiety and motivation as the 150-trial sessions outside the scanner, also inside the scanner only 120 trials (following the preparatory trials) were considered. Sessions where animals did not lick at reward were stopped and the animals were not used for further sessions. Of the 67 completed scanning sessions, 51 sessions in 18 animals performed above criterion (>80% correct hits and rejections per session) (Supplementary Fig. 1e–g).

We investigated a larger forebrain network comprising the olfactory bulb, the primary olfactory and prefrontal cortices, and the ventral and dorsal striatum (Fig. 3b). Upon preprocessing, we computed three different general linear models (henceforth

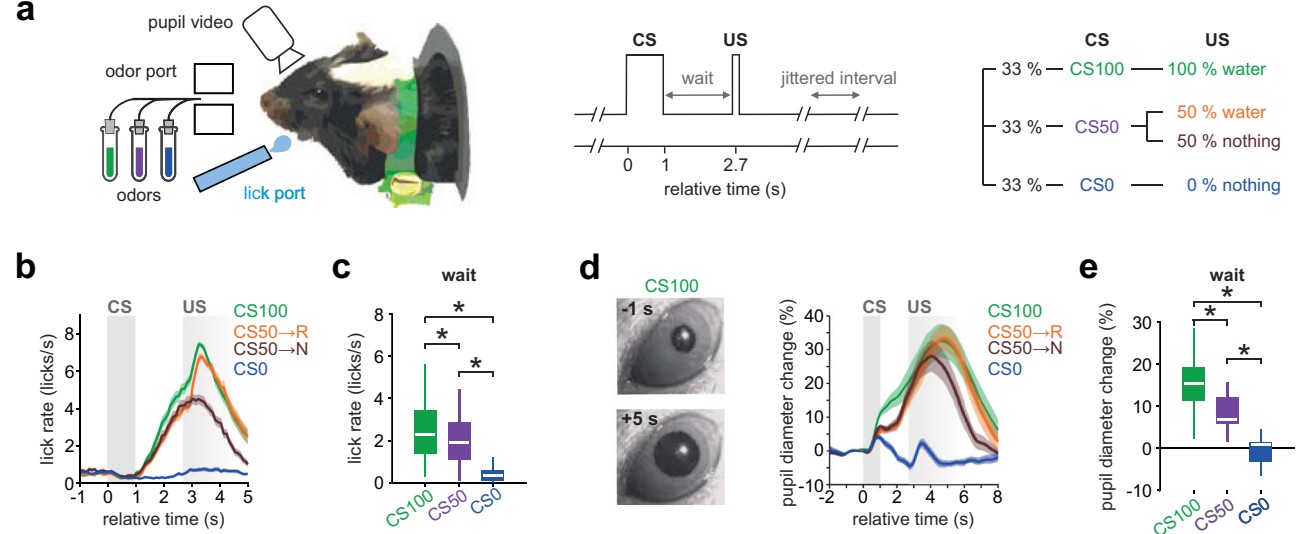

**Fig. 1 Trained mice display differential anticipatory responses to olfactory stimulus-outcome pairs. a** Mice were trained to learn stimulus-outcome pairs: Three different odors (CS) were applied for 1 s followed by a fixed waiting period of 1.7 s before a drop of water was delivered (US). Licking behavior and pupil diameter were monitored simultaneously. Stimulus CS100 and CS50 predicted US with different reward probabilities (100% and 50%). Stimulus CS0 was never rewarded. The time interval between CS onsets of consecutive trials (trial duration) was jittered between 10 and 12 s. **b** Evolution of average lick rates ± SEM differentiated trial types ($n = 69$ training sessions in 23 animals, 3 sessions per animal). **c** Lick rate in the waiting window differentiated the respective trial types with CS100 > CS50 > CS0 ($n = 69$ training sessions in 23 animals; one-way ANOVA with Tukey post hoc comparisons). **d** Video frames illustrating pupil images 1 s before and 5 s after the onset of CS100 (left). Average change ± SEM of the baseline subtracted pupil diameter also differentiated trial types (right; $n = 10$ sessions in ten animals). **e** In the waiting period, average (±SEM) percentage change of pupil diameter revealed a similar pattern as licking ($n = 10$ sessions in ten animals; one-way ANOVA with Tukey post hoc comparisons). In the figure: * indicates $P < 0.05$ (see Supplementary Table 1 for exact $P$ values and test details). Box plots: The bounds of the box represent 25th to 75th percentiles. The center indicates the median. The lower and upper whiskers represent the minimum and maximum values. Source data are provided as a Source Data file.

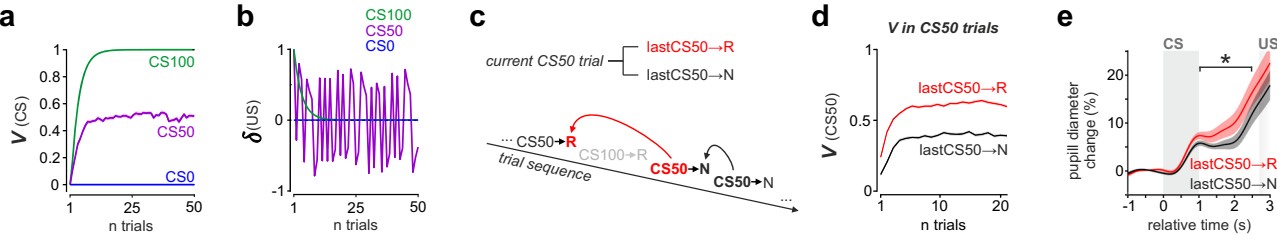

**Fig. 2 Fluctuations in pupil dilation reflect cue-specific value update by recent reward history. a** Averaged RP values $V(CS)$ of the three CS from the TD model ($n = 100$ simulated CS-US sequences, learning rate $\alpha = 0.28$ optimized from pupillary data, initial conditions set to zero). The values approached the true reward probabilities associated with each CS. **b** The prediction error $\delta$ at US approached zero in CS100 and CS0 trials, while it remained large in CS50 trials as shown here for an exemplary session. **c** Scheme illustrating the subdivision of CS50 trials for recent outcome-history analysis. CS50 trials were grouped according to the outcome of the previous CS50 trial (rewarded: lastCS50→R; unrewarded: lastCS50→N). **d** Average modeled $V(CS)$ for CS50 trials divided according to lastCS50→R and lastCS50→N predicts a dynamic update of RP based on the recent outcome history. **e** Average change ± SEM of the normalized pupil diameter in CS50 trials split for lastCS50→R and lastCS50→N. The change in pupil diameter during waiting followed the recent history update (two-tailed paired $t$ test). In the figure: * indicates $P < 0.05$ (see Supplementary Table 1 for exact $P$ values and test details).

termed GLM 1, GLM 2, and GLM 3) on the BOLD time series of 51 sessions, modeling CS and US timepoints as events, which were parametrically modulated as described below, and convolved with a previously determined mouse hemodynamic response function[38] (see "Methods" and Supplementary Fig. 2a–d for detailed information about preprocessing and model designs; for complementary analyses see Supplementary Fig. 3, and for additional control analyses see Supplementary Fig. 4). The average magnitude of BOLD responses (computed as percent signal change) was mostly in the range between 0 and 2% (Supplementary Fig. 2e), similar to a prior behaving fMRI study in rodents[39].

When parametrically modulating the CS regressor with the model-estimated RP value at CS, $V(CS)$ (GLM 1), we observed

broad recruitment of brain regions (Fig. 3c; for a schematic summary of task-related regional activations see Supplementary Fig. 3g) with positive signals in the olfactory bulb, the medial prefrontal cortex (mPFC) and the agranular insular cortex. Negative BOLD signals to $V(CS)$ were found in olfactory primary cortices, namely the ventral parts of the anterior olfactory nucleus and partially also the anterior piriform cortex (aPC). The dorsal striatum and the two ventral striatal brain regions, consisting of the olfactory tubercle (Tu) and nucleus accumbens (NAc), were also correlated with $V(CS)$.

If reward prediction is the dominant feature of a region, response intensities should reshape and follow monotonically the predicted reward probability of the CS after learning. GLM 1 does however not distinguish between a strict monotonic neuronal

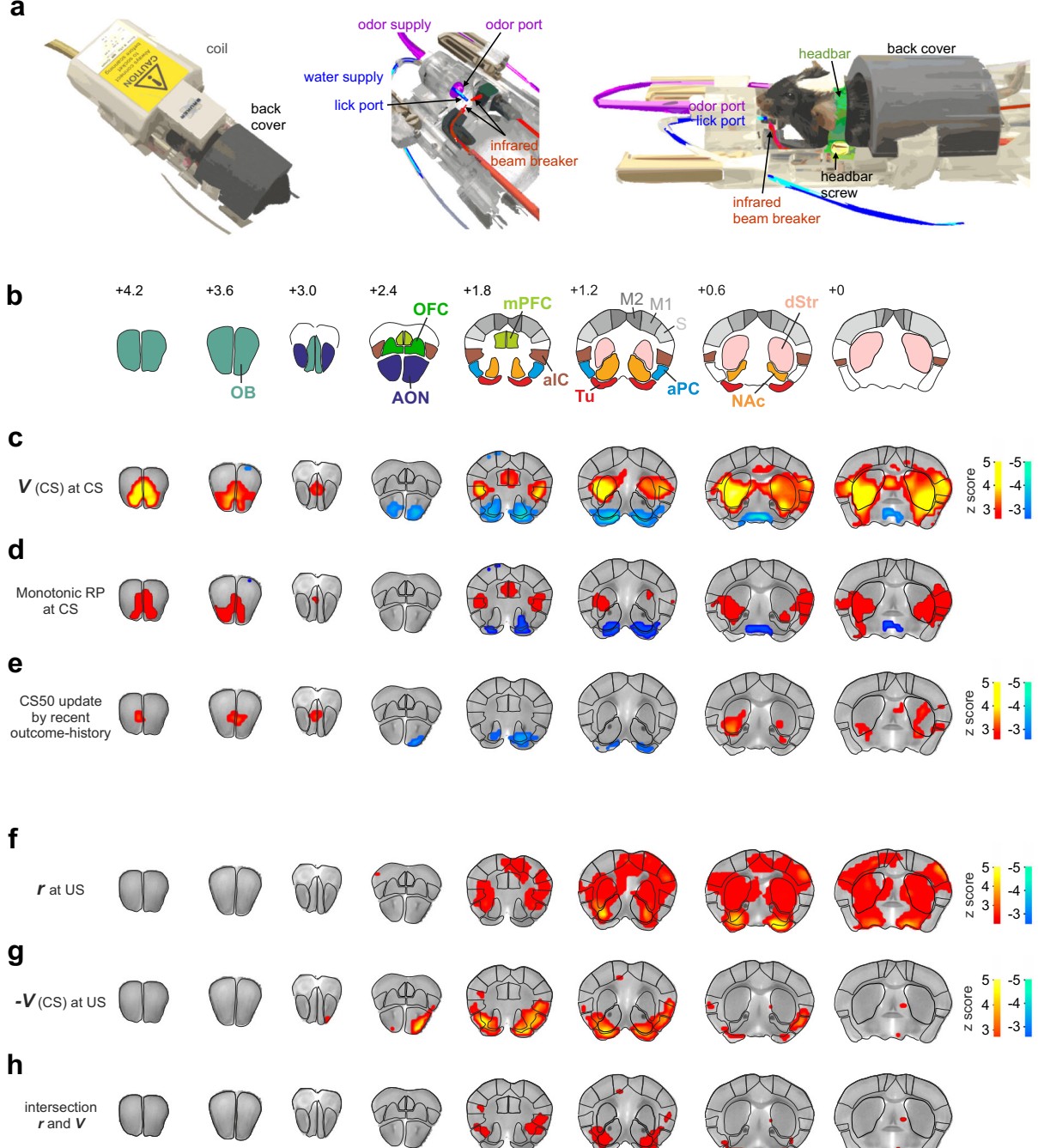

**Fig. 3 Olfactory-association-learning network expresses monotonic RP and prediction error components. a** Illustration of the MRI-compatible behavioral setup with MRI coil (left) comprising an odor port, a lick port, respective supply tubes, an infrared beam breaker, a head-fixation system, and a back cover (middle and right). **b** Anatomical illustration of olfactory, striatal and higher-order regions, mapped onto an MRI template brain (location from Bregma indicated in mm). aIC agranular insular cortex, AON anterior olfactory nucleus, aPC anterior piriform cortex, dStr dorsal striatum, M1/M2 motor cortex, mPFC medial prefrontal cortex, NAc nucleus accumbens, OB olfactory bulb, OFC orbitofrontal cortex, S somatosensory cortex, Tu olfactory tubercle. **c** Group-level Z-statistical maps showing BOLD correlates of RP values $V(CS)$ from the TD model ($n = 51$ sessions in 18 animals). Statistical threshold was set to $P < 0.025$ false discovery rate (FDR)-corrected, for two-sided testing (as for the other maps unless otherwise indicated). Red color spectrum indicates areas with positive correlations to RP values, while blue colors indicate negative correlations. The regression revealed recruitment of olfactory and striatal brain regions with opposing effects on BOLD. **d** Within the regions associated with $V(CS)$ from (**c**), monotonic RP (CS100 > CS50 > CS0 in red or CS100 < CS50 < CS0 in blue) was represented mainly in prefrontal, insular, striatal regions, and olfactory bulb. Maps were created by intersecting contrast maps CS100 > CS50 and CS50 > CS0, each thresholded at $P < 0.025$ FDR-corrected; and vice-versa: CS100 < CS50 and CS50 < CS0 (see the respective contrast maps in Supplementary Fig. 3e, f). **e** Recent outcome history modulating the BOLD response to CS50. Within the $V(CS)$ regions from (**c**), we tested for a CS50 response enhancement by the reward in the previous CS50 trial. In olfactory and striatal regions, the CS50-associated response was indeed strengthened by a positive outcome history. **f**–**h** In the TD model, the prediction error at US is $r-V(CS)$. We identified brain regions correlating with both prediction error components by intersecting the regressor maps of (**f**) $r$ and (**g**) $-V(CS)$ at US, each thresholded at $P < 0.05$ FDR-corrected. The intersection, shown in (**h**), was largely confined to the lateral NAc, the posterior Tu, and aIC.

representation of $V(CS)$ (CS100 > CS50 > C0), and a simple binary differentiation between presence or absence of reward expectation (e.g., CS100 = CS50 > CS0). Both types of representations would result in a significant correlation with $V(CS)$. Indeed, CS100 and CS50 recruited similar brain regions, unlike CS0 (Supplementary Fig. 3a–d). To probe which brain regions represent RP in a monotonic fashion, such that higher reward expectation is reflected by a stronger BOLD or neuronal response (i.e., CS100 > CS50 > CS0), we computed a separate GLM 2, where the three CS trial types were modeled with individual regressors. We determined BOLD correlates of monotonic RP by intersecting significance maps of specific contrasts ((CS100 > CS50) ∩ (CS50 > CS0), or vice-versa for negative contrasts (CS100 < CS50) ∩ (CS50 < CS0), Supplementary Fig. 3e, f). Within the $V(CS)$-significant regions, monotonic RP was expressed primarily in a network comprising the mPFC and agranular insular cortex, as well as striatal circuits (Tu, NAc and lateral striatum) and the olfactory bulb as a primary olfactory region (Fig. 3d). This monotonic RP network was largely determined by the narrow contrast between the two rewarded stimuli CS100 and CS50 (Supplementary Fig. 3e). Again, the Tu and NAc displayed monotonic RP with negative contrasts. We then investigated, among the regions representing $V(CS)$, which brain regions participated in the RP updating based on the recent outcome history, as predicted by the TD model (cf. Fig. 2c, d). To this end, the CS50 regressor was parametrically modulated by the binary outcome of the preceding CS50 trial (Supplementary Fig. 1h, i). Among the regions representing $V(CS)$, the three striatal regions and the early olfactory processing circuits (olfactory bulb, anterior olfactory nucleus) reflected the cue-specific outcome-history updating (Fig. 3e).

In the TD model, the prediction error $\delta$ at US is $r - V(CS)$. In humans, most brain regions contribute to either one of the two components of the prediction error at US ($r$ or $-V(CS)$), and only few regions compute both[40,41]. To test for BOLD correlates of the prediction error at US, we set up GLM 3, where we parametrically modulated the US regressor with $-V(CS)$ and with $r$ within the same GLM. We found correlates of $r$ (Fig. 3f) in the lateral NAc, the posterior Tu, the dorsal striatum and, additionally, also in the insula. In contrast, $-V(CS)$ at US (Fig. 3g) involved broadly a ventral stream of olfactory and striatal regions and additionally the insular cortex. The intersection of $r$ and $-V(CS)$ (Fig. 3h) included the lateral NAc, but comprised also the posterior Tu, and partially the insular cortex; thus indicating a relatively confined set of candidate regions that could be involved in the prediction error at US for olfactory stimulus-outcome learning. Co-localization of both prediction error components ($r$ and $-V(CS)$) could in principle emerge also from two independent processes. The expression of the prediction error in candidate regions will be therefore further explored by electrophysiology.

In summary, (monotonic) RP and the prediction error components recruited only specific olfactory, prefrontal, and striatal regions, respectively. Among them, few regions, including the Tu, also displayed RP updating by the recent cue-specific outcome history. Further, the contribution of the aPC as a primary olfactory cortex to these computations was relatively restricted compared to other olfactory regions.

**Task-related functional connectivity of the aPC and Tu.** From the previous analyses, the Tu appeared as hotspot for the computation of all RP aspects. This can result either from local processes or from interactions with other connected regions (or both). Together with the olfactory bulb, one of the main sources of direct synaptic inputs to Tu is the aPC[42]. Yet, aPC neither

reflected monotonic value nor dynamic value updating and, somewhat surprisingly for a sensory cortex, peaked at US with its value association (cf. Fig. 3c, g). One reason for the functional differentiation of aPC and Tu may reside in their respective task-related functional connectivity with other regions. We therefore first examined the relationships of Tu or aPC with the odor-reward association learning network of olfactory, striatal and higher-order regions (Fig. 4a). Task-related functional connectivity was computed using beta-series correlations, based on an adapted GLM that modeled each event (CS and US) in each trial as a separate regressor[43]. Briefly, task-related functional connectivity (henceforth also called "functional connectivity" for brevity) between either Tu or aPC and regions-of-interests (ROIs) contained within the network was first determined by computing the Pearson's correlation coefficient between the series composed of the region-averaged GLM beta weights of each trial. This correlation was computed at CS and US for each trial type (Fig. 4b, c). Note that this method is correlational in nature and does not allow inferences about causality or common synaptic inputs. These analyses revealed that both Tu and aPC were embedded in a ventral stream of functional connectivity, which included NAc and olfactory regions, such as the anterior olfactory nucleus. aPC and Tu displayed strong functional connectivity with each other (Fig. 4b, c). Yet, while the distributed network functionally correlating with aPC was relatively uniform across stimuli, compatible with stimulus detection, the functional connectivity of the Tu network differentiated between rewarded and unrewarded CS (especially in the coordination of the dorsal striatum and higher-order cortices), in line with the role of Tu in value computation. These findings were confirmed by a voxel-based analysis between each seed region (Tu and aPC) and all the voxels contained within the odor-reward association-learning network, accounting for possible intraregional differentiations (Fig. 4d, e and Supplementary Fig. 5a, b). Consistent with differential circuits recruited in go and no-go trials, the functional connectivity for both Tu and aPC differed between CS0 and the rewarded CS (Supplementary Fig. 5c, d). This differentiation continued into US. Further, a confined functional connectivity differentiation was found at US between rewarded and non-rewarded CS50 trial outcomes selectively for Tu with NAc and the anterior olfactory nucleus. Thus, in summary, although the aPC and Tu displayed strong task-related functional connectivity with each other, they differed in their task-dependent co-activations with the odor-reward association-learning network.

**Task-related neuronal population coding in the olfactory cortex and striatum.** The differential embedding of the aPC and Tu in task-related functional connectivity further supported their distinct roles in olfactory-association learning. We next validated whether the local cellular activity confirms the encoding of the RP and prediction error components as revealed by fMRI and investigated their underlying coding mechanisms. Specifically, we wondered how the cellular coding of value prediction and error is solved in the Tu. Further, we addressed the relative paucity in value coding of the aPC that appeared to evolve surprisingly late for a primary sensory cortex after the stimulus onset. Finally, we wondered how the task-related BOLD negativity may be interpreted in light of the principal neuron activity of the two regions.

To dissect the local cellular coding underlying the computations of the aPC and Tu, we performed chronic single-unit recordings in a separate cohort of 11 mice with a custom-designed tetrode array. Dual-site recordings comprised 16 tetrodes per brain region (Fig. 5a and see also Supplementary Fig. 6a–c). We examined trained animals that performed the task above criterion (Supplementary Fig. 6d–f) and obtained 169

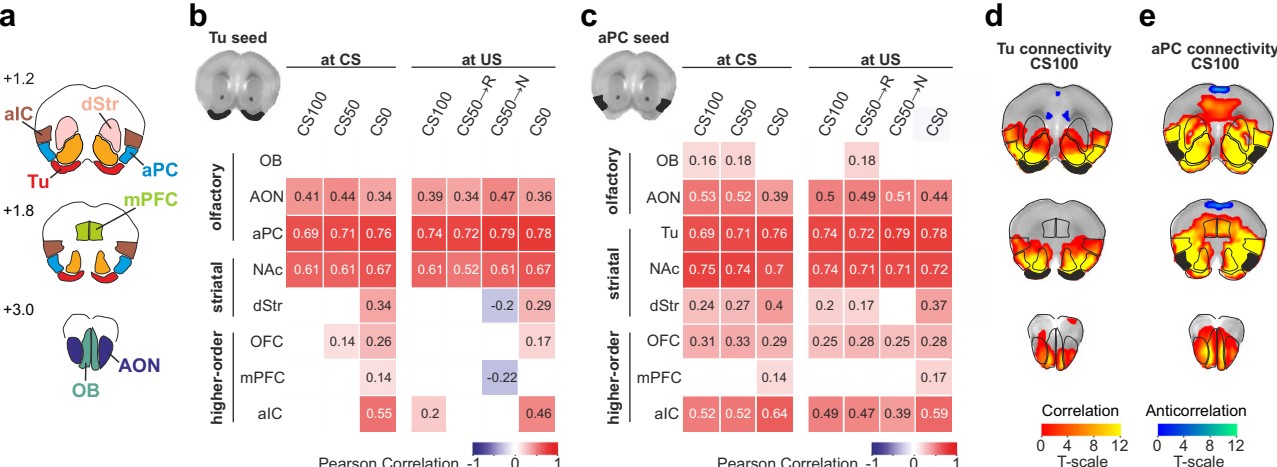

**Fig. 4 Learned odor-reward associations recruit differential networks functionally connected to Tu and aPC. a** Analogous to Fig. 3b. Task-based functional connectivity was examined between regions of the olfactory association-learning network (OB, AON, aPC, Tu, NAc, dStr, OFC, mPFC, and aIC). **b**, **c** Task-based functional connectivity between (**b**) Tu and the other regions for CS and US for each trial type. Pearson correlation coefficients between the region-wise beta-series were calculated for each session (*n* = 51 across 18 animals). The group-level significance was tested using a one-sample *t* test on the Fisher z-transformed correlation coefficients (*P* < 0.05, Bonferroni-corrected for multiple comparisons). Pairwise correlations that did not reach significance are indicated by empty boxes. The connectivity between Tu and higher-order regions depended on the respective CS. **c** Same as (**b**) showing aPC connectivity. **d**, **e** Analogous analysis as in (**b**) and (**c**), but voxel-wise. Group-level maps showing seed-based functional connectivity for CS100 in (**d**) Tu and in (**e**) aPC (*n* = 51 sessions in 18 animals). Significance threshold was set to *P* < 0.025, FDR-corrected for two-sided testing. Red color spectrum indicates a positive correlation, while blue color indicates anti-correlation. Seed regions marked in black. A composite mask of all ROIs listed in (**a**) was used for this analysis. See Supplementary Fig. 5a, b for the complete analysis performed on different task conditions.

putative striatal projection neurons with baseline firing below 5 Hz in the Tu (Supplementary Fig. 6g, h). In addition, we obtained 486 regular firing neurons in the aPC with baseline firing below 10 Hz. To capture coherent positive and negative changes in the population firing rate, we computed the deflection from baseline of the population vector during the trial (Fig. 5b). We computed the Euclidean distance between the population vector at each time instance and at baseline, to detect changes in angle or rate of the population activity. In the Tu, population activity coded for the predicted reward value of the stimulus at CS and during the waiting period (Fig. 5c, d). We quantified then to which extent different odors and outcomes recruited the same units. By computing the Pearson cross-correlation between the average instantaneous population vectors associated with different CS, we found progressive recruitment of units encoding cross-stimulus reward anticipation during the waiting period of CS50 and CS100 trials, and progressive decorrelation between the rewarded and non-rewarded CS trials (Fig. 5e). At US, population trajectories during CS50 trials diverged according to the outcome and, in case of reward, were pushed closer to the trajectory of CS100 trials (Fig. 5f).

The aPC displayed different population responses (Fig. 5g–j). In contrast to the Tu, aPC population activity was most pronounced during CS (Fig. 5g), as expected for a primary sensory cortex, and did not reflect RP (Fig. 5g, h). The CS onset triggered a correlated detection response across all trial types in aPC (Fig. 5i). This was also seen in the initial upshot of the population trajectories common to all stimuli (Fig. 5j, contrary to the Tu cf. Fig. 5f). Thus, while aPC had a pronounced detection response to olfactory stimuli, the Tu population coded for monotonic RP both during odor presentation as well as during the anticipatory waiting in trained animals.

**Task-inhibited value responses dominate aPC.** We then aimed to better understand what cellular activity patterns underlie the negative BOLD correlates of value in the aPC. In trained mice,

after an initial positive response to CS detection, negative mean population rate changes prevailed during waiting and upon US for all trial types (Fig. 6a, b). Interestingly, this negative rate change encoded predicted stimulus value during waiting, but not at CS, and responded to different reward outcomes at US (Fig. 6b). The population coding, and also the fMRI data, were examined in animals trained for at least two weeks in the specific task. As learning may modify the proportions between excited and inhibited responses[44], we wondered whether the predominance of task-inhibited responses emerged with training. We thus analyzed the first session the animal ever faced the task with the three stimuli CS100, CS50, and CS0 (Fig. 6c and Supplementary Fig. 7a–c). Throughout the training, all animals had been presented with the same odor identity-reward probability associations. At the beginning of the first session, the odors elicited different mean responses not reflecting reward probabilities, but most likely intrinsic odor properties (Supplementary Fig. 7c and Supplementary Table 1 for statistical test). In this initial training session, the net population rate response was relatively balanced (Fig. 6c). With training, the fraction of units with task-excited responses did not change. Yet, their relative contribution in the CS50 and CS100 trials markedly dropped due to the increase of task-inhibited responses (Supplementary Fig. 7d, e).

The development of value coding at the single-unit level would require an acquired generalization in the unit responses to rewarded stimuli. However, aPC units with task-excited responses did not change their selectivity with learning (Fig. 6d–f). Considering the strong changes in task-inhibited responses, learning appeared to affect units negatively modulated by the task. To confirm this observation and account for a potential heterogeneity of aPC subpopulations contributing to RP coding, we clustered neurons of trained animals according to their task activation profiles (Fig. 6g). Units clustered in a variety of subpopulations and fell into three main groups according to their dominant characteristics. The main subpopulations consisted of transient CS-excited units (Fig. 6h) and a large group of task-inhibited units (Fig. 6j). Moreover, an additional smaller group of

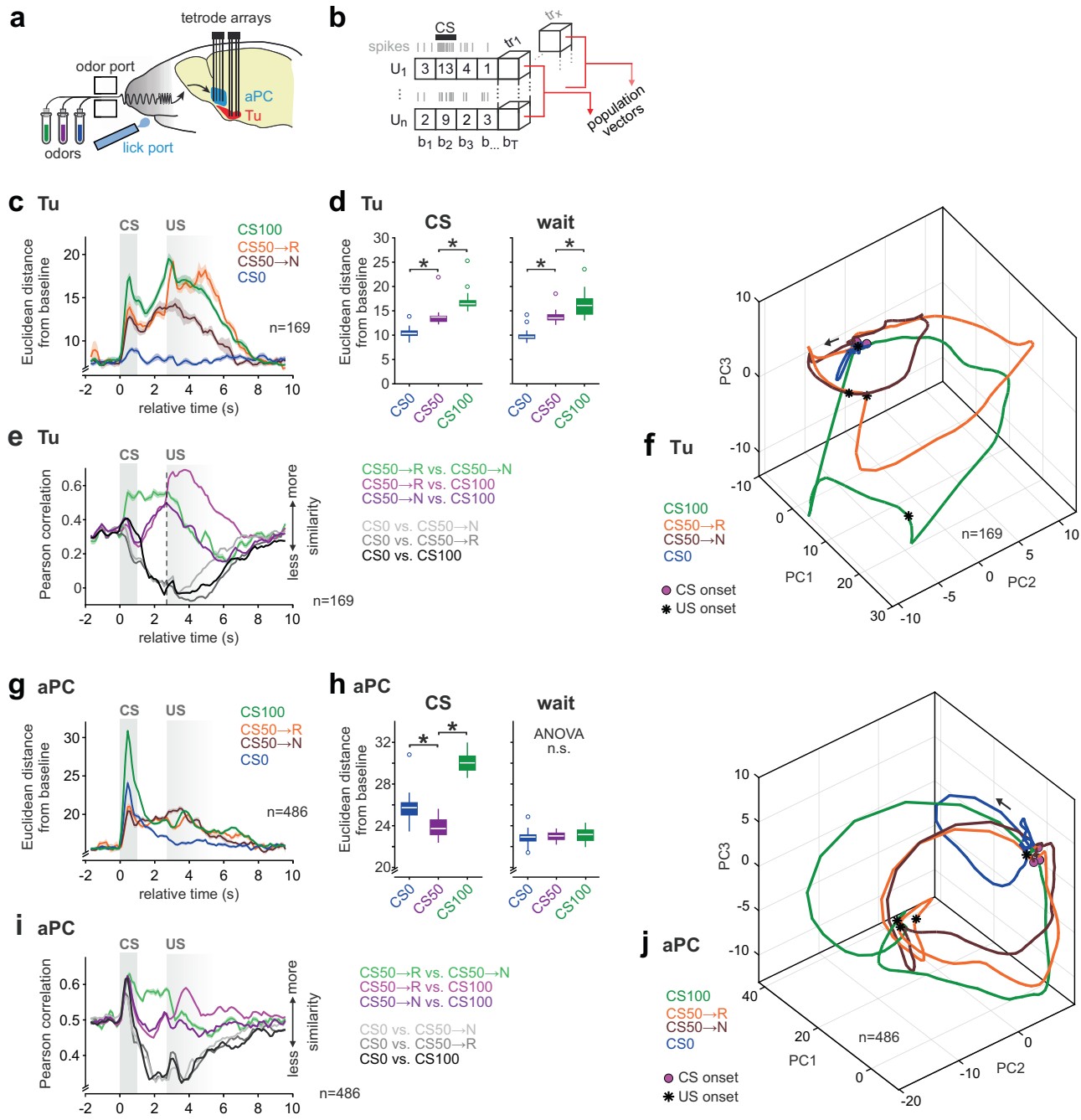

**Fig. 5 Task-related neuronal population coding in the olfactory cortex and striatum. a** Configurations of the dual-site single-unit recordings. Custom-made tetrode arrays were implanted into the Tu and aPC. Mice performed the same stimulus-outcome-associated task in head-fixed configuration as in the scanner. **b** Population vectors were constructed by concatenating the binned spike counts of individual single units $\{U_1, \ldots, U_n\}$ pooled across sessions by matching trial bin $b_i$, trial order $tr_j$, and trial type. **c** The Euclidean distance of Tu population vectors from baseline displayed an RP coding at CS and during the waiting period, as well as reward coding at US. Displayed mean ± SEM, 500 ms bins slid in steps of 125 ms. **d** Euclidean distance from baseline during CS (from 0 to 1 s) and waiting window (from 1 to 2.5 s). One-way ANOVA with Tukey post hoc comparisons performed separately in each window ($n = 169$ units). **e** Pearson correlation between Tu population vectors displayed as mean across pairs of trials ± SEM. The population responses to CS100 and CS50 became more similar during wait, suggesting a partial overlap in Tu units coding for the two stimuli. **f** First three principal components of the time-embedded averaged trajectories of Tu during different trial types. The arrow marks the direction of the temporal evolution. Reward delivery abruptly deflected the population trajectories of CS50→R and CS50→N. **g, h** Same as (**c, d**) for aPC. aPC population displayed a dominant response to all CS, but no monotonic RP coding ($n = 486$ units). **i** Same as (**e**) for aPC. The similarity between aPC population response to all stimulus onsets is consistent with a sensory detection signal. **j** Same as (**f**) for aPC. Also aPC population trajectories reflected an initial indiscriminate sensory detection signal. In the figure: * indicates $P < 0.05$ (see Supplementary Table 1 for exact $P$ values and test details) and $n$ indicates the number of units. Box plots: The bounds of the box represent 25th to 75th percentiles. The center indicates the median. The lower and upper whiskers represent the minimum and maximum values without outliers; outliers indicated by circles. Source data are provided as a Source Data file.

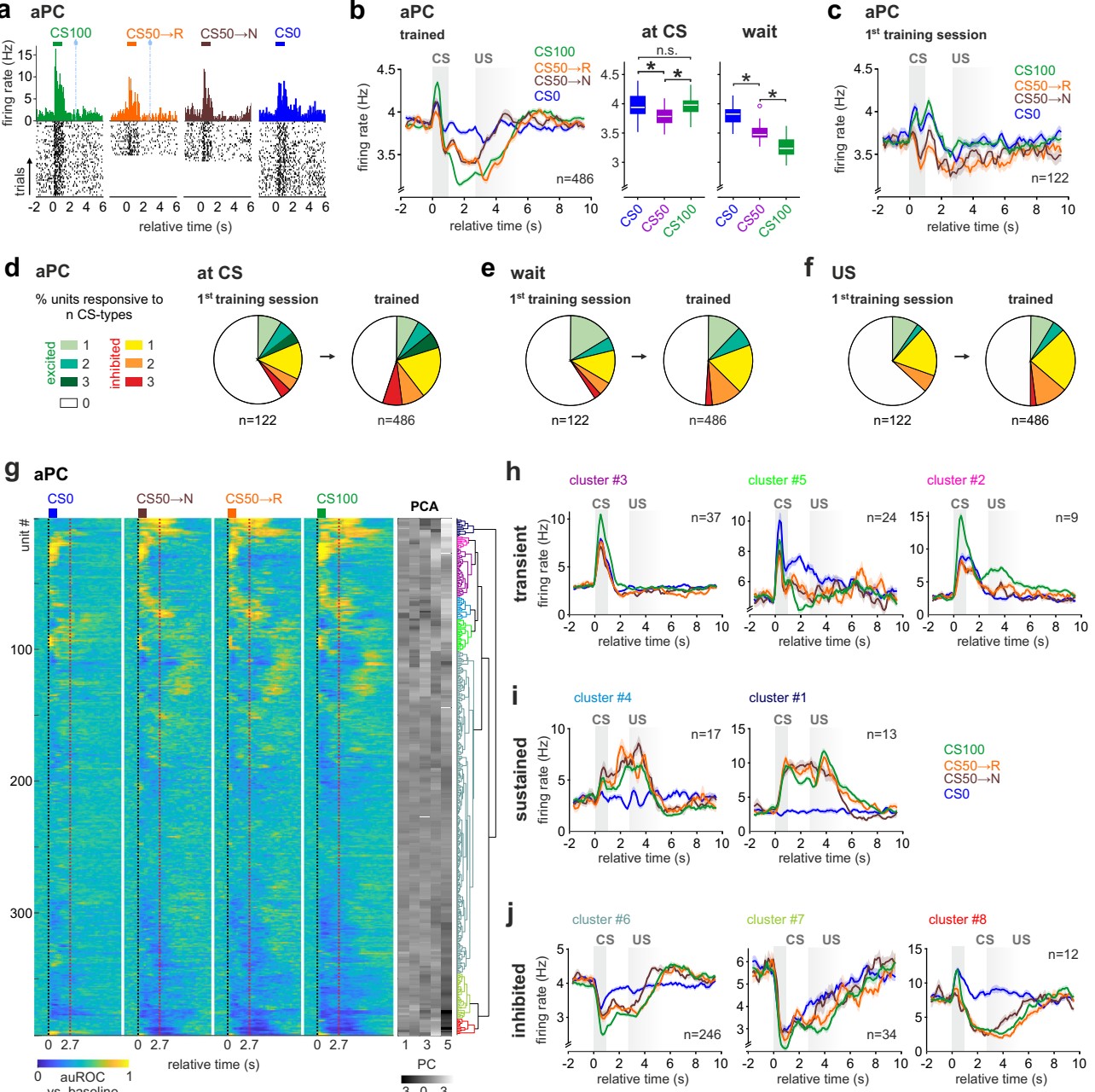

**Fig. 6 Task-inhibited value responses dominate aPC. a** Example of single-unit firing activity in aPC in trained animals. Peri-stimulus time histogram (PSTH) and spike raster plots showed stable task-related responses. **b** Task-related mean firing rate evolution ± SEM in aPC units. After a first detection peak at CS, aPC displayed monotonic RP during the waiting period as a progressive decrease in firing rate. Right: Box plots showing median aPC firing rates for CS and waiting period ($n = 486$ units; one-way ANOVA with Tukey post hoc comparisons). The bounds of the boxes represent 25th to 75th percentiles. Lower and upper whiskers represent minimum and maximum values without outliers; outliers indicated by circles. * indicates $P < 0.05$ (see Supplementary Table 1 for $P$ values and test details). Source data are provided as a Source Data file. **c** Task-related mean firing rate evolution ± SEM in aPC units in the first training session. Note the more balanced net firing rate during the waiting period. **d–f** Fraction of single units showing significant task-excited, task-inhibited, or no responses to either one or more trial types (CS0, CS50, CS100) at CS (**d**), during wait (**e**), and at US (**f**). Unit responsiveness was tested against baseline (Friedman test, $P < 0.05$ with Benjamini–Hochberg correction). The fraction of task-inhibited units increased after training. The fraction of excited units, as well as their level of selectivity to one or more trial types, remained relatively stable. **g** Classification of aPC units into functional clusters. Unit responsiveness during different trial types was quantified by auROC. (left) Hierarchical clustering was performed on the first five principal components (center, grayscale) of the auROC traces and revealed eight prominent task-response clusters (right). **h–j** The task-response clusters identified in (**g**) were grouped according to their responsiveness at CS and during the waiting period. We identified three major groups with (**h**) task-excited transient activity at CS, (**i**) task-excited sustained activity during waiting, and (**j**) task-inhibited responses. Displayed mean firing rate ± SEM. The color of the titles matches the cluster color in the dendrogram in (**g**). Unlike the task-inhibited cluster, none of the task-excited clusters displayed a monotonic RP (see Supplementary Fig. 7f–h). $n$ indicates the number of units.

units displayed heterogeneous sustained activity during waiting (Fig. 6i). Neither of the two task-excited cluster groups showed monotonic RP in their average response (Supplementary Fig. 7f, g), which was, however, again reflected in the task-inhibited population (Fig. 6j and Supplementary Fig. 7h). Thus, with training, aPC shifted toward task-inhibited responses that contributed to value coding.

**Parallel populations encode reward prediction in Tu**. We then examined the cellular dynamics underlying the RP and prediction error correlates in the olfactory striatum revealed by fMRI. In trained mice, the mean population firing responses encoded RP at CS and during waiting, as well as reward surprise at US (Fig. 7a, b). As expected, during the first session, reward surprise, but not RP, dominated task responses (Fig. 7c). Value coding began to emerge during the first session with the stimulus responses to the fully-rewarded CS100 differentiating from the other stimuli (Supplementary Fig. 8a and Supplementary Table 1 for statistical test). Notably, the mean rate changes to CS0 were still positive in this first training session. With subsequent learning, the fraction of units with task-excited responses dropped in CS0 trials and increased in the rewarded trial types. As in aPC, training increased the fraction of Tu units with task-inhibited responses across trial types (Supplementary Fig. 8b, c). Contrary to aPC, the number of Tu units responding to more than one CS increased both for task-excited and task-inhibited responses (Fig. 7d–f). As expected for learned categorization of stimuli according to their predicted outcome value, the increase was caused by shared responsiveness to both CS50 and CS100 (Fig. 7g–i). This trend characterized all task epochs. However, we found some notable differences in the responsiveness of Tu units during the CS and the subsequent waiting interval. At CS, the total number of units with shared responsiveness to CS50 and CS100 was smaller than the sum of units responsive selectively to CS50 or CS100 (Fig. 7g). During waiting, instead, the large majority of recruited units shared responsiveness to CS50 and CS100 (Fig. 7h). This difference could suggest the presence of distinct functional Tu sub-populations active at different task epochs, and heterogeneity in the RP coding scheme during stimulus presentation and the subsequent anticipation of reward while waiting.

To investigate the existence of distinct functional subpopulations in Tu, we clustered neurons of trained animals according to their task activation profiles (Fig. 7j). Tu units divided into three major groups. Task-excited units with transient activity at CS (transient cluster, Fig. 7k), task-excited units with ramping activity during the waiting window (ramping cluster, Fig. 7l), and units with task-inhibited responses (inhibited cluster, Fig. 7m). The transient cluster encoded monotonic RP at CS, which decayed shortly after the odor presentation had ceased (Fig. 7k and Supplementary Fig. 8d). Within the ramping clusters, instead, monotonic RP coding evolved during waiting (Fig. 7l and Supplementary Fig. 8e). In contrast to the task-excited clusters, the average firing rate of the inhibited cluster did not significantly encode a monotonic RP (Fig. 7m and Supplementary Fig. 8f). These findings suggest that stimulus-triggered and anticipatory RP were encoded by parallel task-excited neuronal populations in the Tu.

**Transient and ramping units differently encode monotonic RP in Tu**. To better understand the coding mechanisms employed by such functionally distinct neuronal clusters, we examined whether the RP was computed in single neurons, whose rate response reflects the full set of reward probabilities, or only at the population level. While more than half of the units in the ramping clusters displayed a monotonic RP coding (Fig. 8a, b), in the transient cluster only a quarter did so, and exclusively during

odor presentation (Fig. 8a, b and Supplementary Fig. 9a). Considering that the CS-bound value coding units are reinforced from pre-existing odor-specific responses[28], we wondered whether the single units not encoding monotonic RP still contributed collectively to it. Indeed, upon removal of all units with monotonic RP coding in the transient cluster, the cumulative firing rate of the remaining units still reflected a robust population coding of the monotonic RP (Fig. 8c). This was not the case in the ramping cluster (Supplementary Fig. 9b). Thus, while the ramping cluster encoded the full information at the single-unit level, the transient cluster employed largely a distributed coding scheme. Only a few task-inhibited units individually encoded a monotonic RP (Fig. 8a and Supplementary Fig. 9a). In contrast to Tu, only a small fraction of all aPC units encoded monotonic RP above chance level (Supplementary Fig. 9c), with the exception of a few units from the task-inhibited cluster. In fact, the occurrence of units with response intensities monotonically ordered (CS100 > CS50 > CS0 and CS0 > CS50 > CS100) passed the chance level for all cluster groups in Tu (while no other intensity permutation did), but only for the task-inhibited cluster groups in aPC.

To understand whether the Tu units with transient CS-bound value coding and those with ramping anticipation are differentially involved in prediction error computations at US, we tested if single units encoded reward surprise and if they differentiated outcomes between reward and non-reward in CS50 trials (Fig. 8d). Reward surprise was defined by a larger rate change in rewarded CS50 than in CS100 trials from the waiting period to the US response. Note that negative surprise for unrewarded CS50 trials was not tested because phasic negative responses to US from the wait plateau cannot be distinguished from the progressive passive return to baseline. Approximately half of the units in both the transient- and ramping clusters significantly encoded either reward surprise or outcome discrimination, with a large fraction of them significant for both discrimination and surprise. It is important to note that ramping neurons had a more sustained and pronounced response to reward than transient neurons (cf. Fig. 7k, l), both relative to baseline (Supplementary Fig. 8d, e) and to the anticipatory ramp (Supplementary Fig. 8g). In the task-inhibited cluster, only a minority of units displayed surprise or outcome discrimination at US (Fig. 8d). In the aPC, roughly a quarter of units in all response clusters uniformly encoded outcome discrimination or reward surprise (Supplementary Fig. 9d).

In summary, two parallel populations in the Tu—both encoding the prediction error—distinguished for the coding of RP in different task components: The CS-bound RP was computed with the information distributed in single neurons responsive to specific CS, while anticipatory RP was computed as full information already in single neurons with ramping activity in anticipation of reward.

**Reward prediction is updated by its recent outcome history**. We finally examined how the multiple circuits encoding RP in the Tu and aPC updated according to the recent outcome history. We divided the CS50 trials based on the outcome of the preceding CS50 trial (Fig. 9a). To compare this effect with the unselective effect of recent rewards associated with other CS (termed here "satiety"), we also examined the modulation of expected value in CS50 trials depending on whether the previous trial was CS100 or CS0 (Fig. 9b). We found that during waiting, anticipatory licking following CS50 was positively modulated if the previous CS50 was rewarded, integrating information of at least four prior CS50 trials (Fig. 9c–e), even though other trial types were interspersed (Supplementary Fig. 10a). Satiety had, instead, the opposite effect on the licking behavior. The prior experience of either a CS0 or a

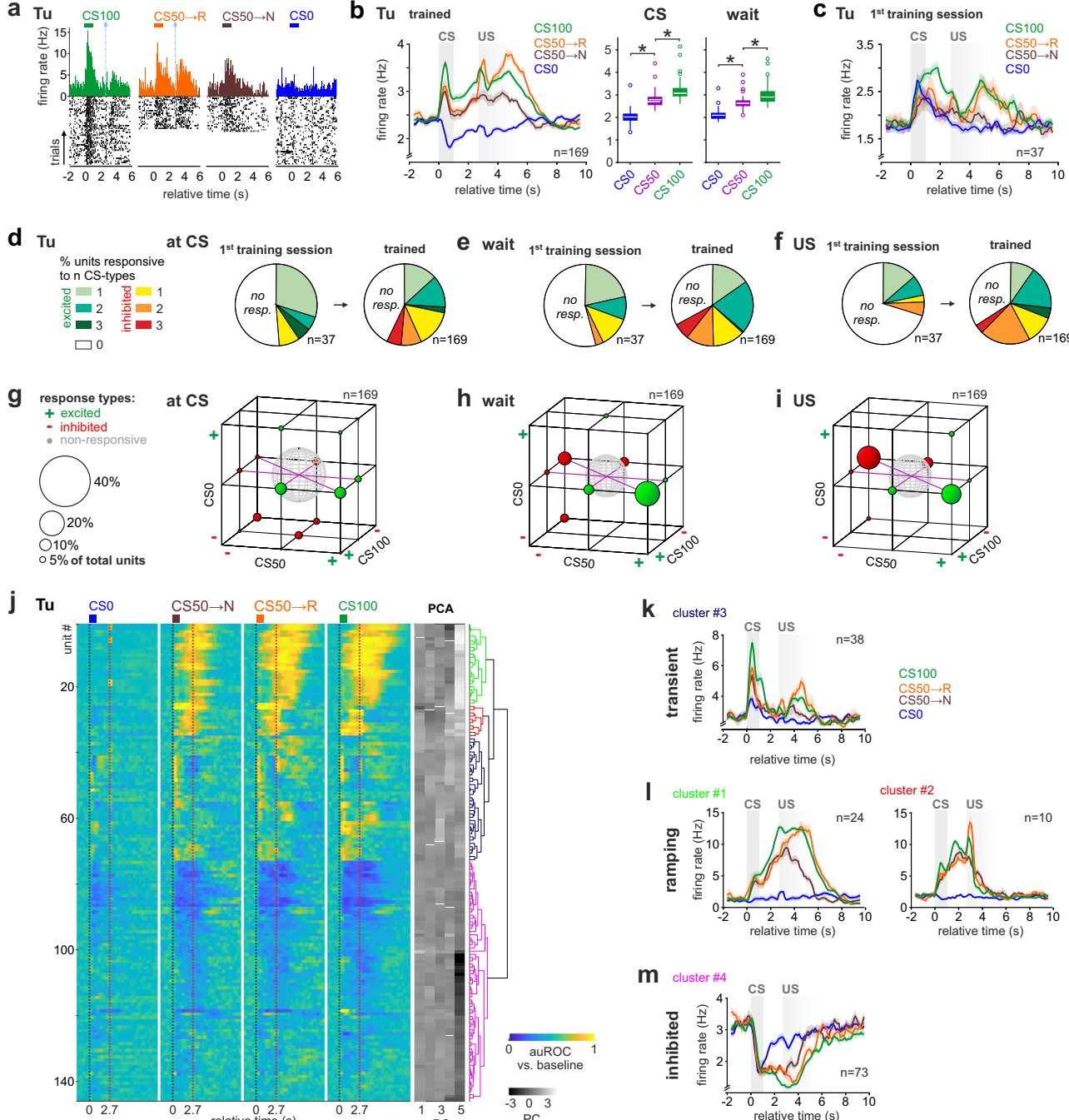

**Fig. 7 Tu units form transient and ramping task-response clusters. a–c** Same as Fig. 6a–c for Tu. After training, the task-related mean firing rate evolution ± SEM in Tu units reflected monotonic RP and reward surprise (*n* = 169 units). The bounds of the boxes represent 25th to 75th percentiles. The center indicates the median. The lower and upper whiskers represent the minimum and maximum values without outliers; outliers indicated by circles. * indicates *P* < 0.05 (see Supplementary Table 1 for exact *P* values and test details). Source data are provided as a Source Data file. **d–f** Fraction of single units showing significant task-excited, task-inhibited or no responses to either one or more trial types (CS0, CS50, CS100) at CS (**d**), during wait (**e**), and at US (**f**). Of note, the fraction of single units responsive to more than one trial type increased with training. **g–i** Fractions of selective and shared unit responses to different trial types (**g**) at CS, (**h**) during wait, and (**i**) at US in the Tu. The size of the spheres corresponds to the respective fraction of units. Green color indicates task-excited responses (+), while red color stands for task-inhibited responses (−). Note the shared coding of rewarded trial types during wait and at US. **j** Same as Fig. 6g for Tu. In the Tu, PCA-based hierarchical clustering revealed four task-response clusters. **k–m** Same as Fig. 6h–j for Tu. Mean firing rate ± SEM for each cluster identified in (**j**). According to their responsiveness at CS and during wait, single units were grouped into three functional clusters composed of units with (**k**) task-excited, transient activity at CS, (**l**) task-excited, ramping activity during waiting, and (**m**) task-inhibited responses. Both the transient- and ramping clusters displayed a monotonic RP (see Supplementary Fig. 8d–f). *n* indicates the number of units.

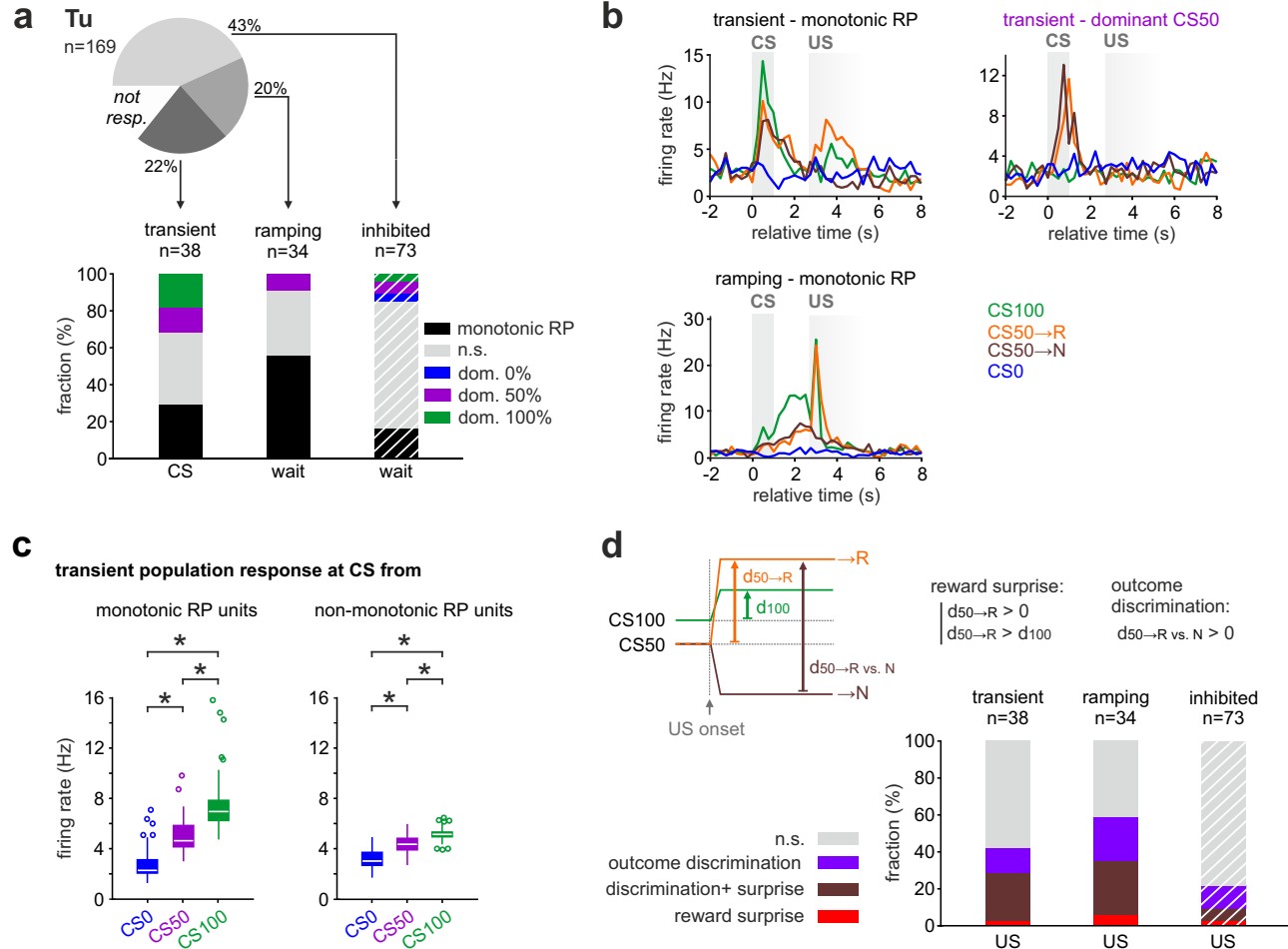

**Fig. 8 Transient and ramping units differently encode monotonic RP in Tu. a** (Top) Percentage of Tu units in the three major response clusters from Fig. 7k–m. (Bottom) Percentage of units in each cluster coding monotonic RP or showing a dominant response to one of the CS (test window indicated below each bar, see Supplementary Fig. 9a for test on complementary windows). Note that more than half of the units in the ramping cluster displayed monotonic RP during wait while only a quarter from the transient cluster did so at CS. RP and CS coding were tested via two-tailed $t$ test with significance threshold at $P < 0.05$. Dashed lines indicate that the task-inhibited cluster was tested for reduction in firing rate (see "Methods"). **b** Example units illustrating (upper left) monotonic RP or (upper right) dominant CS50 activation at CS in two units from the transient cluster, and (lower left) monotonic RP during wait of a unit from the ramping cluster (displayed average firing rate). **c** Median firing rate at CS of the units in the transient cluster that encoded individually positive monotonic RP (left; $n = 11$ units) or non-monotonic (right; $n = 26$ units) responses (one-way ANOVA with Tukey post hoc comparisons). Units not individually coding monotonic RP still displayed a robust monotonic RP as a population. Thus, the transient cluster displayed monotonic RP by distributed coding. For ramping clusters, see Supplementary Fig. 9b. The bounds of the boxplot represent 25th to 75th percentiles. The lower and upper whiskers represent minimum and maximum values without outliers; outliers indicated by circles. * indicates $P < 0.05$ (see Supplementary Table 1 for $P$ values and test details). Source data are provided as a Source Data file. **d** Scheme illustrating reward surprise and outcome discrimination (top). Units were categorized as coding for reward surprise if during CS50 rewarded trials their increase in firing rate from before to after US ($d_{50 \to R}$) was positive and bigger than that in CS100 trials ($d_{100}$). Units were categorized as coding for outcome discrimination if the firing rate at US was higher for rewarded than for unrewarded trials ($d_{50 \to R \text{ vs. } N} > 0$). Tests were performed via two-tailed $t$ test with a significance threshold at $P < 0.05$. (Bottom) Fraction of units from the three Tu clusters with reward surprise or outcome discrimination or both. Dashed lines indicate that the task-inhibited cluster was tested for the reduction in firing rate (see "Methods"). In the figure: $n$ indicates the number of units.

CS100 trial increased or decreased, respectively, the intensity of anticipatory licking in CS50 trials (Fig. 9c, f).

We then tested whether the task-related global population activity would also reflect such outcome history. Confirming the results from the analysis of the BOLD responses, we observed that recent rewards in CS50 trials positively modulated future CS50 responses in the Tu (Fig. 10a, left), but not in the aPC (Fig. 10a, right). In contrast, satiety did not change the population response in Tu (Fig. 10b, left) but negatively modulated the response to CS50 in the aPC (Fig. 10b, right and Supplementary Fig. 10d, e). While monotonic RP was encoded by both transient and ramping Tu populations, RP updates by the cue-specific outcome history

were significant only in the ramping population during waiting (Fig. 10c, d and Supplementary Fig. 10b). Notably, in the ramping population, satiety had the opposite effect of the cue-specific history (Supplementary Fig. 10c).

In summary, the ventral striatal circuit performed parallel computations of the RP in two neuronal populations. One population responded to CS and provided a stable representation of the RP. The second population encoded the anticipated RP during the waiting period before US and integrated the expectation update based on the recent cue-specific outcome history. Cue-specific updating of the RP differed in its direction from satiety that modulated more prominently the aPC.

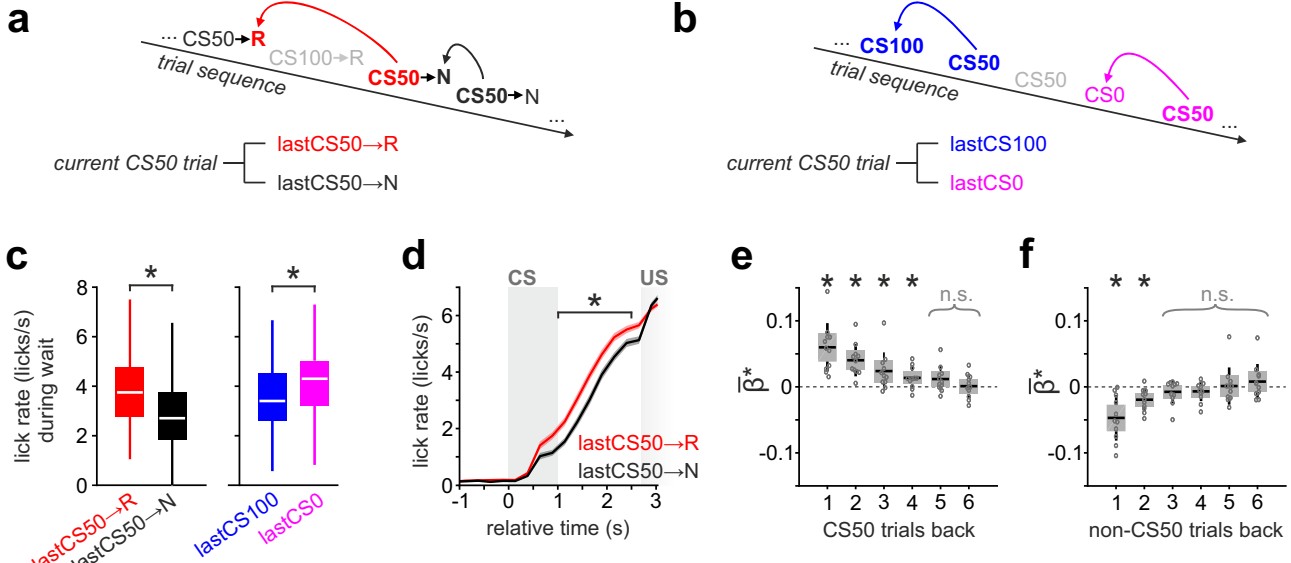

**Fig. 9 Updating of reward behavioral anticipation through the recent cue-specific outcome history. a** Scheme illustrating the division of CS50 trials based on their recent reward outcomes to test for dependence from "cue-specific outcome history". CS50 trials were grouped according to the outcome of the previous CS50 trial being rewarded (lastCS50→R) or not (lastCS50→U). **b** To assess how the dynamic value updating in CS50 trials is influenced by "satiety", i.e., reward outcomes paired to other stimuli, we separated CS50 trials depending on whether the preceding trial was CS0 or CS100. **c** Lick rate during wait in CS50 trials was modulated by the outcome of the preceding trial ($n = 88$ sessions in 11 animals; two-tailed paired $t$ test). Note that receiving a reward has a differential effect on the lick rate during the next trial depending on the stimulus identity. **d** Licking intensities differentiated CS50 trials throughout the waiting interval depending on the outcome of the previous CS50 trial (two-tailed paired $t$ test). Displayed average lick rates ± SEM. **e** Multiple Poisson regression with anticipatory licks in CS50 trials (during delay window) as a dependent variable and the outcome of the six previous CS50 trials as independent variables (one-sample $t$ test on the resulting beta, $n = 88$ sessions in 11 animals). Anticipatory licking in CS50 trials was positively modulated by the outcome of the last four CS50 trials. **f** Same as (**e**) but with the outcome of the six previous CS100 and CS0 trials as independent variables. Anticipatory licks in CS50 trials were negatively modulated by the outcome of the prior non-CS50 trials. In the figure, * indicates $P < 0.05$ (see Supplementary Table 1 for exact $P$ values and test details). Box plots: The bounds of the box represent 25th to 75th percentiles. The center indicates the median. The lower and upper whiskers represent the minimum and maximum values without outliers; outliers indicated by circles. Source data are provided as a Source Data file.

## Discussion

The hierarchical approach developed in this study may serve as a discovery strategy for key circuits in complex behavioral tasks and to reconcile findings in selected brain regions at the system level. The hierarchical approach starts from behavioral modeling to then identify task-relevant brain circuits with functional imaging. Upon such regional identification, the cellular and network coding mechanisms are studied at finer grain resolution by means of electrophysiological recordings; thereby providing a systematic cross-scale integration from behavior, over mesoscale networks, to cellular functions.

Functional MRI in awake rodents offers the opportunity to assess the functional recruitment of deep and superficial circuits during task performance, and the absence of sedation/anesthesia reduces confounds to the hemodynamic response. Several aspects are critical when interpreting BOLD responses. Seminal work has provided insights into the neuronal basis of BOLD responses to isolated sensory stimuli in animals[45–47]. Even though BOLD changes reflect activity patterns of projection neurons, in some regions, such as the striatum, BOLD and rate changes can have an inverse relation[48,49]. Moreover, little is known about the relation between population rate coding and BOLD signals during complex tasks. BOLD responses correlate with regional blood flow that is regulated for instance through nitric oxide synthesized by interneurons in an activity-dependent fashion[50–52]. These interneurons can, but do not need to, be positively correlated with principal neuronal activity[50]. It is, therefore, possible that inputs from sensory or reward-coding regions may differentially inner-vate and drive these interneurons. These factors may eventually generate the observed opposing BOLD contrasts for the RP and prediction error components, even though principal neurons responded to both with the same direction of firing rate changes. Therefore, when using fMRI as a localizer for task-related activity as in this study, both positive and negative BOLD associations can be informative but, at present, cannot serve to predict the specific character of the underlying neuronal activity.

Several factors need to be considered in (behaving) fMRI. Even though the mouse hemodynamic response function is several-fold faster than in primates, it will only partially separate sequential task events that occur typically in the range of seconds. In our experiment, image acquisition and trial onsets were temporally non-aligned in order to increase the effective sampling frequency. Nonetheless, low temporal resolution remains a general limitation of fMRI, and is especially relevant in paradigms where two events (like CS and US) are presented in close temporal proximity. Further, fMRI has a relatively low signal-to-noise ratio even at high field strength, which we aimed to compensate with repeated imaging and an optimized training scheme to generate sufficient cohort sizes. Importantly, Echo Planar Imaging is sensitive to motion-induced distortions of the magnetic field, evoked for instance by head movement during licking activity and chest movement during respiration. Superficial lateral and ventral brain structures like Tu and aPC are particularly susceptible to the effects of head and jaw motion. We employed a combination of denoising methods to minimize head-motion effects (see the section "Functional MRI denoising" for a detailed discussion). The motion level that we observed in the behavioral task, is expected given data of recent awake resting-state imaging in

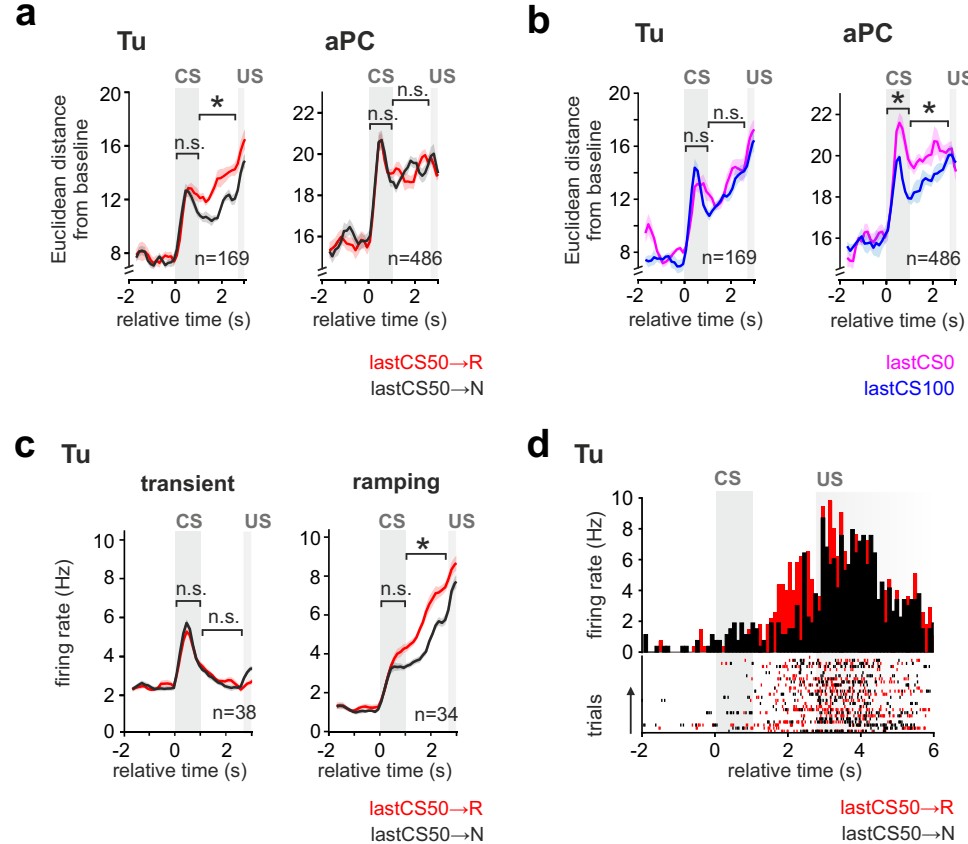

**Fig. 10 Updating of the reward prediction coding through the recent cue-specific outcome history. a** Mean Euclidean distance from baseline ± SEM of the population vector during the lastCS50→R and lastCS50→N trials. If the previous CS50 trial had been rewarded, Tu population displayed a stronger response than if it had been unrewarded (left). In contrast, no recent outcome-history effect was observed in the aPC population (right). **b** Same as (**a**) but dividing CS50 trials based on whether the prior trial was CS100 or CS0. Satiety did not change the population response of Tu at CS and during wait (left), but reduced the population responses of aPC (right). One-way ANOVA on Tu history and satiety combined (left graph) during: CS, $F_{(3,28)} = 0.3$, $P = 0.8$; wait, $F_{(3,28)} = 10.9$, $P = 6.4$ e-5; Tukey post hoc, $P$ (history) = 7.1 e-5, $P$ (satiety) = 0.4. One-way ANOVA on aPC history and satiety (right graph) during: CS, $F_{(3,28)} = 8.4$, $P = 3.7$ e-4; Tukey post hoc, $P$ (history) = 0.6, $P$ (satiety) = 3.1 e-4; wait, $F_{(3,28)} = 4.4$, $P = 1.2$ e-2, $P$ (history) = 0.5 $P$ (satiety) = 0.01. **c** Mean firing rate ± SEM during lastCS50→R and lastCS50→N trials for the transient- and ramping clusters of Tu. The ramping clusters encoded outcome history during the waiting interval, while the transient cluster showed no effect. One-way ANOVA on history and satiety (see Supplementary Fig. 10b, c) of the transient cluster (left graph): CS, $F_{(1.2,44.1)} = 0.2$, $P = 0.7$; wait, $F_{(1.6,59.7)} = 0.9$, $P = 0.4$. One-way ANOVA on history and satiety (see Supplementary Fig. 10b, c) of the ramping cluster (right graph): CS, $F_{(1.9,62.1)} = 16.9$, $P = 2.0$ e-6; Tukey post hoc, $P$ (history) = 0.2, $P$ (satiety) = 1.8 e-4; wait, $F_{(2.3,75.3)} = 20.9$, $P = 1.4$ e-8, Tukey post hoc, $P$ (history) = 2.7 e-7, $P$ (satiety) = 0.02. See Supplementary Fig. 10b–e for a systematic analysis of outcome history and satiety in all response clusters of Tu and aPC. **d** PSTH and raster plot of an example Tu unit from the ramping-cluster coding for outcome history. In the figure: * indicates $P < 0.05$, $n$ indicates the number of units.

mice[36]. Optimizing the head fixation, and modified acquisition techniques (such as continuous field mapping[53,54]), may further aid in reducing these effects. Nevertheless, active reward retrieval is an essential component of reinforcement-learning tasks. Therefore, in many paradigms including ours, the artifacts caused by jaw movements, swallowing or breathing, are correlated with the effects of interest, limiting the possibility to completely remove motion effects from the BOLD data (see also "Methods"). Similarly, while we aimed at separating the neuronal correlates of motor activity from that of reward expectation/experience (by including licks as an additional regressor), we cannot assume a perfect separation between these two processes, due to the correlation between reward expectation/consumption and licking activity. However, in support of that motion does not explain the CS responses, the BOLD response patterns were preserved when only trials with low motion were included (see the section "Functional MRI denoising" and Supplementary Fig. 4). In summary, functional MRI served here as a discovery tool, and the limitations discussed above highlight the importance of

corroborating and disentangling regional BOLD recruitments with the electrophysiological recordings.

Through the mesoscale fMRI approach, we found that RP and prediction error components were differentially represented among the primary olfactory cortices, subcortical circuits, and higher-order brain regions. The identified regions formed a larger ventral functionally connected odor-reward-association-learning network with some notable differentiations. While forebrain regions broadly contributed to value-related information, the complete monotonic RP involved a narrower network. In the olfactory regions, monotonic RP was represented partially in the olfactory bulb. Learning to predict outcome has been shown to modify neuronal representations in the olfactory bulb, possibly through top-down mechanisms[55–57]. Olfactory cortices segregated in their respective contributions to task computations, despite their shared input from the olfactory bulb and the dense reciprocal connections between the anterior olfactory nucleus and aPC. As detailed by single-unit recordings, while in Tu the CS-excited mean rate responses developed monotonic value coding,

such re-arrangement was less prominent in the aPC. Rather, aPC appeared to partially preserve the influence of initial odorant features expressed before learning, in line with the known role of aPC in stimulus identity representation. Compatible with this, functional connectivity of the aPC was notably similar across CS associated with different reward probabilities. During the waiting period, however, task-inhibited monotonic RP responses emerged at the cellular level.

Functional segregation was also observed in higher-order cortices. The mPFC and agranular insular cortex contributed to monotonic RP, but differed in their contribution to the prediction error. While the insula correlated both with the value and reward components of the prediction error at US, the mPFC did not correlate significantly with prediction error in our task. Prefrontal and insular regions prominently project to the lateral striatum and NAc as well as the Tu[23,31]. In the Tu, additional direct inputs from the olfactory cortices and the olfactory bulb converge[23,31]. Tu and NAc represented all RP and prediction error aspects in the olfactory reinforcement-learning task, including dynamic value updating. In line with anatomical input gradients from sensory and reward regions[58], posterior parts of the Tu correlated with the reward component of the prediction error at US. Also, prediction error components were localized in the lateral NAc shell, consistent with its contribution to positive reinforcement[59]. Finally, the dorsal striatum similarly reflected CS-bound value signals, but was dominated by the reward component of the prediction error at US. There is an important difference between the dorsal and ventral striatum. While the dorsal striatum is known to encode the value of particular actions given a particular CS, the ventral striatum has been suggested to encode the expected reward value of CS[12].

Supporting the translatability of these fMRI findings in mice, conditioning in human fMRI studies recruited homologous prefrontal and ventral striatal regions involved across stimulus modalities[9,11,12,40,41,60] as well as modality-specific correlates in olfactory cortices or insular regions[17–21]. The largely non-overlapping forebrain representations of reward and value components of the prediction error at US observed here match those in humans[40,41]. Consistent with humans, in our study the mouse NAc represented both the reward and the value components of the prediction error at US. The Tu, which also represented both prediction error components at US, has rarely been explicitly examined in humans, but a recent fMRI study highlights its unique role in appetitive odor representations[61]. Here we show that mouse Tu displayed a CS- and US-sensitive functional connectivity with the other brain regions while contributing to multiple facets of stimulus-outcome learning.

Recent studies have revealed that Tu neurons are recruited by stimuli that predict rewards upon conditioning[22,26–28,31]. In these studies, the duration of the conditioned stimulus partially overlaps with US. To better capture reward anticipation, in the present study, we separated the CS and US by a waiting gap. Upon learning, we observed the evolution of two prominent task-excited populations of Tu units computing either a transient stimulus-bound RP or a ramping anticipatory RP during the waiting period. The transient stimulus-bound RP signal was encoded by the sum of distributed odor responses of multiple units, many of which were selective for only one odor, but that collectively encoded monotonic RP. This RP signal emerged at CS after cross-session learning and was little influenced by the recent outcome history or satiety; consequently, generating a stabilized representation of the RP. The stability of the RP representation at CS may relate to its formation through synaptic plasticity[62]. In fact, the stimulus-response potentiation evolves gradually upon repeated pairing of the odor with phasic dopamine in awake mice[28]. Once emerged, these RP-coding CS signals in the Tu are

in a position to drive downstream RP responses of midbrain dopamine neurons (DAN)[28].

In contrast to this CS-bound signal, the ramping anticipatory RP signal evolved during the waiting interval after the stimulus has ceased. Ventral striatal anticipatory ramping activity is also found during waiting in head-fixed primates[63] and thought to reflect anticipatory timing to reward[64,65] informing prefrontal cortices[66]. Within the Tu, the ramping RP signal was encoded redundantly, with multiple single units responding to all rewarded odors proportionally to their associated value. Value update was fast and stimulus-specific, in line with the modeled RP. The faster value updating of the ramping population could be interpreted as a neuronal implementation of the TD model, where values more proximal to the prediction error at US are more quickly updated than the distal ones at CS. While stimulus-bound RP responses drive DAN at CS, this might not be the case for Tu ramping activity. In fact, DAN do not express anticipatory ramping in head-fixed trace conditioning[14,15] but only when animals approach rewards in space[15,16,67]. Finally, the enhanced responsiveness to reward of the anticipatory ramping population compared to the CS-bound RP population adds to their differentiation. Potentially, such differentiations could map on the anterior–posterior gradients for a prediction error value and reward representations found here, different olfactory and reward-related projection gradients[58,68], or direct- and indirect-pathway Tu neurons[27]. Taken together, the different coding strategies, the non-overlapping neuronal clusters, the functional dissociation of the two RP signals from DAN, and the different responsiveness to reward, support the presence of parallel networks for stabilized and dynamic reward prediction in Tu.

The hierarchical approach presented here, with its cross-scale analysis, can be applied broadly to reveal the computations in brain networks and can foster the discovery of key brain regions and mechanisms in cognitive and translational sciences. Here, the approach identified the olfactory tubercle of the ventral striatum as one of the key circuits to compute multiple non-redundant olfactory reward predictions in parallel networks of projection neurons. While the CS-bound stabilized RP signal provides an initial safe estimate to drive dopamine midbrain neuron coding, the subsequent anticipatory RP dynamically integrates recent experiences in preparation of reward retrieval. This olfactory prediction coding hub operates within a network identified by fMRI of functionally segregated olfactory and higher-order regions.

## Methods

**Animals and husbandry.** Ten-week-old male C57BL/6N mice were obtained directly from Charles River Laboratories (23 animals for the fMRI measurements, 11 animals for single-unit recordings). Food and water were given ad libitum, except when water supply was controlled for behavioral training. Mice were housed individually at a 12' day-and-night-cycle (room temperature 24 °C, air humidity 55%). Mice were 3–6 months old when recordings and fMRI were performed. All procedures were in accordance with the National Institutes of Health Guide for the Care and Use of Laboratory Animals and the EU 2010/63 directive, and approved by the local animal welfare authority (Referat 35, Regierungspräsidium Karlsruhe, Karlsruhe, Germany). Only mice of one sex were used in the study to minimize confounding effects of sex-related distress or distraction elicited by residual scents in the recording apparatus and MRI cradle.

**Implantation of the head bar.** All surgeries used standard aseptic procedures and conformed to common veterinary practice. Analgesia (meloxicam, Metacam Boehringer Ingelheim) was administered before and after surgery. At least 12-week-old mice were anesthetized with isoflurane. The animals were then transferred to a stereotactic apparatus with nonrupturing ear bars and placed in a custom-built platform of the same dimension as the MRI cradle. It was ensured that the fixed licking spout of the MRI cradle was at the same height as the animal's lower lip at a distance of 5 mm. A roughly circular flap of skin was removed from the skull and lidocaine was administered topically. The lateral and nuchal muscles were left intact. The skull was cleaned and disinfected. Tissue adhesive (3 M Vetbond) was applied to the margins of the skin attached to the circumference of

the exposed skull to avoid soft tissue damage and contamination. The remaining periost on the exposed skull was removed. A layer of dental glue (C&B Superbond, Sun medical) was applied on the (inter)parietal bone, followed by a layer of dental cement (Kulzer Palladur) that connected the custom-designed head bar produced with stereolithography (Accura55, 3D-Systems). Animals received additionally a subcutaneous injection of buprenorphine directly after surgery. All animals recovered within 10–20 min after cessation of isoflurane anesthesia. Thereafter, they were transferred to their home cages in good general condition. If signs of post-surgery pain were observed, additional meloxicam was administered. All animals were monitored daily during the entire experiment.

**MRI-compatible behavioral setup**. We developed an MRI-compatible setup for the odor-guided reward learning task (see Fig. 3 and Supplementary Figs. 1 and 2). The apparatus comprised of an MRI cradle, an olfactometer, a programmable syringe pump for reward delivery (AL4000-220, World Precision Instruments), an optical licking detector, and Arduino microcontrollers (Arduino Mega 2560). Odors were delivered using a custom-made olfactometer. Odors were kept in the liquid phase (diluted 1:100 in mineral oil) in dark vials and mixed into a nitrogen stream that was diluted by 1:10 into a constant air stream in the olfactometer. The following natural flower odors were used: geranium, ylang–ylang and rose (Sigma Aldrich W250813, W311936, and W523704, respectively). Water as reward was delivered through an independent tubing system and controlled by a high-precision water pump. Odorized air and water were both guided from the setup located in the control room to the animal bed inside the magnet bore through 1/32 inch I.D. inert Polytetrafluoroethylene tubing (NResearch) connected to the odor and lick ports. Odor valves and syringe pumps were controlled through Arduino microcontrollers. A single delivery tubing was used for all odors. For all MRI experiments, a 3.0 m long tubing was used to deliver the odorized air to the odor port. For this tubing, the latency for odor detection at the odor port after the final valve opening at the olfactometer was determined initially with 1:10 diluted amyl acetate outside the scanner with a photo-ionization detector (miniPID, Aurora-Scientific). The latency was ~400 ms. Thus, air was exchanged in the tubing by clean air at least 25 times between two consecutive odor applications (considering the airflow rate of the olfactometer). The steepness (time from 10 to 90% of peak) of the odor onset was ~20 ms and the offset approx. 40 ms (time from 90 to 10% of peak). The olfactometer was placed in the MR control room and the odor port placed in front of the snout of the animal head fixed on the MRI cradle (Fig. 3a). Animals were head-fixed without the need of sedation or anesthesia.

Licking behavior was measured using a custom-made MRI-compatible optical lickometer. Infrared light was delivered via fiber optics and miniature roof prisms from an LED source (Thorlabs M660F1; 660 nm), collimated at the lick port with lenses (Thorlabs 354140-B) and returned to the control room via another optic fiber to the detector (Thorlabs DET10A2). Animals were positioned so that, when licking, their tongue broke the beam on each lick. Output signals from the olfactometer, the optical lick detector, the water pump, and TTL pulses from the scanner were all recorded with the same RHD2000 interface board (Intan Technologies) with a sampling frequency of 1 kHz. These data were used to align trial times during analyses. For training outside the scanner, a custom-made simplified MRI cradle was used.

**Behavioral training**. Mice were trained in an odor-guided reward learning task to learn associations of stimulus-outcome pairs. Three days before behavioral training, water intake was controlled in their home cages (90% of baseline bodyweight was targeted). Bodyweight was monitored daily and always maintained above 85% of baseline bodyweight. Mice were placed in the head-fixed setup for habituation. The habituation sessions did not exceed 15 min. In general, mice habituated to head fixation after 2–3 days. Then, conditioning sessions started. Each trial started with 1 s of odor presentation followed by a waiting period of 1.7 s. Reward (5 µl water) was delivered immediately after the waiting window (see Fig. 1a, middle). Reward timing and reward size were not varied. Licking responses had no influence on whether reward was delivered or not. The trial duration was randomly drawn from a uniform distribution between 10 and 12 s. The training is comprised of two stages. In Stage 1, a single odor was presented and rewarded at 100%. When mice licked consistently, they progressed to the next stage. Stage 2 corresponded to the final paradigm and consisted of three distinct odors delivered in a pseudo-random order to keep the proportion of odors constant between sessions. No stimulus was consecutively applied more than three times in a row. No more than three consecutive trials were rewarded. Animals performed 150 trials per session (in the MR-environment, the first 20–30 trials of which were regarded as for acclimatization). Odor cues predicted reward at 100, 50, and 0% (hereafter, the odor cues were labeled CS100, CS50, and CS0, respectively) (see Fig. 1a). Across sessions, the odor and reward contingency pairs were unchanged. The first session of Stage 2 was considered the "first training session" in Figs. 6 and 7. Generally, CS100 and CS50 were treated as "Go" cues and odor CS0 as a "No-go" cue. If mice licked at least three times during the anticipatory window (from 1.5 to 2.8 s after odor onset) or in the reward window (from 2.8 to 4.1 s), it was considered as a go-response. Fulfilling the lick criterion during "Go" trials was regarded as "Hit", while it was assessed as "False alarm" during "No-go" trials. Not meeting the lick criterion was regarded as "Correct rejection" for "No-go" trials and as "Miss" for "Go" trials. There was no punishment for false choices. The performance within each session was calculated as (in percent):

$$\text{Performance} = \frac{\text{number of 'Hit' trials} + \text{number of 'Correct rejection' trials}}{\text{total number of trials}}$$
(1)

After mice designated for functional imaging reliably performed the task above the criterion (>80% performance, normally after 6–12 training sessions), a sham coil was placed above the head and recorded MRI pulse sequence noise was replayed in 'mock scanning sessions' to acclimate mice to the MRI environment. Sound levels in the mock scanner were increased gradually to the noise levels in the scanner bore. In the behavioral analysis, only sessions with performance higher than 80% were included.

Both human and animal subjects are expected to experience elevated stress levels in MRI due to limitations of movement and scanner noise. Thus, in general, it can be assumed that learning in the scanner occurs under elevated stress conditions. As described above, habituation to the MRI environment and noise served to reduce physiological stress parameters and motion[69,70]. The efficacy of the habituation procedure is supported in this study by the comparable behavioral performance observed inside and outside the scanner (Supplementary Fig. 1a, f).

**Pupil imaging**. Pupils were imaged unilaterally in 10 trained mice (one session per mouse). Pupil data was collected in the recording chamber with ambient light illumination (blue LED, 465 nm). To fully capture the pupil dynamics during task performance, the intensity of the LED light was set so that the pupil was moderately dilated. DinoCapture 2.0 software was used for video recording. Videos were acquired at 20 frames per second (1280 × 1024 pixel) with a digital infrared USB camera (AD4113T-I2V Dino-Lite Pro2 digital microscope) providing infrared illumination by LEDs (940 nm). Infrared illumination did not affect pupil diameter. At the beginning and at the end of each session, an additional infrared LED was switched on for 1 s in order to generate timestamps. The timestamps were then used for post hoc alignment of the pupil and behavioral data.

**Pupil diameter analysis**. Pupil diameter was detected frame-wise using in-house developed MATLAB scripts based on a pulse-coupled neural network (PCNN) algorithm. Briefly, pupil videos were read into MATLAB and converted to grayscale. Then, pupil videos were automatically centered and cropped around the pupil. Based on the adjusted frames, the PCNN algorithm calculated binary images reliably segmenting the pupil from the surrounding tissues. A convex hull was fitted to the contour of the detected pupil area. The pupil radius was detected for each frame by calculating the mean distance between all points on the convex hull and its midpoint. Outliers in the detection, e.g., during blinks, were excluded from the data by removing the entire trial (exclusion criterion: increase or decrease in pupil diameter of more than 5% within two frames). The accuracy of the algorithmic fit was verified visually for each session. Pupil diameter was expressed as a percent change from baseline. The baseline was derived from the average pupil diameter in the time window from −2 to 0 s relative to odor onset. Mean responses were calculated for each trial type for each animal and, in turn, averaged to get the group response ± SEM. Pupil data were tested for normality using a Kolmogorov–Smirnov test. An ANOVA test was performed between CS100, CS50, and CS0 trials during the waiting window (from 1 s to 2.5 s relative to odor onset).

**Reinforcement-learning model**. We parametrized the RP and prediction error with a temporal difference model TD(0) with no eligibility trace or discount factor[71]. The model was designed on a trial-by-trial basis. In each trial, we considered three timepoints: the baseline before the trial started ($t = 0$), the odor presentation ($t = 1$), and the reward delivery ($t = 2$). Each timepoint was characterized by a state $s$ defined by the trial epoch and, when present, the co-occurring stimulus, for a total of five states. At $t = 0$, there was only one possible state $s_0$. At $t = 1$, there were three possible states $s_1$ corresponding to the three conditioned stimuli (CS100, CS50, CS0). At $t = 2$, there was only one state $s_2$ representing the moment in the trial associated with the unconditioned stimulus (the specific presence/absence of reward in each trial is captured in the model by the separate variable $r_t$, see next paragraph).

To each state $s_t$ was assigned a value $V(s_t)$, reflecting the RP. Further, an outcome $r_t$ was assigned to each timepoint. $r_t$ encodes the reward at time $t$ as a binary variable 1 or 0. At baseline and during CS presentation, respectively, $r_0$ and $r_1$ were set to 0. At US, $r_2$ was 1 in case of reward and 0 otherwise. $V(s_2)$ was set to 0 as the model was designed on a trial basis and $t = 2$ was the last timepoint of the trial.

The values $V(s_t)$ were updated trial by trial by a prediction error $\delta$. The prediction error compares two successive value predictions and is defined as:

$$\delta_t = r_t + V(s_t) - V(s_{t-1})$$
(2)

Intuitively, the current reward $r_t$ and the current expectation for the immediate future $V(s_t)$ are summed and compared to the preceding expectation $V(s_{t-1})$. A positive/negative prediction error $\delta$ has the effect of increasing/decreasing the value of the previous state according to:

$$V(s_t) \leftarrow V(s_t) + \alpha\delta_{t+1}$$
(3)

with learning rate $\alpha$ (in our model, $\alpha$ was set as free parameter and estimated using pupillary data, see next paragraph). For example, a reward delivery ($r_2 = 1$) after CS50 will lead to a positive $\delta_2$, and this will update (increase) the value of the CS50 stimuli $V(s_1)$.

In the main text, we refer to $V(s_1)$ as V(CS), $V(s_2)$ as V(US), and $\delta_2$ as $\delta(US)$.

**Parameter estimation and selection**. The learning rate $\alpha$ was set as free parameter of the model and estimated using pupillary data from the pupil imaging sessions described above ($n = 10$ sessions). The average pupil dilation $d$ (or more precisely the average percent change of pupil dilation from baseline) was modeled as a function of the expected value $V$ and the prediction error $\delta$:

$$d(t) = \sum_{\tau \in \mathbb{N}=0}^{t} a_\tau V(s_{t-\tau}) + \sum_{\tau \in \mathbb{N}=0}^{t-1} b_\tau |\delta(t-\tau)| \qquad (4)$$

where $t = \{0, 1, 2\}$ (0 for the baseline, 1 for the odor presentation and 2 for the reward time). $a_\tau$ and $b_\tau$ were set as free model parameters. $d(t)$ indicated the average pupil dilation at different times in the trial: we defined $d(0)$ as the average dilation in the interval from $-2$ to $0$ s; $d(1)$ as the average dilation in the interval from $0$ to $2.7$ s, and $d(2)$ as the average dilation in the interval from $2.7$ to $5.4$ s. The model parameters $\vartheta = \{\alpha, a_\tau, b_\tau\}$ were estimated minimizing the distance between the modeled (Eq. (4)) and the real average pupil dilation.

After the average learning rate $\alpha$ had been estimated on the basis of the 10 sessions with pupil imaging (estimated $\alpha = 0.28$), it was used to build a temporal difference model for 151 different trial realizations. Of those, 100 realizations were simulated on as many CS-US sequences, randomly generated with the algorithm used for the behavioral sessions, and used to produce Fig. 2a, d; and 51 were simulated on the actual trial sequences of the fMRI sessions and used in the fMRI analyses. We applied the learning rules in Eqs. (2) and (3) to model learning (Fig. 2a, b). In this way, we formulated specific predictions on RP updates and regressed on them the fMRI data of mice performing the task in the scanner to identify key forebrain regions. The initial condition of the values of all stimuli was set to 0 in the simulations of Fig. 2a, d and set to 0, 0.5, and 1 for CS0, CS50, and CS100 respectively in the fMRI sessions (as the animal were already performing when the fMRI scanning started, c.f. Supplementary Fig. 1f, h).

**Functional MRI acquisition**. Experiments were conducted on a small-animal 9.4 Tesla MRI scanner (94/20 Bruker Biospec, Ettlingen, Germany) with Avance III hardware, BGA12S gradient system with a maximum strength of 705 mT/m, and running ParaVision 6 software. We utilized a whole-body linear volume transmitter coil combined with an anatomically shaped 4-channel receive-only coil array. Mice underwent fMRI scans after habituation to the MRI environment (see above). Awake mice were head-fixed in the MRI cradle; their trunks were gently held within a plastic cylindrical tube to further prevent massive body motion. Once positioned inside the scanner bore, the task was initiated, allowing the subject to orient to the task and minimize distress. The 20–30 trials presented before the onset of the fMRI sequence were not recorded. A localizer sequence was used to confirm the correct positioning of the animal and a fieldmap was acquired. Before the fMRI scan started, the main magnetic field (B0) was homogenized by automatic shimming (map shim, shim volume covering all relevant brain regions). After shim calculation, an additional fieldmap was acquired for later correction of geometrical distortions. The fMRI time series were acquired while mice performed the odor task using an echo-planar imaging (EPI-FID) sequence with the following parameters: TR/TE: 1300/17 ms; flip angle: 50°; 21 slices; matrix size: 64 × 64; slice thickness: 0.5 mm; interslice gap: 0.1 mm; voxel size: 0.25 × 0.25 × 0.6 mm; 1400 volume acquisitions. More posterior slices frequently had variable signal dropout due to B0-field inhomogeneity; therefore, only slices from +4.2 mm to 0 mm anterior to Bregma (comprising the olfactory bulb, striatal regions, anterior olfactory, and higher-order cortices) were considered (cf. Figure 3b). Note, however, that multiple-comparison correction during statistical inference, unless otherwise indicated, was always based on the voxels in all slices originally acquired. Each EPI session lasted 24 min. The behavioral session was followed by a high-resolution T2-weighted RARE anatomical image acquisition (TR/TE 1200/50 ms, matrix size 96 × 113 x 48, voxel size 0.16 × 0.16 × 0.31 mm, RARE factor 16). To increase statistical power, animals were repeatedly measured (up to 5 sessions per animal), but not more than once per day. Mice that did not lick at reward in the scanner were excluded from further measurements to avoid unnecessary distress. This yielded a dataset of 67 sessions from 23 animals. Of these, 51 sessions from 18 animals fulfilled the performance criterion described above and were included in fMRI group statistics.

**Image data processing**. All data were processed using Statistical Parametric Mapping version 12 (SPM12) (http://www.fil.ion.ucl.ac.uk/spm/) and custom-written MATLAB scripts. After brain extraction using in-house developed code based on a pulse-coupled neural network algorithm, structural images were non-linearly normalized by segmentation to high-resolution tissue probability maps[72], which had been transformed to Paxinos space. fMRI data were preprocessed with the following steps[73]: discarding the first five volumes in the series to avoid influences of magnetization before the scanner achieves steady state, correction for head movement by realignment to the middle volume using a rigid-body

transformation, correction for geometrical distortions using the acquired B0-field maps, slice-timing correction, and spatial normalization to a mouse brain template in the Paxinos stereotactic coordinate system, by applying the nonlinear normalization parameters of the structural images to the functional images. Normalized functional images were additionally smoothed with a 0.6-mm isotropic Gaussian kernel.

**Functional MRI denoising**. Functional MRI is susceptible to movement artifacts, which is of particular concern given the association between the paradigm, licking activity and corresponding head and jaw motion. To quantify head motion, absolute values of the realignment parameters' first derivative were averaged per session for the three translations and three rotations (see Supplementary Fig. 2b). To account for motion artifacts, we explored several denoising methods. Regression of realignment parameters, which is widely applied, has several shortcomings, as it depends on the accuracy of the realignment process, does not reliably capture nonlinear or delayed effects, is anatomically unspecific and does not differentiate between task-correlated neuronally based activity and artifacts[74,75]. The latter effect is of particular concern in the context of task-based fMRI, where motion is correlated with the effect of interest. Indeed, we found that plausible gray-matter activation in response to rewarded CS types (e.g., in dorsal striatum) was substantially decreased (while areas of deactivation became more extended) when the six realignment parameters and their derivatives were included in the GLM, exclusively or in addition to the other denoising methods described subsequently.

Removal of high-motion volumes (termed "scrubbing" or "censoring") has been shown to outperform motion regression under certain conditions[76,77], but again suffers from important drawbacks, namely that it is difficult to find valid criteria for frame removal, especially in the context of task-based fMRI[77]. Further, excessive amounts of data may be removed with the corresponding loss of temporal degrees of freedom[75,78], even when delayed movement effects are not addressed. In task-based fMRI, censoring may inherently introduce bias and disproportionately affect image frames of certain kinds, such as during rewarded trials or in high motivational states.

Other approaches have been proposed that allow for a more specific removal of presumed motion artifacts, by considering their typical spatial and/or temporal characteristics. Among these methods, we explored Wavelet Despiking, which exploits the divergent frequency characteristics of motion artifacts, and despikes the BOLD timecourse locally in a temporal, spatial, and frequency sense[78]. However, Wavelet Despiking has not been systematically assessed in task-related fMRI, where the frequency spectrum may be more heterogeneous than in resting-state fMRI. Additionally, in rodents, the hemodynamic response function (HRF) is faster than in primates[38,39,79] and therefore has a weaker low-pass filtering effect, potentially allowing true neuronally based BOLD signal to contain higher frequencies. In line with this, we found Wavelet Despiking to introduce implausible temporal "smearing" such that activations started slightly before the trial. Such effects would have been of particular concern since events were relatively close in time in our paradigm.

We, therefore, alternatively, employed group-independent component analysis (ICA) for removing motion-related noise, using the fastICA toolbox (http://research.ics.aalto.fi/ica/fastica/)[37]. ICA-based denoising exploits typical anatomical patterns of motion artifacts, and has been applied in previous awake rodent imaging[36,37]. In our group-ICA, 4 (out of 25) components were manually identified as motion-related (see Supplementary Fig. 2d); their session-specific time courses were removed before reconstructing the images. Criteria for component removal were typical location at the edges of the brain and/or in and around ventricles. One widespread, anatomically unspecific component C1 in Supplementary Fig. 2d) was also removed due to high correlations with realignment parameters and with the mean cerebrospinal fluid (CSF) timecourse. ICA was performed after other preprocessing including smoothing, to maximize signal-to-noise ratio and anatomical overlap between sessions/subjects for ICA, and also in accordance with previously described ICA denoising procedures[37,75]. As a limitation of this denoising method, it should be noted that the components classified as motion-related due to their typical location at the surface of the brain and around ventricles (C2 and C4) overlap with the regions showing value- and reward-associated BOLD responses (including Tu and aPC). Given that in our paradigm, there is an inherent correlation between the parameters of interest (value and reward), licking activity and head motion, a perfect separation of neuronally based BOLD response and BOLD changes caused by jaw/head motion is, per se, not possible. Therefore, residual effects of head motion could remain in the data, or some BOLD responses reflecting true task-related neuronal activity might be falsely removed as part of the ICA components classified as motion (especially in the regions that are located superficially, including aPC and Tu).

We, therefore, tested whether the BOLD response patterns were affected by stratification into "low-motion" and "high-motion" events. We also assessed whether for the critical feature of value coding, namely the contrast of CS50 and CS100 at CS, the anatomical pattern was preserved in trials with low frame-wise displacement (FD). FD was computed for each image frame as the sum of the absolute values of the derivatives of the six realignment parameters[80]. Granjean et al. defined an FD threshold of 0.1 mm[81], and Gutierrez-Barragan et al. used 0.075 mm[82]. For our stratification, we defined low-motion CS trials by a maximum

FD of 0.05 mm in the two image frames at and following the CS timepoint. These image frames include the peak of the CS response as modeled by the fast-peaking mouse HRF. We selected all sessions ($n = 16$) with at least 10 trials (per trial type) with low motion during CS and delay. This threshold was chosen to yield a balanced number of low- vs. high-motion CS100 events, low- vs. high-motion CS50 events, and low-motion CS50 vs. low-motion CS100 events (with an average of FD < 0.05 and FD ≥ 0.05 trials of 21.5 and 18.5, respectively, for CS50, and of 18.4 and 21.6 for CS100). The low- and high-motion CS100 events were modeled by separate regressors, substituting the CS100 event regressor in GLM 2. Low- and high-motion CS50 events were modeled analogously. The BOLD response patterns to CS100 were similar between strata (Supplementary Fig. 4a–c), and reflected the patterns found in the main analysis (cf. Supplementary Fig. 3b). Further, in the low-motion trials, the pattern of the contrast between CS100 and CS50 at CS was also preserved (Supplementary Fig. 5d compared to Supplementary Fig. 3e). Together, the data support that these main effects are robust against the effects of motion.

As described below, we included CSF time courses in all SPM general linear models, to further account for physiological and nonphysiological noise. Lick events were also included in the model (see below), which served as an additional control for the effects of head motion, which is highly correlated with licking activity.

**Functional MRI analysis**. As mentioned, the HRF varies significantly between species, with faster kinetics in mice compared to humans[38,39,79]. We, therefore, used a mouse-specific HRF estimated by our group[38]. In all models, CS and US timepoints were modeled as events ("stick functions", duration 0 s).

Specifically, we computed three different general linear models (GLM; SPM12) at the session-wise analysis level:

GLM 1: First, to locate the brain regions encoding reward prediction, we modeled all CS timepoints (irrespective of trial type) as one event type, and parametrically modulated this with the $V(CS)$ estimated from the TD model. The four event types at the US timepoint (reward after CS100; reward after CS50, non-reward after CS50, non-reward after CS0) were each modeled with a separate event regressor. In this model as in all other models, lick events were also modeled as events (i.e., convolved with the HRF), to disentangle neuronal correlates of motor activity from correlates of reward expectation and prediction error per se. The bursts of licks, convolved with the HRF, translated into predicted hemodynamic responses in such a way that a higher number of licks within the burst lead to a higher amplitude of the predicted hemodynamic response. Therefore, this regressor might be reasonably expected to model the correlates of licking activity in a quantitative fashion, with the limitation that there is a high co-variance between licking activity and the effects of interest (neuronal correlates of reward expectation and consumption), such that we cannot expect a perfect statistical separation between these two processes (see also "Discussion"). In addition, time series averaged over cerebrospinal fluid voxels were added as a nuisance regressor.

GLM 2: Second, to disentangle more specific components of value coding, especially monotonic RP and recent outcome history, we created a separate GLM, where all CS event types (CS100, CS50, and CS0) and all US event types (reward after CS100; reward after CS50, non-reward after CS50, non-reward after CS0) were each modeled with a separate stick function regressor. The CS50 regressor was parametrically modulated by the recent CS50 outcome history (namely whether the last CS50 trial had been rewarded or not). Note that such a parametric modulation with recent outcome history would not be equally informative in GLM 1, since $V(CS)$ is already included in GLM 1 as a parametric modulator, and partially contains the variability of interest (outcome history) (cf. Fig. 2d). Analyses of monotonic RP and outcome history were restricted to those voxels that were significantly associated with $V(CS)$ at the group level, based on the GLM 1. Note that in this model (GLM 2), the same variability (namely the reward/non-reward outcome of a given CS50 trial) is modeled twice, firstly at the US timepoint of the trial itself, and secondly as a parametric modulator (outcome history) of the next CS50 trial. However, since the trial duration (jittered between 10 and 12 s) is long compared to the assumed duration of the hemodynamic response in mice[38,39], it is unlikely that an effect of the parametric modulator should be driven by a prolonged hemodynamic response to the preceding CS50 reward event itself. Also note that in a majority of cases, the previous CS50 trial is not the immediately preceding trial (since a given trial is preceded in two-third of cases by CS0 or CS100 trials). Further, note that the history modulation was also observed in neuronal recordings from the Tu.

GLM 3: Third, to test which brain regions encode the prediction error components ($V(CS)$, $r$) at the US timepoint, we computed a GLM where the three CS event types (CS100, CS50, CS0) were modeled with separate stick regressors, and all US event types were modeled by one single stick regressor, and the latter was parametrically modulated by two regressors, namely $-V(CS)$ and $r$. While $V(CS)$ was estimated by the TD model (identical to GLM 1), we used binary values for $r$ (1 and 0 representing reward and no reward, respectively). Of note, we did not orthogonalize one of the two parametric modulators with respect to the other; in this way, the order of entering the two parametric modulators did not influence the parameter estimation, and the effects of the variability shared between the two parametric modulators were removed. In other words, the effect of each parametric modulator was adjusted for the other[83].

Statistical maps representing the betas of the regressors of interest were then fed into group statistics. For this, we used the SPM12-based Sandwich Estimator toolbox (SwE). SwE is specifically designed for repeated measures of neuroimaging data and allows the number of sessions per subject to vary[84]. The equivalent of one-sample $t$ tests was used within the SwE framework to test for effects on the group level. Unless otherwise stated, the significance threshold was set to $P < 0.025$ false discovery rate (FDR)-corrected, for two-sided testing. Voxels surpassing the significance threshold were gray-matter masked.

To explore the magnitude of BOLD responses, percent signal change (PSC) was computed from the beta estimates using the ArtRepair toolbox for SPM (https://cibsr.stanford.edu/tools/human-brain-project/artrepair-software.html). In Supplementary Fig. 2e, this is shown for the following event types, whose beta estimates were computed in GLM 2, namely: CS100, CS50, CS0, reward after CS100, reward after CS50, and no reward after CS50. In order to account for regional variation in signal intensities with a surface receiver coil (lower signal intensity at the ventral portions of the brain), we modified the method by scaling the PSC not to the whole-brain signal intensity, but to the region-wise signal intensity (i.e., beta of the constant term averaged over the respective region, not over the whole brain). Compared to the original method, this leads to relatively larger PSC values for the ventral regions (Tu and aPC), which have both lower absolute signal intensities and lower absolute response magnitudes, compared to more dorsal regions (e.g., dorsal striatum).

**Task-related functional connectivity analysis**. The beta-series correlation analysis was based on previously described methods for seed- and ROI-based task-related functional connectivity in humans[85], including utilization of the BASCO toolbox[43]. In the context of event-related designs with many repetitions, as is the case in our design, it has been suggested that the beta-series method has more statistical power compared to the psychophysiological interaction (PPI) method[86]. Specifically, the same preprocessed and spatially normalized images as for the univariate fMRI analysis (see Fig. 3), were incorporated into another session-wise GLM, in which each event (CS and US) of each trial was modeled as a separate regressor to obtain the respective beta weights. For example, since there were 40 trial presentations of the CS100 trial type in each session, this resulted in the estimation of 40 corresponding beta weights at CS and 40 corresponding beta weights at US. For ROI-based analyses, betas were averaged over all voxels of a given region.

We then examined the event-related seed-ROI functional connectivity by computing, for each event type (e.g., US timepoint of CS100 trial type), the Pearson's correlation coefficient. This correlation was computed between the beta-series of either Tu or aPC and that of other regions in the odor-reward association-learning network. The group-level significance of the Fisher z-transformed (inverse hyperbolic tangent) coefficients was tested using a one-sample $t$ test ($P < 0.05$, Bonferroni-corrected for multiple comparisons, see Fig. 4b, c). The mean Fisher z-transformed results were subsequently converted into their hyperbolic tangent to derive the group-level correlation coefficients. The odor-reward association-learning network contained the following ROIs: olfactory bulb, anterior olfactory nucleus, aPC, as well as Tu, NAc, dorsal striatum, orbitofrontal cortex, mPFC and agranular insular cortex (see Fig. 4a). ROI parcellations were anatomically defined as per Paxinos and Watson (2009), with the exception of a refined parcellation for the aPC seed, which comprised the anterior portion of the piriform cortex delimited posteriorly at +0.8 mm relative to Bregma. The posterior piriform cortex was not included in the task-related connectivity analyses as its posteroventral parts were affected by signal dropout (see also section "Functional MRI acquisition"). To further investigate functional connectivity at below-ROI resolution, we performed an analogous analysis in a voxel-wise manner. Specifically, we correlated the mean beta-series of each seed region (Tu or aPC) with the beta-series of each voxel (2014 voxels total) contained within the odor-reward association-learning network. The beta-series correlation maps were derived for each session, Fisher z-transformed, and used for group-level analyses. At the group level, one-sample $t$ tests were computed to examine the correlated or anti-correlated seed network activity (see Fig. 4d, e and Supplementary Fig. 5a, b) and paired $t$ tests for the differential connectivity assigned to specific condition pairs (e.g., CS100 versus CS0) (see Supplementary Fig. 5c, d). For both analyses, the significance threshold was set to $P < 0.025$, FDR-corrected for multiple comparisons.

**Electrophysiological recording array**. We used an in-house designed tetrode array for dual-site recordings[28]. In brief, custom-designed printed circuit boards (PCB) (±10 µm, Würth Electronics) with a soldered Molex SlimStack connector served as an electrode interface board (EIB). The tetrodes, spun from 12.5-µm teflon-coated tungsten wire (California Fine Wire), were placed parallel to each other with the help of a guiding scaffold. The tetrodes were fixed to the PCB with a drop of liquid acrylic adhesive. For electrical contact, the single wires were soldered to the EIB after threading them through 200-µm vias and through holes. For protection, the single wires were coated with a two-component epoxy. The tetrode tips were gold-plated with a NanoZ-device (Multi Channel Systems), targeting an impedance of 300 kOhm. During recordings, the arrays were connected to the IntanRHD2164 head stages using a custom-built adapter (Molex SlimStack connector to two 36 Omnetics Nano Strip connectors). A custom-built titanium head bar was glued to the array.

**Implantation of recording array**. Eleven male C57BL/6 mice were implanted with two 16-tetrode arrays, each positioned in Tu and aPC. Pre- and post-surgery analgesia was administered. Mice were anesthetized with isoflurane and attached to a stereotactic apparatus with nonrupturing ear bars. A roughly circular flap of skin was removed from the skull and local anesthesia was administered. The lateral and nuchal muscle insertions were left intact. Tissue adhesive (3 M Vetbond) was applied to the margins of the remaining skin and holes drilled into the skull above the brain regions of interest and above the cerebellum for grounding. We then coated the skull using Super-Bond C&B (Sun Medical). An insulated copper wire attached to the recording array was connected to a small gold pin (Neuralynx) and placed into the cerebellum. The tetrodes were inserted into the target regions with a motorized 3-axis micromanipulator (Luigs&Neumann) (target coordinates relative to the center of the respective array for Tu: 1.6 mm anterior to bregma, 1.3 mm lateral from bregma and 4.9 mm ventral from dorsal brain surface; target coordinates for aPC: 1.4 mm anterior to bregma, 2.3 mm lateral from bregma, and 3.8 mm ventral from the medial dorsal brain surface). Once the target depth was reached, dental cement (Kulzer Palladur) was applied to fix the recording array in the final position. Animals recovered in their home cage and were monitored after surgery.

At the end of recording sessions, mice were euthanized and perfused with paraformaldehyde (4%). Due to low levels of scar formation and microglial activation with this tetrode array, we could not reliably detect fiber tracts in histological sections with Nissl staining. Therefore, the head was postfixed in PFA 4% for at least 2 weeks before sectioning. Rostrocaudal and mediolateral placement in the borders of Tu and aPC were confirmed histologically before sectioning (see Supplementary Fig. 6b). Subsequent removal of the tetrode array left visible scars in the tissue so that the recording location could be confirmed by post hoc histological examination (see Supplementary Fig. 6c).

**Electrophysiology head-fixed setup**. The same olfactometer and behavioral control system setup configuration as in the fMRI was used for the electrophysiological study. Single-unit recordings were conducted using two Intan 64 channel RHD 2164 miniature amplifier boards connected to the RHD2000 interface board. Software provided by Intan Technologies was used for data storage. Data were sampled at 30 kHz during neural recordings.

**Behavioral paradigm**. The same training procedure, task, performance criteria, and behavioral analyses were applied to the electrophysiology cohort as for the fMRI cohort.

**Poisson regression on the licking data**. To assess whether the anticipatory licking in the delay period of CS50 trials was influenced by (1) the cue-specific outcome of the previous CS50 trials (outcome history) and by (2) the outcome of previous CS100 and CS0 trials (satiety), we conducted two separate multiple Poisson regressions on the licking data (see Fig. 9e, f). The Poisson model was defined by the equation: $\log(\mu_t) = \beta \cdot X$ where $\mu_t$ is the expected value of the anticipatory licking in the current CS50 trial, $X$ is the regressor matrix and the $\beta$ are the regression coefficients. The columns of the matrix $X$ are the regressor vectors. The elements of the regressor matrix assumed values $x_{ij} \epsilon \{1, -1\}$, where 1 codes for a rewarded trial and $-1$ for an unrewarded trial. In case (1), the regressor vectors $X$ were built as described, using the n-back CS50 → R and CS50 → N trials ($n = 1,2,...,N$ with $N = 6$). In case (2), the regressor vectors $X$ were built as described, using the n-back CS100 and CS0 trials ($n = 1,2,...,N$ with $N = 6$).

To better compare the anticipatory licking from different sessions, the coefficients of each regressor $n$ were standardized as follows $\beta_n^* = \beta_n \cdot \frac{\sigma(x_{*,n})}{\sigma(\mu_t)}$ and, for better interpretability, transformed as $\beta_n^* \rightarrow \exp(\beta_n^*) - 1$. Thus, a positive $\beta_n^*$ indicates a positive correlation between the anticipatory licks and the $n$-th regressor when the other regressors are constant. Vice-versa, a negative $\beta_n^*$ indicates a negative correlation. We performed one regression per session and plotted the average standardized coefficients ($\bar{\beta}^*$) for each animal ($n = 88$ sessions in 11 animals).

**Data preprocessing: spike detection**. Noise and movement artifacts affecting all recorded channels were reduced by conducting a median subtraction. Therefore, we calculated the median voltage trace of all channels from the same recording site and subtracted the median from each recorded channel. The resulting signal was passed via a band-pass filter (300–5000 Hz, 4th-order Butterworth filter, built-in MATLAB function). All local maxima crossing a certain amplitude threshold (7.5× of the median absolute deviation of the filtered signal) were identified as spiking events. To prevent a multiphasic spike from being detected multiple times, the minimum distance between threshold crossing peaks was set to 1 ms. If a spiking event was detected on more than one channel of the same tetrode, it was assigned to the timestamp of the highest detected peak. For each cluster, waveforms were obtained by extracting −10 to +21 sampling points around the peak.

**Data preprocessing: spike sorting**. We clustered the detected spiking events using a custom-built graphical user interface in MATLAB developed by A.

Koulakov (CSHL). Single units were separated based on different metrics including peak height or amplitude of the spikes and the respective principal components over channels. When spiking events were predominantly recorded on one channel, the first three principal components of the waveforms were considered. Single-unit quality was quantified using the mlib toolbox by Maik Stüttgen (Version 6, https://de.mathworks.com/matlabcentral/fileexchange/37339-mlib-toolbox-for-analyzing-spike-data). In particular, we estimated cluster quality through refractory period violations (fraction of spikes during the refractory period <2 ms) and waveform variance. Only if clusters had a ratio of refractory period violations to the total number of spikes of less than 2%, they were considered as single units. In Tu, units with less than 5 Hz baseline firing rate were classified as putative striatal projection neurons and considered for subsequent analysis. Single units with a firing rate >5 Hz were excluded as fast-spiking neurons from the Tu or from the neighboring anterior ventral pallidum. A total number of 169 putative striatal projection neurons were included in the analysis. In aPC, a total of 486 putative principal units with less than 10 Hz baseline firing rate were included in the analysis.

**Analysis of single-unit responses**. We classified single units according to their task-related spiking activity. We calculated averaged spike counts for all trial types for the baseline, CS, waiting and US windows (baseline: from −1.5 to −0.5 s; CS: from 0 to 1 s; waiting: from 1 to 2.5 s; US: from 2.7 to 3.7 s relative to odor onset). Single units were considered as responsive during CS, waiting or US if they showed a significant difference compared to baseline (Friedman test, $P < 0.05$ with Benjamini correction for multiple comparisons on all units tested for each trial type and each task window). Single units displaying a significant increase in spiking activity were defined as "task-excited" while units displaying a decrease were defined as "task-inhibited" (see Figs. 6d–f and 7d–i and Supplementary Figs. 7d, e and 8b, c).

**Population analysis: the population vector**. The session-specific population vector $\boldsymbol{v}_t^s = [fr_t^{s,1}, \dots, fr_t^{s,n}]$ is a vector composed of the firing rate of $n$ simultaneously recorded neurons during a time-bin centered at time $t = 1...T$, with $T$ total number of time steps per trial. Given the relatively small cell yield per session, session-specific population vectors were concatenated to compose a global population vector $\boldsymbol{V}_t = [\boldsymbol{v}_t^1, \dots, \boldsymbol{v}_t^S]$. Trials of different sessions were concatenated by matching CS, trial order, and, for CS50, trial outcome. Since, in CS50 trials, reward was delivered with a 0.5 probability, not all sessions had the same number of rewarded CS50 trials. Population vectors for CS50 were, thus, built by selecting the minimum number of trials available among sessions and, in sessions with supernumerary trials, by omitting the last trials.

**Population analysis: deviation from baseline**. To investigate the temporal evolution of aPC and Tu responses to CS and US, we computed the deviation of the population vector from its baseline configuration. At each time step $t$, deviation from baseline was computed as Euclidean distance between the population vector $V_t$ and the baseline vector $B$. The $B$ vector was composed of the firing rate of each unit in the window −2 to −1.25 s before odor onset averaged across all trials. To reduce trial-to-trial variability, such analysis was performed by using the population vectors $\bar{V}_t$, built by grouping population vectors $V_t$ of consecutive trials in pairs and averaging. This procedure reduces the number of samples available but produces more stable population responses. For visualization, population vectors were computed on bins of 500 ms, moved in time with steps of 125 ms. Statistical tests of changes in population response to different trial types were conducted on distances computed with non-overlapping bins of 250 ms.

**Population analysis: cross-trial correlation**. To assess if the different trial types recruit overlapping sets of units and to quantify the degree of such overlap throughout the trial progression, we computed the Pearson cross-correlation between the population vectors of trials with different trial types. Trials were first grouped according to their CS trial type. Then, for each time step $t$, each vector $V_t$ was cross-correlated, at turn, with the $V_t$ vectors of all trials from other CS trial types. For example, to quantify the degree of overlap between the set of units supporting the encoding of CS100 and CS0 during the trial we computed, at every time step $t$, the average Pearson cross-correlation between $V_t$ of all CS100 trials with $V_t$ of all CS0 trials. An increase (decrease) in cross-correlation will imply an increase (decrease) in the size of the subpopulation commonly recruited by the two stimuli at a certain time $t$. As above, such analysis was performed by using population vectors $\bar{V}_t$, built by grouping population vectors $V_t$ of consecutive trials into groups of two and averaging. Population vectors were computed on bins of 500 ms, moved in time with steps of 125 ms.

**Population analysis: population trajectories**. To visualize the differences between the temporal evolution of the population vectors in response to different CS and trial types, we averaged the population vectors $V_t$ across all trials to obtain, for each trial type, a matrix of size N × T. To improve the reconstruction of the neuronal dynamics[87], we applied time embedding on the multivariate time series obtained. We used $m = 4$ delayed coordinates with a delay constant of one bin. Finally, for visualization, we reduced with PCA the space dimensionality to 3.

**Identification of functional unit clusters**. To identify functional unit clusters within aPC and Tu populations we used a clustering approach adapted from Cohen et al.[14]. Functional clusters were established according to the similarity in unit responses to different trial types. We first quantified the deviation at time $t$ of the response of each unit to its baseline distribution by computing the area under the receiver operating characteristic curve (auROC). The baseline distribution was computed by pooling the firing rate of the unit in the window from $-1.8$ to $-1.4$ s before odor onset. The distribution of the unit response at time $t$ was computed similarly, but pooling from a window of 500 ms shifted along the trial in steps of 125 ms. auROC values were computed for each unit and each condition—that is CS0, unrewarded CS50, rewarded CS50, CS100—and concatenated as shown in Fig. 6g and Fig. 7j. The values 0, 0.5 and 1 indicate lower, equal, and higher firing rates than during baseline, respectively. We then performed PCA on the concatenated profiles and applied hierarchical clustering. Hierarchical clustering was performed on the Euclidian distance between vectors comprising of the first five PCs of each unit. For aPC and Tu, we used a cutoff in the linkage tree of 0.45 and 0.5, respectively. Such cutoffs were chosen to balance between generality and representativeness of the clusters with respect to their composing units.

**Complete and distributed coding**. To assess if aPC and Tu encoded a monotonic RP in a homogeneous or distributed fashion (Fig. 8 and Supplementary Fig. 9), we tested each unit for monotonic RP coding and single CS dominance. To test monotonic RP coding during CS and the waiting period, we computed the unit firing rate in the CS window (0 to 1 s after odor onset) and waiting period (1 to 2.5 s after odor onset), respectively. Unit responses were computed for each trial $tr$ and grouped by CS into $R0_{tr}$, $R50_{tr}$, $R100_{tr}$. Units from the transient-, ramping-, and sustained clusters were labeled as monotonic RP coding if $\{R0_{tr}\}<\{R50_{tr}\}$ and $\{R50_{tr}\}<\{R100_{tr}\}$ when tested with a two-tailed Wilcoxon rank-sum test ($P$ value <0.05). If the test for monotonic RP coding failed, units were further tested for CS dominance. CS dominance required that responses to the CS with stronger unit response were significantly higher than those to the CS with the second-strongest response (two-tailed Wilcoxon rank-sum test, $P$ < 0.05). For units from the inhibited clusters, monotonic RP was established by testing $\{R0_{tr}\}>\{R50_{tr}\}$ and $\{R50_{tr}\}>\{R100_{tr}\}$ while CS dominance required that responses to the CS with the strongest task-inhibited response dropped to significantly lower response rates than those to the CS with the second-lowest response.

**Reward surprise**. To test if a unit encoded surprise in receiving an unexpected reward (reward surprise) we tested if the rate jump $d_{tr} = fr_{tr}^{after} - fr_{tr}^{prior}$ from prior to after US onset was bigger for rewarded CS50 trials than for CS100 trials. To encode surprise we enforced two prerequisites: (a) the average rate jump for CS50 rewarded trials had to be positive, $\langle d_{tr}^{50}\rangle_{trials}>0$; (b) the rate jump for CS50 rewarded trials had to be bigger than that for CS100 in a two-tailed Wilcoxon rank-sum test (significance fixed at $P$ value < 0.05), $\{d_{tr}^{50}\}>\{d_{tr}^{100}\}$. From visual inspection of the activation profile of Tu and aPC units, two characteristic US response profiles emerged, with different latency from US and different duration. Clustering units in transient- or ramping populations did not segregate the two response profiles. To capture both response types, we computed reward surprise in two distinct trial windows. To capture short-latency short-duration responses, $fr_{tr}^{after}$ was computed in the window from 2.7 to 3.2 s after odor onset. To capture longer latency responses with longer duration, $fr_{tr}^{after}$ was computed in the window from 3.2 to 4.5 s after odor onset. In both cases, $fr_{tr}^{prior}$ was the firing rate of the unit at trial $tr$ in the 500 ms window prior US delivery. All units were tested in both windows and were flagged as encoding reward surprise if significant in either window. The criteria listed here were applied to all units from the transient-, ramping- and sustained clusters. For units from the inhibited cluster, reward surprise required: (a) $\langle d_{tr}^{50}\rangle_{trials}<0$; (b) $\{d_{tr}^{50}\}<\{d_{tr}^{100}\}$.

**Outcome discrimination**. To assess if a unit encoded outcome discrimination, we used a two-tailed Wilcoxon rank-sum test and tested if $\{fr_{tr}^{after}\}$ for CS50 rewarded trials was bigger than $\{fr_{tr}^{after}\}$ for CS50 unrewarded trials. We did not consider the extremely rare opposite case, thereby focusing on the mechanisms contributing to the computation of the prediction error. Similar to the test for reward surprise, we tested both short- and long-latency responses and tested units from the task-inhibited cluster using the opposite inequity sign (see "Reward surprise").

**Chance level for monotonic RP coding**. We selected all $\#_{discriminative\ units}$ units which showed a different response intensity to the three CS. This was obtained by requiring a $P$ value < 0.05 when testing the responses of each unit to different pairs of odors with a two-tailed Wilcoxon rank-sum test. Since the number of permutations without repetition of $n = 3$ elements is $n! = 6$, the number of units with a specific order in odor response (e.g., CS100 > CS50 > CS0) expected by chance is $\#_{discriminative\ units}/6$.

**Data analysis**. Behavioral and electrophysiological data were analyzed using built-in and custom-made MATLAB routines (Mathworks) and SPSS (IBM). fMRI sessions were analyzed using SPM12 and the SPM12-based Sandwich Estimator toolbox (SwE) at the group level.

**Reporting summary**. Further information on research design is available in the Nature Research Reporting Summary linked to this article.

## Data availability
The fMRI and electrophysiology data generated in this study are under active use by the reporting laboratory; all data presented in this manuscript are available upon reasonable request (for MRI data: christian.clemm@zi-mannheim.de, for electrophysiology data: wokelsch@uni-mainz.de). BOLD fMRI statistical maps and electrophysiological single-unit spike counts are available for download at https://doi.org/10.6084/m9.figshare.c.6000757.v1. Source data are provided with this paper.

## Code availability
Matlab code and pipeline description for pupil data analyses and MRI brain mask creation are available at https://github.com/DrCarbonCIMH/extractPupil and https://github.com/DrCarbonCIMH/extractBrain, respectively.

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

## Acknowledgements

The authors thank Dr. Peter Dayan for discussions and comments on the manuscript and Drs. Urs Braun and Heike Tost for initial discussions. We thank Ariana Frömmig, Felix Hörner and Catrin Loeb for excellent technical assistance. This work is supported by the BMBF grant n. 01GQ1708 to W.K., Boehringer Ingelheim Foundation grant "Complex Systems" to W.K. and E.R., DFG Priority Program SPP1665 KE1661/2-2 to W.K., the Ch. and H. Schaller Foundation to E.R., and the Focus Program Translational Neurosciences to R.H., the BMBF (01EW2010B) in the frame of ERA-Net NEURON to A.S. and J.R., and the Clinician Scientist Program 'Interfaces and Interventions in Complex Chronic Conditions' by the German Research Foundation (DFG) (EB187/8-1) to J.R.

## Author contributions

C.C.v.H., W.K., A.M.L., and L.W. conceived the project. M.Sch. designed the recording array and behavioral setups. L.W. performed behavior, fMRI, and neurophysiology. C.F. designed and implemented the behavioral model. M.Sa., R.B., and L.W. developed the pupil analyses script. W.K., E.R., and L.W. designed and implemented the analysis for neurophysiology. D.W. helped with neurophysiology. M.F.G., C.C.v.H., R.H., and L.W. designed and implemented the fMRI analyses. A.S., J.R., and W.W.F. provided scripts, resources, and helped to implement the fMRI analyses. C.C.v.H., W.K., E.R., and L.W. wrote the manuscript.

## Funding

## Competing interests

The authors declare no competing interests.
