## [Peer Review File · Nature Communications]

Striatal hub of dynamic and stabilized prediction coding in forebrain networks for olfactory reinforcement learningREVIEWER COMMENTS

Reviewer #1 (Remarks to the Author):

This study uses an impressive range of techniques, including fMRI, computational modelling, and single unit electrophysiology to investigate the circuits responsible for an olfactory driven reinforcement learning task in mice. The strength of this study is the combination of the low resolution “mesoscale” and high-resolution local recording techniques. The study identifies the olfactory tubercle as a key hub where multiple aspects of reward prediction and prediction error coding are represented. In contrast areas upstream of the tubercle are shown to represent a subset of value components. In addition, the single unit electrophysiology identifies two population of neurons in the tubercle that are differentially influenced by recent reward history. Value predictions in the ramping neurons are rapidly updated by changes in predicted value, while transient neurons represent stable value predictions that are insensitive to recent changes in reward history. Together the study is well written, and the conclusions are well supported by the data. I have no major concerns just a few minor points that should not distract from the tremendous amount of work already done in this study.

1. I found the use of abbreviations distracting. Could the authors just write out the terms instead? Even if the abbreviations are used, I think refereeing to the prediction errors as reward prediction errors is more appropriate. There are lots of types of prediction errors, sensory, unsigned, novelty etc.. given that they specify that the predictions are based on reward then reward prediction error is what is most likely being observed so RPE should be used if they want to use an abbreviation.
2. I am not an expert on olfactory coding, but it was surprising to me that scaled reward predictions were already observed in the olfactory bulb. Is this already known? Could these results be discussed?

Reviewer #3 (Remarks to the Author):

Winkelmeier et al. investigated which brain regions encode reward prediction and prediction error signals during an olfactory learning task with different reward probabilities. Using task-based fMRI and a behavioral model, the authors identified the forebrain neural network that computes monotonic reward prediction and prediction error signals, including specific prefrontal, striatal and olfactory regions. They further characterized the functional connectivity of these regions using the olfactory tubercle and anterior piriform cortex as seeds. Moreover, the authors compared task-related neuronal firing in the olfactory tubercle and piriform cortex, and found that the olfactory tubercle neurons differentially encode reward prediction, prediction error, and its dynamic updating. Although the main findings are not entirely novel, the current study does expand on the literature of olfactory reward network via a cross-scale approach combining task-based fMRI and simultaneous single-unit recordings from multiple brain regions. The data collection and analysis are impressive. The manuscript is clearly written, and the conclusions are statistically justified. The manuscript can be further improved by addressing the following minor points.

1. In Supplemental Figure 1a, it is surprising that the correct trials for CS50 also reach ~90% given that only 50% of the trials are rewarded and the prediction error remains high as shown in Fig. 1g. An explanation would be helpful. If mice do not reach this criterion are excluded, the authors should provide this information.
2. Three odors (geranium, ylang-ylang, and rose) are used as CS. It seems that for a given mouse, the odors are fixed as CS100, CS50, or CS0. It is not clear whether they are switched around for different mice (for example, geranium as CS100 for some mice but as CS50 or CS0 for other mice). This needs to be clarified in the Results.
3. Related to the above point, do single units in the OT or aPC respond differently to these odors before training? If these odors are not switched around for different mice, the intrinsic differences in response patterns to these odors may contribute to the different CS responses after training. For example, in Fig. 4g, h, the responses of aPC units to CS0 is significantly higher than CS50. It would be helpful to clarify.
4. In Figure 5d-f, it is not very clear what 1, 2, 3 and the colors stand for. Please clarify.
5. Similarly in Figure 6d-f, please clarify what 1, 2, 3, and the colors stand for. What does the SOP in the figure legend mean?
6. Figure 7b, what the numbers #961, 113, and 1501 mean? If they are example
7. Line 513, in the sentence “We finally examined how the multiple circuits encoding RP in the limbic system...”. In fact, the olfactory tubercle and anterior piriform cortex are examined here. I would state “in the Tu and aPC” rather than “the limbic system”.
8. Supplemental Figure 7c, the percentage of “transient” type should be $70/486=14\%$ instead of 60%.
9. The authors may want to justify why only male mice are used in this study.

Reviewer #4 (Remarks to the Author):

In their work “Striatal hub of dynamic and stabilized prediction coding in forebrain networks for olfactory reinforcement learning”, the authors aim to study forebrain regions involved in olfactory reinforcement learning by combining modeling of expected behavior, functional neuroimaging, and electrophysiological recordings. The imaging results indicate that specific olfactory, prefrontal, and striatal regions were involved in processing of odor-related sensory processing and (updated) reward prediction. Importantly, regions appeared to have distinct roles in the behavioral processing. These observations were supported by the electrophysiological recordings that further indicated the different coding roles of olfactory cortex and tubercle in odor processing and reward prediction. I found the experimental setup and findings interesting. The manuscript has relevant aim, is well-designed and mostly well-written, and includes a good amount of technically demanding experiments. My comments / concerns are the following:

Major ones:

1. Movement is a serious concern in fMRI, as also briefly stated by the authors. Echo planar imaging is particularly sensitive to any movement-induced B0 field distortions, as even chest movement can stretch and/or shift the voxels inside the brain in the phase-encoding direction. Such distortions are difficult to correct without advanced techniques, such as continuous field

mapping. My concern is that the ICA components in Supp. Fig 2C (particularly C2 but also C4) were categorized as movement artefacts by the authors, and these components overlap exactly the main regions of interest, namely piriform cortex and olfactory tubercle. These suggest that jaw movement and licking affected these regions considerably, and it remains unclear whether the movement-related artefacts still have significant impact on results. Thus, the differences between CS100/50 and CS0 may originate partially or even substantially from the amount of jaw/body movement. I think these limitations should be stated or discussed in more detail.

2. As the authors already state, fMRI suffers from poor temporal resolution and signal that is confounded by hemodynamic response function. As the HRF can have a decay of several seconds, it might be difficult to conclude whether the brain region was activated during the wait period or US if it was activated during CS (as there was only 1.7 s gap between). I think this should be discussed or mentioned as a limitation related to the fMRI findings.

3. The licking rate during US was 4-7 Hz after CS100 / CS50, which is presumably in line with the “correct” behavior. If I understood correctly, authors aimed to regress the motor activity out in GLM by modeling a licking event as an impulse and convolving it with HRF. However, such high frequency of impulses transforms into constant activity in hemodynamic response. Therefore, it might be difficult to separate somatomotor activation from olfactory and cognitive processing as they typically occurred simultaneously, and one would expect to see a robust and well-localized activity in motor and somatosensory pathways. Furthermore, somatosensory and motor cortices are mentioned in some of the figure legends (e.g., Figure 2), but they are not present in images or covered in the manuscript text. Can authors discuss the expected / missing presence of somatosensory and motor activity particularly in Figure 2F-H?

4. As stated by the authors, there were image distortions near the posterior piriform cortex and olfactory tubercle, which were key regions of interest. Can authors comment the possibility / likelihood whether because of, e.g., signal pile-up artefacts, the activation in these regions could have been located falsely at, e.g., the striatal regions, perhaps compromising some of the conclusions? Raw fMRI images including the regions of interest as a supplement figure could clarify the severity of this issue.

5. Behavior is known to be modified by stress. Discussion related to the effects of body restraint- and noise-induced stress on learning and related processing, and subsequently on the findings and their generalization would be required.

6. At general level, the description of methods related to data analysis would greatly benefit of revisioning. It is possible to get a rough idea by multiple readings and reasoning, but the flow of the text could be much better. I could not find a clear definition for “s” or “st” (lines 792-817). Therefore, it was difficult to determine the values in V. The definitions should be clear and the order of equations and explanations easy for reader. It was unclear what $V(0)$ means? When is $s = 0$? Also, the definition for initial V comes very late for readability.

7. There are some uncertainties related to the GLM analyses at lines 938-951. First, can authors elaborate the rationale for modelling monotonic RP (GLM #2) in addition to the V-modulated CS (GLM #1)? Apparently, these model the same outcome, but doesn't monotonic RP have less signal and is less suitable? The only clearly different part between GLM #1 and

GLM #2 is the outcome history in the CS50-US. As the previous CS50 outcome is encoded as a modulation in the present CS50 event, how does the model in GLM #2 separate the present CS50 response and the response to the previous round CS50 from each other? Would it be possible to model the history equally well in GLM #1 as an additional binary (or 3-valued pair (if you want to exclude CS100 and CS0) of) regressor(s)? A carefully designed regressor might be even more appropriate if it was specifically designed to correspond only to the event history contrast. Second, the GLM #3 models for complement of $V(\text{CS})$ and r . It appears that a single GLM was done, so it remains unclear how the US events with both $-V(\text{CS})$ and r were modulated. What weights were used and how were they chosen? If the modulation worked, I see no problems with the individual $-V$ and r maps. However, considering the map intersection as representing the common properties of both in the same spatial location is questionable. Isn't it possible that the common regions are highlighted for completely different reasons but are accidentally spatially located in the same spot? Modelling the interaction between $-V$ and r explicitly in the GLM would make the result more convincing.

Minor ones:

8. As stated at Line 49 and in the abstract, fMRI allows rather unbiased identification of brain regions involved in behavioral processing. However, as authors imaged only forebrain, I would not emphasize the unbiased identification aspect in the current study.

9. The exact setup of odor delivery system remains unclear. As the tubing to the MRI bore is quite long, there are many important factors to be clarified. I did not find the exact location of odor supplies and photo-ionization detector, and how long was the distance from these to the mice. Were different odors delivered with dedicated tubes, or was the same tube used for all odors? Was there a risk that cross-contamination would confound the experiments? If odor ports and detector were outside the bore, what was the delivery time to the mouse and how was it measured to obtain exact timing with the fMRI? If odor ports and detector were inside the MRI bore, of which material were they made of?

10. The stimuli were short and expected BOLD responses are thus very small, likely less than 1%. I'm sure it would be highly interesting for readers to see examples of BOLD time series and amplitudes, perhaps together with the general linear models.

11. The term functional connectivity may be misleading in this context, as it typically refers to correlating low-frequency fluctuations during long time periods. Perhaps "co-activation" would be more justified? Moreover, brain regions may be activated at the same time but by different inputs that may not be linked closely to each other, questioning the functional connectivity.

12. In Figure 3B-C, it remains unclear why there are empty slots in the matrices. Is it because the correlations in those slots did not exceed statistical significance?

13. At line 687, it would be better to also mention the active compound of the drug (Metacam). Commercial products or their names may disappear from the market.

14. There are no information about the post-operative care after the head-bar implantation, which I assume would be needed according to the regulations.

15. There are no information whether anesthesia was used during the habituation or MRI preparations. Does this mean that the animals were fixed to the holder while they were awake? This should be clarified.

16. The inter-trial interval would be worth of mentioning in the Figure 1 legend, as I had difficulties to find it. Moreover, was the same 10-12 s used in all experiments?

17. The type of motion correction in fMRI data should be stated, e.g. affine or rigid.

18. In addition to average realignment parameters, the maximum values of the realignment parameters should be reported. This would indicate the robustness of the head-fixation, and the magnitude of the movement, which both function as indicators of the data quality. The importance of this aspect is emphasized in the current work as the authors did not perform motion scrubbing.

19. Value for learning rate ($\alpha = 0.28$) could be mentioned in the text (results or methods), as I found it only in figure legend.

20. At lines 793-795, the authors write "This reward prediction (RP) signal is adjusted in time through the difference between the predicted and the actual outcome namely the reward prediction error (PE) at US." What does "adjusted in time" exactly mean?

21. At lines 879-881, the authors state "...spatial normalization to a mouse brain template in the Paxinos stereotactic coordinate system, by applying the non-linear normalization parameters of the structural images to the functional images". However, I did not notice that any processing was reported for structural images that could be applied on the functional images.

22. As a minor note, can authors comment whether four equal-sized groups (instead of three) would have been more suitable, regarding e.g. statistical power?

Reply to Reviewer #1 (Remarks to the Author):

This study uses an impressive range of techniques, including fMRI, computational modeling, and single unit electrophysiology to investigate the circuits responsible for an olfactory driven reinforcement learning task in mice. The strength of this study is the combination of the low resolution “mesoscale” and high-resolution local recording techniques. The study identifies the olfactory tubercle as a key hub where multiple aspects of reward prediction and prediction error coding are represented. In contrast areas upstream of the tubercle are shown to represent a subset of value components. In addition, the single unit electrophysiology identifies two population of neurons in the tubercle that are differential influenced by recent reward history. Value predictions in the ramping neurons are rapidly updated by changes in predicted value, while transient neurons represent stable value predictions that are insensitive to recent changes in reward history. Together the study is well written, and the conclusions are well supported by the data. I have no major concerns just a few minor points that should not distract from the tremendous amount of work already done in this study.

We thank the Reviewer for the feedback and helpful comments. We addressed them accordingly.

The line numbers indicated in this reply refer to the numbers of the attached manuscript .pdf with highlighted changes in red for reviewers.

1) I found the use of abbreviations distracting. Could the authors just write out the terms instead? Even if the abbreviations are used, I think refereeing to the prediction errors as reward prediction errors is more appropriate. There are lots of types of prediction errors, sensory, unsigned, novelty etc. given that they specify that the predictions are based on reward then reward prediction error is what is most likely being observed so RPE should be used if they want to use an abbreviation.

We thank the Reviewer for these suggestions. We reduced the number of abbreviations in the manuscript to improve the flow of reading. The abbreviation PE is now replaced by ‘prediction error’. We avoided using the abbreviation RPE for the prediction error at US, as the term RPE is frequently used to describe the whole phenomenon of prediction error at US and predicted value at CS. We also removed several less frequently used abbreviations in particular of anatomical regions (agranular insular cortex, anterior olfactory nucleus, dorsal striatum, olfactory bulb, orbitofrontal cortex, ventral tegmental area). For readability, these changes were not highlighted in the manuscript.

2) I am not an expert on olfactory coding, but it was surprising to me that scaled reward predictions were already observed in the olfactory bulb. Is this already known? Could these results be discussed?

We thank the Reviewer for this important question and discuss this point now in more detail in l. 425 of the Main text: “Learning to predict outcome has been shown to modify neuronal representations in the olfactory bulb, possibly through top-down mechanisms (Doucette et al., 2011; Lepousez et al., 2014; Jordan et al., 2018).”

Reply to Reviewer #3 (Remarks to the Author):

Winkelmeier et al. investigated which brain regions encode reward prediction and prediction error signals during an olfactory learning task with different reward probabilities. Using task-based fMRI and a behavioral model, the authors identified the forebrain neural network that computes monotonic reward prediction and prediction error signals, including specific prefrontal, striatal and olfactory regions. They further characterized the functional connectivity of these regions using the olfactory tubercle and anterior piriform cortex as seeds. Moreover, the authors compared task-related neuronal firing in the olfactory tubercle and piriform cortex, and found that the olfactory tubercle neurons differentially encode reward prediction, prediction error, and its dynamic updating. Although the main findings are not entirely novel, the current study does expand on the literature of olfactory reward network via a cross-scale approach combining task-based fMRI and simultaneous single-unit recordings from multiple brain regions. The data collection and analysis are impressive. The manuscript is clearly written, and the conclusions are statistically justified. The manuscript can be further improved by addressing the following minor points.

We thank the Reviewer for their careful consideration of our manuscript and helpful comments.

The line numbers indicated in this reply refer to the numbers of the attached manuscript .pdf with highlighted changes in red for reviewers.

1) In Supplemental Figure 1a, it is surprising that the correct trials for CS50 also reach ~90% given that only 50% of the trials are rewarded and the prediction error remains high as shown in Fig. 1g. An explanation would be helpful. If mice do not reach this criterion are excluded, the authors should provide this information.

The performance curves in Supplementary Fig. 1a display the percentage of trials meeting the performance criterion computed for each of the three CS types. The data comprises 69 training sessions in 23 animals (the last three training sessions per animal). We added this information to the legend of Figure 1b: “**Evolution of average lick rates \pm SEM differentiated trial types (n = 69 training sessions in 23 animals, 3 sessions per animal).**” We did not exclude any session for these analyses, which thus include also sessions where the animals did not reach the performance criterion.

The performance criterion parametrizes the animals’ ability to differentiate between ‘Go’ (potentially rewarded trials, i.e, all CS100 and CS50) and ‘No-go’ (all CS0) trials within the behavioral paradigm. Thus, both eventually rewarded and unrewarded CS50 trials were considered equally as ‘Go’ trials (being potentially rewarded). Animals tried to retrieve the potential reward by licking during delay (anticipation) or US (retrieval attempt) in about 90% of CS50 trials. We further clarified our performance criterion in the legend of Supplementary Fig. 1a: “**Performance curves for the three CS types in the mouse cohort shown in Fig. 1. The data contain the last three training sessions before fMRI (total n=69 session from 23 mice). Performance curves discriminate learning of no-go (all CS0) and go trials (all CS50 and CS100). The performance criterion was met for no-go trials if animals licked less than 3 times in total in the anticipatory and reward window, or at least three times for the go trials (see Methods for detailed information). One trial block had 5 trials of each type. Mice performed consistently above criterion. Solid lines indicate average performance while dashed lines indicate median performance. Box plots showing medians with solid bars indicating the 25th and 75th percentiles.**”

2) Three odors (geranium, ylang-ylang, and rose) are used as CS. It seems that for a given mouse, the odors are fixed as CS100, CS50, or CS0. It is not clear whether they are switched around for different mice (for example, geranium as CS100 for some mice but as CS50 or CS0 for other mice). This needs to be clarified in the Results.

We thank the Reviewer for addressing this important specification. We have now better highlighted in the Results which pairs of odors and reward probabilities have been used; these pairs remained unchanged across sessions (starting l. 74): “**Mice were conditioned to odor stimuli that signaled different probabilities of upcoming reward (100% - geranium, 50% - ylang-ylang, or 0% - rose; hereafter, these trial types are labeled CS100, CS50, or CS0, respectively) (Fig. 1a). Across sessions, odor and reward contingency pairs were kept unchanged.**”

3) Related to the above point, do single units in the OT or aPC respond differently to these odors before training? If these odors are not switched around for different mice, the intrinsic differences in response patterns to these odors may contribute to the different CS responses after training. For example, in Fig. 4g, h, the responses of aPC units to CS0 is significantly higher than CS50. It would be helpful to clarify.

We agree with the Reviewer that an initial difference in response to the odorants might influence the difference in CS responses observed after training. This would occur if the learning of the distinct reward probabilities associated with CS induces little or no change in the stimulus responses (to reflect the predicted reward value). This is indeed the case for the responses to CS in aPC (and more prominently for the initial excitatory component). In the first training session, the mean aPC response to CS50 is significantly smaller than that to CS0 and CS100 (one way ANOVA computed on aPC at CS in the first third of the session: $F(2,45)=20.2$, $p=5.4 \times 10^{-7}$, post hoc: (CS0,CS50) $p=2.8 \times 10^{-6}$, (CS0,CS100) $p=9.9 \times 10^{-1}$, (CS100,CS50) $p=4.8 \times 10^{-6}$) and the same difference in odor responses is found after training (Fig. 5b). aPC change in odor responses at CS due to conditioning, if at all present, is therefore not strong enough to overwrite intrinsic differences in odor responses. On the other hand, aPC activity during the delay period (emerging task-inhibited delay value coding) and Tu activity during CS and delay period (emerging task-excited CS and delay value coding) significantly changed with learning (Fig. 5b-c). In these latter cases, while during the first session of training response intensities were ordered non-monotonically according to odor value (one way ANOVA on the first third of the session: Tu at CS, $F(2,45)=1.0$, $p=3.7 \times 10^{-1}$; Tu during delay, $F(2,45)=6.3$, $p=4.0 \times 10^{-3}$, post hoc: (CS0,CS50) $p=1.0$, (CS0,CS100) $p=1.2 \times 10^{-2}$, (CS100,CS50) $p=9.9 \times 10^{-3}$; aPC during delay, $F(2,45)=23.1$, $p=1.3 \times 10^{-7}$, post hoc: (CS0,CS50) $p=1.8 \times 10^{-3}$, (CS0,CS100) $p=5.5 \times 10^{-3}$, (CS100,CS50) $p=6.0 \times 10^{-8}$, statistical analyses reported now in the figure legend of Supplementary Fig. 6,7 and in Table 1), after learning their relative order became monotonic (Fig. 5b, 6b). These findings thereby highlight two aspects: first, that (consistently across animals) intrinsic properties of odors are encoded more prominently in primary olfactory cortices compared to the ventral striatal region; and second, that during learning prominent changes occur in the Tu and also in the task-inhibited delayed responses of the aPC reflecting the prediction of future reward.

In addition to the original description that odor contingencies were not changed (method section in previous ms. l. 739, now l. 791), we now highlight these points for clarity in the initial explanation of the rationale for examining monotonic value representations with fMRI (l. 137): “If reward prediction is the dominant feature of a region, response intensities should reshape and follow monotonically the predicted reward probability of the CS after learning.” We also added to the main results (starting l. 254): “Throughout training, all animals had been presented with the same odor identity-reward probability associations. At the beginning of the first session, the odors elicited different mean responses not reflecting reward probabilities, but most likely intrinsic odor properties (Supplementary Fig. 6c and Supplementary Table 1 for statistical test).” And in the discussion at l. 429: “..., while in Tu the CS-excited mean rate responses developed monotonic value coding, such re-arrangement was less prominent in the aPC. Rather, aPC appeared to partially preserve the influence of initial odorant features expressed before learning, ...” And in the legend of Supplementary Fig. 6c: “Note that in the first trials of the first training session, mean aPC responses to the three odors partially differed at CS and during wait (see Supplementary Table 1 for exact p-values and test details), possibly reflecting intrinsic properties of the odors.” and Supplementary Fig. 7a: “Note that while aPC encodes at CS the odors with responses of different intensities already at the beginning of the training phase (Supplementary Fig. 6c), this is not the case for Tu (see Supplementary Table 1 for exact p-values and test details).”

Finally, we also replaced the graphs of Supplementary Fig. 6c and 7a now showing the mean firing rate of the whole aPC and Tu populations (compared to the previous plots displaying the mean firing rate of only task-responsive units) to facilitate the comparison with the main Fig. 5b-c and 6b-c (which show the full population).

4) In Figure 5d-f, it is not very clear what 1, 2, 3 and the colors stand for. Please clarify.

We thank the Reviewer for highlighting this point. These panels indicate the extent to which unit responses are cue-specific. The green colors indicate the fraction of units that showed significant CS-excited responses to either 1, 2 or 3 different CS, respectively. Conversely, red colors indicate CS-inhibited responses. For clarification, we added a color legend in Figure 5d and specified this information more explicitly in the legend of Figure 5d-f: “Fraction of single units showing significant task-excited, task-inhibited or no responses to either one or more trial types (CS0, CS50, CS100) at CS (d), during wait (e), and at US (f). Of note, the fraction of single units responsive to more than one trial type increased with training.”

5) Similarly in Figure 6d-f, please clarify what 1, 2, 3, and the colors stand for. What does the SOP in the figure legend mean?

As in point 4), we added a color legend in Figure 6d and explained the color code and labeling in the corresponding figure legend: “Fraction of single units showing significant task-excited, task-inhibited or no responses to either one or more trial types (CS0, CS50, CS100) at CS (d), during wait (e), and at US (f). Of note, the fraction of single units responsive to more than one trial type increased with training.”

The abbreviation “SOP” was used for “stimulus-outcome pair” in an earlier version of the manuscript. We now removed the remaining “SOP” from the legend of Figure 6. We thank the Reviewer for this remark.

6) Figure 7b, what the numbers #961, 113, and 1501 mean? If they are example

We agree that the meaning of these numbers in Figure 7b may not be intuitive to the reader. The numbers correspond to the IDs of the shown example units. We now removed the ID numbers from the figure and better conveyed in the corresponding figure legend that the results represent data from example units: “Example single units illustrating (upper left) monotonic RP coding or (upper right) dominant CS50 activation at CS in two units from the transient-cluster, and (lower left) monotonic RP coding during wait of a unit in the ramping-cluster. Average firing rate for each trial type displayed.”

7) Line 513, in the sentence “We finally examined how the multiple circuits encoding RP in the limbic system...”. In fact, the olfactory tubercle and anterior piriform cortex are examined here. I would state “in the Tu and aPC” rather than “the limbic system”.

We agree and provide this more accurate description in the sentence as suggested by the Reviewer (starting l. 347): “We finally examined how the multiple circuits encoding RP in the Tu and aPC updated according to the recent outcome-history.”

8) Supplemental Figure 7c, the percentage of “transient” type should be 70/486=14% instead of 60%.

We apologize for this mistake and thank the Reviewer for noticing it. We corrected the percentage of the transient cluster in the figure (now Supplementary Fig. 8c).

9) The authors may want to justify why only male mice are used in this study.

The decision of only including male mice was made based on three main reasons. First, we were concerned that female behavior might be more variable due to fluctuations in hormone levels across the female estrous cycle. Second, we did not want to use the same recording apparatus for male and female mice to prevent triggering arousal, distraction, or stress during behavioral testing caused by triggering sexual instinctive behavior. Third, and most importantly, we did not expect any sex differences for such basic computations of reward prediction and prediction error investigated in this study. This point may be confirmed in further studies. We added to the Method section “Animals and Husbandry” (starting l. 722): “Only mice of one sex were used in the study to minimize confounding effects of sex-related distress or distraction elicited by residual scents in the recording apparatus and MRI cradle.”

Reply to Reviewer #4 (Remarks to the Author):

In their work “Striatal hub of dynamic and stabilized prediction coding in forebrain networks for olfactory reinforcement learning”, the authors aim to study forebrain regions involved in olfactory reinforcement learning by combining modeling of expected behavior, functional neuroimaging, and electrophysiological recordings. The imaging results indicate that specific olfactory, prefrontal, and striatal regions were involved in processing of odor-related sensory processing and (updated) reward prediction. Importantly, regions appeared to have distinct roles in the behavioral processing. These observations were supported by the electrophysiological recordings that further indicated the different coding roles of olfactory cortex and tubercle in odor processing and reward prediction. I found the experimental setup and findings interesting. The manuscript has relevant aim, is well-designed and mostly well-written, and includes a good amount of technically demanding experiments. My comments / concerns are the following:

We thank the Reviewer for their helpful and constructive comments and addressed them accordingly.

The line numbers indicated in this reply refer to the numbers of the attached manuscript .pdf with highlighted changes in red for reviewers.

Major ones:

1. Movement is a serious concern in fMRI, as also briefly stated by the authors. Echo planar imaging is particularly sensitive to any movement-induced B0 field distortions, as even chest movement can stretch and/or shift the voxels inside the brain in the phase-encoding direction. Such distortions are difficult to correct without advanced techniques, such as continuous field mapping. My concern is that the ICA components in Supp. Fig 2C (particularly C2 but also C4) were categorized as movement artefacts by the authors, and these components overlap exactly the main regions of interest, namely piriform cortex and olfactory tubercle. These suggest that jaw movement and licking affected these regions considerably, and it remains unclear whether the movement-related artefacts still have significant impact on results. Thus, the differences between CS100/50 and CS0 may originate partially or even substantially from the amount of jaw/body movement. I think these limitations should be stated or discussed in more detail.

We agree with the Reviewer that movement is a significant limitation in our study, especially because of the inherent correlation between the effects of interest (anticipation and reward retrieval) and head and jaw movement. We modified the Discussion and Methods sections accordingly, to highlight and discuss this general limitation more in detail:

In the Discussion (starting l. 401): “Echo Planar Imaging is sensitive to motion-induced distortions of the magnetic field, evoked for instance by head movement during licking activity and chest movement during respiration. Superficial lateral and ventral brain structures like Tu and aPC are particularly susceptible to the effects of head and jaw motion. We employed a combination of denoising methods to minimize head-motion effects, but cannot fully exclude a remaining impact of such artifacts (see Methods section ‘Functional MRI denoising’ for a detailed discussion). Optimizing the head fixation, and modified acquisition techniques (such as continuous field mapping 53,54), may further aid in reducing these effects. Nevertheless, active reward retrieval is an essential component of reinforcement learning tasks. Therefore, in many paradigms including ours, the artifacts caused by jaw movements, swallowing or breathing, are correlated with the effects of interest, limiting the possibility to completely remove motion effects from the BOLD data (see also Methods section).”

And further elaborated technical aspects in the Methods (starting l. 971): “As a limitation of this denoising method, it should be noted that the components classified as motion-related due to their typical location at the surface of the brain and around ventricles (C2 and C4) overlap with the regions showing value- and reward-associated BOLD responses (including Tu and aPC). Given that in our paradigm, there is an inherent correlation between the parameters of interest (value and reward), licking activity and head motion, a perfect separation of neuronally-based BOLD response and BOLD changes caused by jaw/head motion is, per se, not possible. Therefore, residual effects of head motion could remain in the data, or some BOLD responses reflecting true task-related neuronal activity might be falsely removed as part of the ICA components classified as motion (especially in the regions that are located superficially, including aPC and Tu). This highlights the importance of the electrophysiological experiments, which are less sensitive to head motion and serve to corroborate/disentangle the fMRI findings.”

2. As the authors already state, fMRI suffers from poor temporal resolution and signal that is confounded by hemodynamic response function. As the HRF can have a decay of several seconds, it might be difficult to conclude whether the brain region was activated during the wait period or US if it was activated during CS (as there was only 1.7 s gap between). I think this should be discussed or mentioned as a limitation related to the fMRI findings.

We agree with the Reviewer that the low temporal resolution of BOLD fMRI is an important limitation. We therefore added the following to the Discussion (starting in l. 394): “Several factors need to be considered in (behaving) fMRI. Even though the mouse hemodynamic response function is several-fold faster than in primates, it will only partially separate sequential task events that occur typically in the range of seconds. In our experiment, image acquisition and trial onsets were temporally non-aligned in order to increase the effective sampling frequency. Nonetheless, low temporal resolution remains a general limitation of fMRI, and is especially relevant in paradigms where two events (like CS and US) are presented in close temporal proximity.”

3. The licking rate during US was 4-7 Hz after CS100 / CS50, which is presumably in line with the “correct” behavior. If I understood correctly, authors aimed to regress the motor activity out in GLM by modeling a licking event as an impulse and convolving it with HRF. However, such high frequency of impulses transforms into constant activity in hemodynamic response. Therefore, it might be difficult to separate somatomotor activation from olfactory and cognitive processing as they typically occurred simultaneously, and one would expect to see a robust and well-localized activity in motor and somatosensory pathways. Furthermore, somatosensory and motor cortices are mentioned in some of the figure legends (e.g., Figure 2), but they are not present in images or covered in the manuscript text. Can authors discuss the expected / missing presence of somatosensory and motor activity particularly in Figure 2F-H?

First, the authors apologize for the incomplete labeling of regions in Fig. 2. We now added the outlines of the somatosensory and motor regions throughout this figure and the corresponding supplementary figure and noted in them in the legend (Fig. 2b “... motor cortex (M1/M2), ... somatosensory cortex (S) ...”).

As the Reviewer correctly points out, we aimed at regressing out the motor activity in the GLM by modeling a licking event as an impulse and convolving it with the HRF. A burst of several licks did indeed translate into a single predicted hemodynamic response, but such that the predicted hemodynamic response returned to baseline between lick bursts of two successive trials. Moreover, the number of licks within a burst modulated the amplitude of the predicted response. Thus, with the chosen method, licking intensity (and not just an unspecific plateau of constant licking activity) is regressed out.

Yet, there is a high correlation between licking activity (and thus, the expected neuronal activation in motor regions) and the neuronal processes of main interest. This is a limitation inherent in instrumental tasks. In consequence, due to the high covariance described, “regressing out” the motor activity by including convolved licks in the model will not lead to a perfect separation between motor-related activation and expectation-/reward-related activation. This is a possible explanation of why there is some, but limited, BOLD activation in motor areas. The only condition where we observe broad recruitment of motor regions is when we regress r (eward) at US; however, in case of regression of $-V$ (CS) at the same US time point, there was only minimal recruitment in these motor regions (cf. Fig. 2f vs 2g). Similarly, little motor activation was observed for the three different CS (new Supplementary Fig. 3b-d), even though there is anticipatory licking already before US, particularly in the rewarded trial types (Supplementary Fig. 3b-c). This, together, suggests that regressing out of lick activity was at least partially effective.

We therefore added the following to the Discussion section (starting l. 412): “Similarly, while we aimed at separating the neuronal correlates of motor activity from that of reward expectation/experience (by including licks as an additional regressor), we cannot assume a perfect separation between these two processes, due to the high correlation between reward expectation/consumption and licking activity. Overall, only experimental designs that would largely decouple task-related muscle activity, outcome and its anticipation would solve these limitations. Yet, such correlations are inherent to instrumental tasks. In this study, functional MRI serves as a discovery tool, and these limitations highlight the importance of corroborating and disentangling regional BOLD recruitments with the electrophysiological recordings.”

In the Methods section, we added the following regarding the HRF convolution of the licks (starting l. 998): “The bursts of licks, convolved with the HRF, translated into predicted hemodynamic responses in such a way that a higher number of licks within the burst lead to a higher amplitude of the predicted hemodynamic response. Therefore, this regressor might be reasonably expected to model the correlates of licking activity in a quantitative fashion, with the limitation that there is a high co-variance between licking activity and the effects of interest (neuronal correlates of reward expectation and consumption), such that we cannot expect a perfect statistical separation between these two processes (see also Discussion).”

4. As stated by the authors, there were image distortions near the posterior piriform cortex and olfactory tubercle, which were key regions of interest. Can authors comment the possibility / likelihood whether because of, e.g., signal pile-up artefacts, the activation in these regions could have been located falsely at,

e.g., the striatal regions, perhaps compromising some of the conclusions? Raw fMRI images including the regions of interest as a supplement figure could clarify the severity of this issue.

We observed ventral signal dropouts in the posterior parts of the acquired EPI volumes. These dropouts may most probably be related to dephasing caused by local magnetic field inhomogeneities. We therefore generally restricted our analysis to the rostral part of the brain containing the olfactory, striatal and frontal regions as our main regions of interest. These signal dropouts affected also the postero-ventral parts of the posterior piriform cortex, preventing reliable detection of BOLD (de-)activation in this area. Due to this limitation, we entirely removed the few minor biological statements about the p(osterior)PC from the manuscript. The pPC was already excluded from the ROI-based functional connectivity analyses before.

For consistency, we revised the composite mask used for the voxel-wise seed connectivity analysis, so that the mask now excludes the pPC. This is described in the Methods section at l. 1069 : “The odor-reward association-learning network contained the following ROIs: olfactory bulb, anterior olfactory nucleus, aPC, as well as Tu, NAc, dorsal striatum, orbitofrontal cortex, mPFC and agranular insular cortex (see Fig. 3a). ROI parcellations were anatomically defined as per Paxinos and Watson (2009), with the exception of a refined parcellation for the aPC seed, which comprised the anterior portion of the piriform cortex delimited posteriorly at +0.8 mm relative to Bregma. As described in the section ‘Functional MRI acquisition’, the posterior piriform cortex was also not included in the task-related connectivity analyses as we observed signal dropout.” The total number of voxels is updated to 2014 (l. 1077). This modification resulted only in minute changes in the connectivity maps (that were replaced in Fig. 3d,e and Supplementary Fig. 4) and did not affect our previous conclusions.

Importantly, we did not find comparable signal dropouts in key regions of interest, namely olfactory tubercle and anterior piriform cortex. After field map distortion correction, we also do not observe any evidence of signal pile-up artefacts. To illustrate image quality, as suggested by the Reviewer, we now exemplarily show fMRI images of three different subjects in Supplementary Figure 2c, described in the corresponding figure legend: “Bottom: Examples of functional images in 8 coronal slices for three different animals. Displayed images are after initial preprocessing (realignment, field map distortion correction, slice-timing correction, spatial normalization), but without smoothing or motion artifact correction. For display purposes, images are masked with a template brain mask.”

We also included the following sentence in the Methods section (l. 906): “More posterior slices frequently had ventral signal dropout due to B0 field inhomogeneity, so that we restricted the analysis to the rostral parts of the brain, containing the olfactory, striatal and higher-order regions.”

5. Behavior is known to be modified by stress. Discussion related to the effects of body restraint- and noise-induced stress on learning and related processing, and subsequently on the findings and their generalization would be required.

We thank the reviewer for raising this point. In awake rodent imaging, stress due to scanner noise and body restraint is a potential confound that can affect learning, behavioral performance, and the neuronal signal. To minimize stress, mice were gradually habituated to head fixation, the odor-guided learning task and MRI noise. Former studies have proven that an in-depth habituation to the MRI environment reduced physiological stress parameters and motion significantly (King et al., 2005; Yoshida et al., 2016). We cannot fully exclude or quantify the impact of stress on the biological findings. However, our behavioral data support that mice were able to complete a complex cognitive task in the MRI environment with performance levels comparable to the sessions outside of the scanner. This suggests a rather minor impact of stress on our fMRI findings. We now discuss the influence of stress on behavior and neuronal processing in more detail (starting l. 808): “Both human and animal subjects are expected to experience elevated stress levels in MRI due to limitations of movement and scanner noise. Thus, in general, it can be assumed that learning in the scanner occurs under elevated stress conditions. As described above, habituation to the MRI environment and noise served to reduce physiological stress parameters and motion 69,70. The efficacy of the habituation procedure is supported in this study by the comparable behavioral performance observed inside and outside the scanner (Supplementary Fig. 1a,f).”

6. At general level, the description of methods related to data analysis would greatly benefit of revisioning. It is possible to get a rough idea by multiple readings and reasoning, but the flow of the text could be much better. I could not find a clear definition for “s” or “st” (lines 792-817). Therefore, it was difficult to determine the values in V. The definitions should be clear and the order of equations and explanations easy for reader. It was unclear what V(0) means? When is s = 0? Also, the definition for initial V comes very late for readability.

We apologize for the initial lack of clarity in the description of the Temporal Difference Model. We shortened and restructured the section to improve reading comprehension (starting l. 840):

“Reinforcement learning model

We parametrized the RP and prediction error with a temporal difference model TD(0) with no eligibility trace or discount factor ⁷¹. The model was designed on a trial-by-trial basis. In each trial, we considered three time points: the baseline before the trial started ($t = 0$), the odor presentation ($t = 1$), and the reward delivery ($t = 2$). Each time point was characterized by a state s defined by the trial epoch and, when present, the co-occurring stimulus, for a total of five states. At $t = 0$, there was only one possible state s_0 . At $t = 1$, there were three possible states s_1 corresponding to the three conditioned stimuli (CS100, CS50, CS0). At $t = 2$, there was only one state s_2 representing the moment in the trial associated with the unconditioned stimulus (the specific presence/absence of reward in each trial is captured in the model by the separate variable r_t , see next paragraph).

To each state s_t was assigned a value $V(s_t)$, reflecting the RP. Further, an outcome r_t was assigned to each time point. r_t encodes the reward at time t as a binary variable 1 or 0. At baseline and during CS presentation, respectively, r_0 and r_1 were set to 0. At US, r_2 was 1 in case of reward and 0 otherwise. $V(s_2)$ was set to 0 as the model was designed on a trial basis and $t = 2$ was the last time point of the trial.

The values $V(s_t)$ were updated trial by trial by a prediction error δ . The prediction error compares two successive value predictions and is defined as:

$$\delta_t = r_t + V(s_t) - V(s_{t-1}). \quad (1)$$

Intuitively, the current reward r_t and the current expectation for the immediate future $V(s_t)$ are summed and compared to the preceding expectation $V(s_{t-1})$. A positive/negative prediction error δ has the effect of increasing/decreasing the value of the previous state according to:

$$V(s_t) \leftarrow V(s_t) + \alpha \delta_{t+1}. \quad (2)$$

with learning rate α (in our model, α was set as free parameter and estimated using pupillary data, see next paragraph). For example, a reward delivery ($r_2 = 1$) after CS50 will lead to a positive δ_2 , and this will update (increase) the value of the CS50 stimuli $V(s_1)$.

In the main text, we refer to $V(s_1)$ as $V(\text{CS})$, $V(s_2)$ as $V(\text{US})$, and δ_2 as $\delta(\text{US})$.

Parameter estimation and selection

The learning rate α was set as free parameter of the model and estimated using pupillary data from the pupil imaging sessions described above (n=10 sessions). The average pupil dilation d (or more precisely the average percent change of pupil dilation from baseline) was modeled as a function of the expected value V and the prediction error δ :

$$d(t) = \sum_{\tau \in \mathbb{N}=0}^t a_\tau V(s_{t-\tau}) + \sum_{\tau \in \mathbb{N}=0}^{t-1} b_\tau |\delta(t-\tau)| \quad (3)$$

where $t = \{0, 1, 2\}$ (0 for the baseline, 1 for the odor presentation and 2 for the reward time). a_τ and b_τ were set as free model parameters. $d(t)$ indicated the average pupil dilation at different times in the trial: we defined $d(0)$ as the average dilation in the interval from -2 to 0 s; $d(1)$ as the average dilation in the interval from 0 to 2.7 s, and $d(2)$ as the average dilation in the interval from 2.7 to 5.4 s. The model parameters $\vartheta = \{\alpha, a_\tau, b_\tau\}$ were estimated minimizing the distance between the modeled (eq. 3) and the real average pupil dilation. ...”

7.1 There are some uncertainties related to the GLM analyses at lines 938-951. First, can authors elaborate the rationale for modeling monotonic RP (GLM #2) in addition to the V-modulated CS (GLM #1)? Apparently, these model the same outcome, but doesn't monotonic RP have less signal and is less suitable?

We thank the Reviewer for these questions, and aim to clarify several aspects of the GLM analysis.

In GLM #1, when modeling $V(\text{CS})$, this will detect also those regions whose activation follows a pattern of “CS100 \sim CS50 \gg CS0” (i.e., regions that are generally active in response to potentially rewarded stimuli but do not differentiate between different probabilities of reward). By modeling the monotonic RP (GLM #2), we probed which brain regions represent RP in a stricter fashion, such that higher reward expectation is reflected by a stronger BOLD or neuronal response (i.e. “CS100 > CS50 > CS0”). Inherently, all units/voxels showing this activation pattern will also correlate with $V(\text{CS})$, but not vice versa. It would therefore be expected that the regions representing monotonic RP (GLM #2) are a subset of those generally correlating with $V(\text{CS})$ (GLM #1).

We aimed to clarify this in the relevant sections of the manuscript. First, we used the numbering of the GLMs (“GLM 1”, “GLM 2” and “GLM 3”) both in the Results and the Methods section, for easier cross-reference. We then added in the Result section a general description of the GLMs performed in the manuscript (starting l. 121): “... we computed three different general linear models (henceforth termed GLM 1, GLM 2 and GLM 3) ...” and rephrased the relevant sentences, to make the concept of monotonic RP clearer (starting l. 138): “GLM 1 does however not distinguish between a strict monotonic neuronal representation of $V(CS)$ ($CS100 > CS50 > C0$), and a simple binary differentiation between presence or absence of reward expectation (e.g., $CS100 = CS50 > CS0$). Both types of representations would result in a significant correlation with $V(CS)$. Indeed, CS100 and CS50 recruited similar brain regions, unlike CS0 (Supplementary Fig. 3a-d). To probe which brain regions represent RP in a monotonic fashion, such that higher reward expectation is reflected by a stronger BOLD or neuronal response (i.e. “ $CS100 > CS50 > CS0$ ”), we computed a separate GLM 2, where the three CS trial types were modeled with individual regressors. We determined BOLD correlates of monotonic RP by intersecting significance maps of specific contrasts ($(CS100 > CS50) \cap (CS50 > CS0)$, or vice versa for negative contrasts $(CS100 < CS50) \cap (CS50 < CS0)$, Supplementary Fig. 3e-f).”

7.2 The only clearly different part between GLM #1 and GLM #2 is the outcome history in the CS50-US.

Again, the authors apologize for an initial lack of clarity. By answering the previous part of this comment, the authors hope to have clarified that there are two substantial differences between GLM #1 and GLM #2: First, GLM #2 models each CS type (CS0, CS50, CS100) with a separate regressor, whereas GLM #1 represents them with a single regressor, which is then parametrically modulated by a scalar (namely, $V(CS)$). The second difference, as the Reviewer points out, is that GLM #2 includes the recent outcome history as a parametric modulator of the CS50 regressor. Such a parametric modulation would not have been equally informative in GLM #1, because, as we show in **Figure 1i**, $V(CS)$ varies with the recent outcome history – in other words, recent outcome history and $V(CS)$ are partly collinear, and the shared variability is the variability of interest (corresponding to the impact of recent outcome history on reward expectation). This variability would have been doubly introduced into the model, making interpretation more difficult.

To make this point clearer, we added the following to the Methods section (starting l. 1010): “Note that such a parametric modulation with recent outcome history would not be equally informative in GLM 1, since $V(CS)$ is already included in GLM 1 as a parametric modulator, and partially contains the variability of interest (outcome history) (cf. Figure 1i).”

7.3 As the previous CS50 outcome is encoded as a modulation in the present CS50 event, how does the model in GLM #2 separate the present CS50 response and the response to the previous round CS50 from each other?

As the reviewer points out, the same variability (namely the outcome of the previous CS50 trial) is modeled twice in GLM 2: firstly, at the US timepoint of the previous CS50 trial; secondly, as a parametric modulator of the current CS50 regressor at the CS timepoint. However, the long temporal delay between these two regressors (since we had a trial duration of 10-12 seconds), together with the relatively short mouse HRF, should minimize any possibility that the same neuronal activity is modeled by the two regressors. Specifically, in case of the minimum trial duration (10 sec), and considering that the US timepoint is 2.7 seconds after onset of CS, the two events are still $10 - 2.7 = 7.3$ seconds apart. Further, in the majority of cases, the previous CS50 trial will not be the immediately preceding trial (since a given trial is preceded in two-third of cases by CS0 or CS100 trials). Overall, it is therefore very unlikely that the effect of recent outcome history can be explained by a prolonged hemodynamic response to the preceding reward event. Further, we here provide also electrophysiological ground-truth that the history effect is truly expressed in the Tu. Nevertheless, we included a note concerning this issue in the methods section (starting l. 1014): “Note that in this model, the same variability (namely the reward/non-reward outcome of a given CS50 trial) is modeled twice, specifically at the US timepoint of the trial itself, and as a parametric modulator (outcome history) of the next CS50 trial. However, since the trial duration (jittered between 10 and 12 seconds) is long compared to the assumed duration of the hemodynamic response in mice (Lebhardt et al., 2016; Chen et al., 2020), it is unlikely that an effect of the parametric modulator should be driven by a prolonged hemodynamic response to the preceding CS50 reward event itself. Also note that in a majority of cases, the previous CS50 trial is not the immediately preceding trial (since a given trial is preceded in two-third of cases by CS0 or CS100 trials). Further, note that the history modulation was also observed in neuronal recordings from the Tu.”

7.4 Would it be possible to model the history equally well in GLM #1 as an additional binary (or 3-valued pair (if you want to exclude CS100 and CS0) of) regressor(s)? A carefully designed regressor might be even more appropriate if it was specifically designed to correspond only to the event history contrast.

As discussed above (7.2), additionally modeling the history in GLM #1 would in our view not be equally powerful/informative, since the outcome history would be doubly modeled, as $V(\text{CS})$ already contains the effect of outcome history (cf. Figure 1i).

7.5 Second, the GLM #3 models for complement of $V(\text{CS})$ and r . It appears that a single GLM was done, so it remains unclear how the US events with both $-V(\text{CS})$ and r were modulated. What weights were used and how were they chosen?

Again, we have modified the Methods section to clarify. Indeed, the parametric modulation with $-V(\text{CS})$ and r was done within the same GLM (#3), and such that one single regressor (a generic US stick regressor) had two parametric modulators. Regarding the $-V(\text{CS})$, we used the scalars generated by the reinforcement learning model. For r , we used binary values (1 for “reward” and 0 for “no reward”). We added this information to the Methods section (starting l. 1025): “... and the latter was parametrically modulated by two regressors, namely $-V(\text{CS})$ and r . While $V(\text{CS})$ was estimated by the TD model (identical to GLM 1), we used binary values for r (1 and 0 representing reward and no reward, respectively).”

7.6 If the modulation worked, I see no problems with the individual $-V$ and r maps. However, considering the map intersection as representing the common properties of both in the same spatial location is questionable. Isn't it possible that the common regions are highlighted for completely different reasons but are accidentally spatially located in the same spot? Modeling the interaction between $-V$ and r explicitly in the GLM would make the result more convincing.

We agree that testing the statistical interaction between $-V(\text{CS})$ and r would also be a way of testing for correlates of prediction error. However, inherent in the Temporal Difference (TD) learning model which we used (cf. Seymour et al., 2004), the prediction error at US was defined as the difference between reward and prior expectation (i.e., $r - V(\text{CS})$), not as their statistical interaction (i.e. product term). We therefore, following (Niv et al., 2012), sought to identify regions whose BOLD response is correlated with both components of the prediction error. In performing this analysis, we did not attempt to make inferences about potential *mechanisms* of prediction error computation. Our aim was – more modestly – to detect regions where both components of prediction error (namely $-V(\text{CS})$ and r) are represented. We rephrased the relevant sentence in the Results section to account for this (starting l. 164): “... thus indicating a relatively confined set of candidate regions that could be involved in the prediction error at US for olfactory stimulus-outcome learning. Co-localization of both prediction error components (r and $-V(\text{CS})$) could in principle emerge also from two independent processes. The expression of the prediction error in candidate regions will be therefore further explored by electrophysiology.” In the Abstract and Discussion, we also re-phrased accordingly (l. 29): “... the downstream olfactory tubercle of the ventral striatum expresses comprehensively reward prediction, its dynamic updating, and prediction error *components*.”; and in the Discussion (starting l. 419): “Through the mesoscale fMRI approach, we found that RP and prediction error components were differentially represented among the primary olfactory cortices, subcortical circuits and higher-order brain regions.” The wording (“prediction error components”) is now generally used in the manuscript when discussing our findings in the Discussion.

Minor ones:

8. As stated at Line 49 and in the abstract, fMRI allows rather unbiased identification of brain regions involved in behavioral processing. However, as authors imaged only forebrain, I would not emphasize the unbiased identification aspect in the current study.

As suggested by the Reviewer, we removed the emphasis on the ‘unbiased’ identification of brain regions from the abstract (l. 24) “Identifying the circuits responsible for cognition and understanding their embedded computations is a challenge for neuroscience.” and in the introduction at former l. 49 (now l. 53): “To identify the forebrain regions involved in these computations, a possible solution would be fMRI in task-performing rodents combined with behavioral parametrization.”

9.1 The exact setup of odor delivery system remains unclear. As the tubing to the MRI bore is quite long, there are many important factors to be clarified. I did not find the exact location of odor supplies and photo-ionization detector, and how long was the distance from these to the mice.

The tubing system for odor application ran from the olfactometer located in the MRI control room to the odor port integrated into the MRI cradle. We used a tubing system with a fixed length of 3m for all subsequent fMRI sessions. As determined with a photo-ionization detector (PID), the measurement of the odor latency from final

odor valve opening to odor exposition at the odor port with the 3 m tubing was determined outside of the MR scanner room. We did not perform PID measurements during fMRI acquisition. The PID measurements revealed an odor latency of 400 ms with sharp onset/offset kinetics (10-to-90% of approx. 20 ms, and 90%-to-10% of approx. 40 ms). We apologize for the incomplete description of both PID measurements and setup configuration.

We now provide more detailed specifications about the MRI-compatible behavioral setup in the Methods (l. 751): “Odorized air and water were both guided from the setup located in the control room to the animal bed inside the magnet bore through 1/32 inch I.D. inert Polytetrafluoroethylene tubing (NResearch) connected to the odor and lick ports.” and (starting l. 754): “A single delivery tubing was used for all odors. For all MRI experiments, a 3.0 m long tubing was used to deliver the odorized air to the odor port. For this tubing, the latency for odor detection at the odor port after final valve opening at the olfactometer was determined initially with 1:10 diluted amyl acetate outside the scanner with a photo-ionization detector (miniPID, AuroraScientific). The latency was approx. 400 ms. Thus, air was exchanged in the tubing by clean air at least 25 times between two consecutive odor applications (considering the airflow rate of the olfactometer). The steepness (time from 10% to 90% of peak) of the odor onset was approx. 20 ms and the offset approx. 40 ms (time from 90% to 10% of peak). The olfactometer was placed in the MR control room and the odor port placed in front of the snout of the animal head-fixed on the MRI cradle (Fig. 2a). Animals were head-fixed without the need of sedation or anesthesia.”

9.2 Were different odors delivered with dedicated tubes, or was the same tube used for all odors? Was there a risk that cross-contamination would confound the experiments?

The same tube was used for application of the three distinct odors. We considered the risk of cross-contamination as negligible. First, we used a Polytetrafluoroethylene tubing system with chemically inert material properties. Second, the olfactometer produced a constant airflow running to the animal’s nose. Considering a trial duration of 10 to 12 s and an air transit time of 0.4 s (transit time from the final valve to the odor port), air was exchanged on average $11 \text{ s} / 0.4 \text{ s} = 27.5$ times between two consecutive trials. Third, the analysis of licking frequency and pupil dilation proved the animals’ ability to clearly distinguish the three distinct odor cues. Also, the study did not examine fine odor discrimination for instance of odor mixtures where spurious cross-contamination would be a serious concern. We added the information in the paragraph of the Method section referred to in point 9.1 above.

9.3 If odor ports and detector were outside the bore, what was the delivery time to the mouse and how was it measured to obtain exact timing with the fMRI? If odor ports and detector were inside the MRI bore, of which material were they made of?

The MRI-compatible odor port (Polytetrafluoroethylene) was located inside the MRI bore. The odor port was installed into the MRI cradle and positioned directly in front of the animal’s nose (see Figure 2a). Importantly, the last metallic part was the final valve (located outside the scanner room) that was at the beginning of the 3 m tubing that ended at the odor port in the MR bore (odor delivery time 400 ms, see 9.1). Digital traces of both odor valve opening and TTL pulses from the MR scanner were recorded with an RHD2000 interface board (Intan Technologies) and later used for temporal alignment. We now write in the Methods (l. 761): “The olfactometer was placed in the MR control room and the odor port placed in front of the snout of the animal head-fixed on the MRI cradle (Fig. 2a).”, and (l. 769): “Output signals from the olfactometer, the optical lick detector, the water pump, and TTL pulses from the scanner were all recorded with the same RHD2000 interface board (Intan Technologies) with a sampling frequency of 1 kHz. These data were used to align trial times during analyses.”

10. The stimuli were short and expected BOLD responses are thus very small, likely less than 1 %. I’m sure it would be highly interesting for readers to see examples of BOLD time series and amplitudes, perhaps together with the general linear models.

As suggested by the reviewer, we now provide BOLD percent signal changes for selected regions of interest (olfactory tubercle, anterior piriform cortex, dorsal striatum) in the new Supplementary Fig. 2e. We calculated BOLD percent signal changes using the ArtRepair Toolbox for SPM (<https://cibsr.stanford.edu/tools/human-brain-project/artrepair-software.html>). The toolbox scales beta weight estimates obtained from the GLM analyses into percent signal changes using three different scaling factors: (1) the peak value of the design matrix X (*peak*), (2) the average of the constant term (*bmean*), (3) the sum of the positive terms in the contrast (*contrastsum*) (Luo and Nichols, 2003). Using only beta images with a *contrastsum* = 1, the scaling factor (3) was negligible for the subsequent calculations. Thus, the scaling for percent signal change of beta weight estimates was conducted according to the equation 1:

$$\beta(\%) = \beta * \left(\frac{peak}{bmean} \right) * 100 \quad (1)$$

where β was the averaged beta weight over all voxels of a given region. The average percent signal changes for the event types {CS100, CS50, CS0} at CS and {CS100, CS50→R} at US ranged mostly between 0 and 2% in the example regions, with similar amplitudes for inner (i.e., dorsal striatum) and outer (i.e., Tu and aPC) brain regions. Despite the fact that little is known about BOLD responses and their amplitudes in awake, behaving mice, this percentage range of BOLD response amplitudes is in line with results from other fMRI studies in awake mice (Han et al., 2019, *Neuroimage* 188, 733-742).

In the Results section, we added (l. 126): “The average magnitude of BOLD responses (computed as percent signal change) was mostly in the range between 0 and 2% (Supplementary Fig. 2e), similar to a prior behaving fMRI study in rodents (Chen et al. 2020).”

We added a plot showing percent signal changes (Supplementary Fig. 2e), and the corresponding explanation in the legend of new Supplementary Fig. 2e: “Magnitude of BOLD responses, expressed as percent signal change values (PSC) averaged over all sessions (n = 51) for Tu and aPC. Additionally, the dorsal striatum is shown as an exemplary region with widespread positive BOLD response in the paradigm. As detailed in the Methods section, we scaled PSC values in a region-specific manner, to avoid underestimating PSC in ventral regions, which have lower absolute signal intensities owing to the dorsally-located surface receive coil. Red rhombuses indicate average PSC. Box plots showing medians with solid bars indicating the 25th and 75th percentiles.”

In the Methods section, we added the following (starting l. 1039): “To explore the magnitude of BOLD responses, percent signal change (PSC) was computed from the beta estimates using the ArtRepair toolbox for SPM (<http://cibsr.stanford.edu/tools/ArtRepair/ArtRepair.htm>). In Supplementary Fig. 2e, this is shown for the main “active” conditions, whose beta estimates were computed in GLM 2, namely: CS100, CS50, CS0, reward after CS100, and reward after CS50. In order to account for regional variation in signal intensities with a surface receiver coil (lower signal intensity at the ventral portions of the brain), we modified the method by scaling the PSC not to the whole-brain signal intensity, but to the region-wise signal intensity (i.e. beta of the constant term averaged over the respective region, not over the whole brain). Compared to the original method, this leads to relatively larger PSC values for the ventral regions (Tu and aPC), which have both lower absolute signal intensities and lower absolute response magnitudes, compared to more dorsal regions (e.g., dorsal striatum).”

11. The term functional connectivity may be misleading in this context, as it typically refers to correlating low-frequency fluctuations during long time periods. Perhaps “co-activation” would be more justified? Moreover, brain regions may be activated at the same time but by different inputs that may not be linked closely to each other, questioning the functional connectivity.

We thank the reviewer for raising this point. We agree that the most common meaning of functional connectivity (FC) refers to low-frequency co-fluctuations, very often during resting state experiments. Here, we investigated inter-regional correlations of a different sort, namely temporal co-fluctuations of beta-coefficients. This was implemented using the Beta Series Correlation (BASCO) method (described in the Methods section *Task-related functional connectivity analysis*). By this method, inter-regional correlations are computed in a “within-condition” fashion – i.e., testing whether pairs of regions show similar *variations* in their typical task response. Therefore, the functional connectivity computed with this method is independent of whether two regions are on average co-active in response to a given task event. In other words, within this approach, region A and region B might show high functional connectivity even if, for example, region A generally shows activation and region B does not in response to a certain stimulus. Therefore, we would fear that the term “co-activation” might be misleading (as it might be understood that two regions are activated/deactivated by the same event types). We therefore propose to use “task-related functional connectivity” as a less ambiguous terminology. This (Göttlich M et al. (2015) *Front Syst Neurosci* 9:126.) and similar expressions, such as “context-modulated functional connectivity” (Cisler JM et al. (2014) *Neuroimage* 84:1042-52) have been used by others to describe such analyses. Taken together, we agree with the Reviewer that the underlying regional inputs cannot be differentiated by this analysis and, therefore, causal relationships (“effective connectivity”) cannot be inferred.

For clarification, we used the term “task-related functional connectivity” when introducing and describing the analysis in the Results and Methods sections, and clearly marked “functional connectivity” as an abbreviation (starting l. 185, Results): “Briefly, task-related functional connectivity (henceforth also called “functional connectivity” for brevity) between either Tu or aPC ...”. We also added the following (Results section, starting l. 189): “Note that this method is correlational in nature and does not allow inferences about causality or common synaptic inputs.”

12. In Figure 3B-C, it remains unclear why there are empty slots in the matrices. Is it because the correlations in those slots did not exceed statistical significance?

Indeed, in Figure 3b-c, the uncolored “empty” slots in the Pearson correlation matrices indicate that the correlation for the respective pair of regions did not exceed statistical significance at the group-level. We added this information to the legend of Figure 3b: “Pairwise correlations that did not reach significance are indicated by empty boxes.”

13. At line 687, it would be better to also mention the active compound of the drug (Metacam). Commercial products or their names may disappear from the market.

We now mention the active compound of the analgesic used for pre- and post-surgical treatment (l. 726): “Analgesia (meloxicam, Metacam Boehringer Ingelheim) was administered before and after surgery.” We now also specify the administered local anesthesia (l. 731): “... and lidocaine was administered topically.”

14. There are no information about the post-operative care after the head-bar implantation, which I assume would be needed according to the regulations.

We apologize that no information about the post-operative care was given in the manuscript. We now describe the post-operative care in the Methods (starting l. 737): “Animals received additionally a subcutaneous injection of buprenorphine directly after surgery. All animals recovered within 10-20 minutes after cessation of isoflurane anesthesia. Thereafter, they were transferred to their home cages in good general condition. If signs of post-surgery pain were observed, additional meloxicam was administered. All animals were monitored daily during the entire experiment.”

15. There are no information whether anesthesia was used during the habituation or MRI preparations. Does this mean that the animals were fixed to the holder while they were awake? This should be clarified.

We thank the Reviewer for highlighting this critical step in the experiment. No anesthesia or sedation was used in our experimental design, neither during the habituation process nor during the entire fMRI acquisition including the preparation phase. The animals were head-fixed in the awake state. We now stress explicitly in the Results (l. 111): “No anesthesia or sedation was used at any stage of the fMRI experiment, including habituation and image acquisition.” And in the methods (l. 763): “Animals were head-fixed without the need of sedation or anesthesia.”

16. The inter-trial interval would be worth of mentioning in the Figure 1 legend, as I had difficulties to find it. Moreover, was the same 10-12 s used in all experiments?

We thank the reviewer for highlighting this. The same trial duration (between 10 to 12s) was used in both fMRI and electrophysiological experiments. As suggested, we now indicate the trial duration in the legend of Figure 1a: “The time interval between CS onsets of consecutive trials (trial duration) was jittered between 10-12 s.” We replaced inter-trial interval (ITI) in the Methods by ‘trial duration’ to avoid confusion.

17. The type of motion correction in fMRI data should be stated, e.g. affine or rigid.

We had applied a rigid-body transformation to each volume for spatial realignment to the middle volume. We added the information about the type of realignment to the Methods (l. 921): “..., correction for head movement by realignment to the middle volume using a rigid-body transformation, ...”

18. In addition to average realignment parameters, the maximum values of the realignment parameters should be reported. This would indicate the robustness of the head-fixation, and the magnitude of the movement, which both function as indicators of the data quality. The importance of this aspect is emphasized in the current work as the authors did not perform motion scrubbing.

We agree and added this information to the new Supplementary Fig. 2. Therefore, we replaced the old Supplementary Fig. 2b (which showed mean values) and now show the distributions of the realignment parameters for each spatial dimension. Besides the average realignment parameters (Mean), we now indicate the average session maximum (i.e., the mean of all session maxima) (Max) for each translation/rotation in the new Supplementary Fig. 2b.

New legend of Supplementary Fig. 2b: “Distributions of realignment parameters for the three translations and the three rotations. Realignment parameters are extracted from the fMRI preprocessing and serve as quantification of head motion after habituation to the scanner environment. Average values (Mean), and average session-wise maximum values (Max) are given for each spatial dimension (n = 51 sessions in 18 animals).”

19. Value for learning rate ($\alpha = 0.28$) could be mentioned in the text (results or methods), as I found it only in figure legend.

We now specify the inferred value for the learning rate in the Methods in (l. 878): “After the average learning rate α had been estimated on the basis of the 10 sessions with pupil imaging (estimated $\alpha=0.28$), it was used to build a temporal difference model for 151 different trial realizations.”

20. At lines 793-795, the authors write “This reward prediction (RP) signal is adjusted in time through the difference between the predicted and the actual outcome namely the reward prediction error (PE) at US.” What does “adjusted in time” exactly mean?

We agree with the Reviewer that the term “adjusted in time” was confusing. The term “adjusted in time” was intended to specify that the reward prediction signal was built and updated on a trial-by-trial basis, which implies a temporal dimension. In general, we revised the description of the Temporal Difference (TD) model in the Methods to improve readability (see point 6 of the reply); the expression “adjusted in time” is now removed.

21. At lines 879-881, the authors state “...spatial normalization to a mouse brain template in the Paxinos stereotactic coordinate system, by applying the non-linear normalization parameters of the structural images to the functional images”. However, I did not notice that any processing was reported for structural images that could be applied on the functional images.

We thank the Reviewer for this remark. We now mention the preprocessing of the structural images more explicitly in the Methods. (l. 916): “After brain extraction using in-house developed code based on a pulse-coupled neural network algorithm, structural images were non-linearly normalized by segmentation to high-resolution tissue probability maps (Biedermann et al., 2012), which had been transformed to Paxinos space.”

22. As a minor note, can authors comment whether four equal-sized groups (instead of three) would have been more suitable, regarding e.g. statistical power?

We agree with the Reviewer that an experimental design with four equal-sized groups (CS100, CS50R, CS50N, CS0) would have increased statistical power in the analyses dividing CS50 into subgroups (e.g. depending on the outcome). However, we were concerned about modifying the salience of the three conditioned stimuli (CS100, CS50, CS0) when conducting twice as many CS50 trials within one session. The higher frequency of CS50-trials would have made CS0 and CS100 trials as ‘rare’ events potentially more salient. To avoid possible distortions of salience within the behavioral task, we defined three equal-sized groups (CS100, CS50 and CS0).

REVIEWER COMMENTS

Reviewer #4 (Remarks to the Author):

Overall, I appreciate the responses of the authors, as many of the comments or issues were taken into account in excellent manner. Particularly the explanation of methodology has improved significantly. Nevertheless, I still have major concerns related to the fMRI data reliability.

If I interpret the new supplementary data correctly, average translation across all volumes in, e.g., up-down direction (0.059 mm) was as high as 23.6 % of the in-plane voxel size (0.250 mm), while maximum translation (0.494 mm) was almost twice the in-plane voxel size. Proper handling of such continuous large-scale movement would very likely require more advanced techniques, such as continuous field mapping to correct, e.g., movement-induced image distortions. Also, as stated by the authors, certain commonly used denoising methods diminished the fMRI responses, which naturally raises the concern whether the responses are mainly lacking motion-related artefacts. Additionally, I was wondering why BOLD responses during CS50->N were not included in the new Supp. Fig 2E? These would potentially provide evidence related to the specificity of the fMRI findings.

I appreciate that the authors emphasize fMRI as a discovery tool in the revised manuscript due to motion-related limitations, but considering the current role of fMRI in the manuscript, it is difficult to pass the fact that most of the fMRI results, which are quite many, may represent partially or even substantially motion artefacts.

Reply to the point of Reviewer #4: *Overall, I appreciate the responses of the authors, as many of the comments or issues were taken into account in excellent manner. Particularly the explanation of methodology has improved significantly. Nevertheless, I still have major concerns related to the fMRI data reliability.*

In the following we address the concerns raised by the reviewer and discuss several points including additional analyses supporting that the fMRI findings are not due to motion-related artifacts, but reflect the underlying neuronal information.

Reviewer Point 1.1: *If I interpret the new supplementary data correctly, average translation across all volumes in, e.g., up-down direction (0.059 mm) was as high as 23.6 % of the in-plane voxel size (0.250 mm), while maximum translation (0.494 mm) was almost twice the in-plane voxel size.*

The maximum motion displacement in one dimension („up“) referred to by the reviewer represents the average maximum displacement of all sessions. The influence of the maximal frame displacement becomes small as a rare event within the fMRI statistical model. The level of motion in our experiment is expected given data from two recent high-rank publications performing resting-state fMRI in passive awake mice (Fig. S3B in Tsurugizawa et al. 2020 Sci Adv 6:eaav4520; Fig. S6B in Gutierrez-Barragan et al. 2022 Curr Biol 32:631-644, where frame-wise displacement was determined after ‘time despiking’, which in our experience reduces the calculated displacement). We added this point in the discussion (l. 407 of the manuscript file with highlighted changes): “The motion level that we observed in the behavioral task, is expected given data of recent awake resting-state imaging studies in mice (Tsurugizawa et al. 2020; Gutierrez-Barragan et al. 2022).”

Reviewer Point 1.2: *Proper handling of such continuous large-scale movement would very likely require more advanced techniques, such as continuous field mapping to correct, e.g., movement-induced image distortions.*

The Reviewer suggested continuous field mapping as a motion correction technique. Unfortunately, most of the correction methods we found in this regard are prospective modifications of the EPI sequences involved, requiring the additional acquisition of multiple echo-times and/or reversed EPI blip directions. These sequences are not readily available on our scanner (94/20 Bruker Biospec, running ParaVision 6 software). Importantly, this approach would limit the spatial or temporal resolution of the measurements. This creates fundamental problems when combined with the short hemodynamic response function (HRF) of mice and with an event related design. The reduced time resolution makes it difficult to obtain sufficient sampling points in each trial and drastically reduces the sampling of the underlying HRF, which is several-fold faster in mice than in primates (e.g. Lehardt et al., 2016). Thus, as a trade-off, the data sampling and possibility to distinguish task events (CS and US) would be lost, making this method unfeasible for the study of such learning processes.

Reviewer Point 1.3: *Also, as stated by the authors, certain commonly used denoising methods diminished the fMRI responses, which naturally raises the concern whether the responses are mainly licking motion-related artefacts.*

If we understand correctly, the reviewer assumes that some commonly used denoising methods diminished the fMRI responses in our data. We think that this assumption is not correct – the approaches we explored, did not generally reduce beta values (as described in the Methods section ‘Functional MRI denoising’). However, for instance wavelet despiking introduced implausible temporal ‘smearing’ such that activations started slightly before the trial.

Regarding the remaining concern about whether the signal – after denoising – is still substantially driven by motion artifacts, there are two ways one can answer this doubt: 1) show that the same result occurs also when limiting the analysis to low motion conditions (see Point 1.5 A,B,C); and 2) provide evidence of the underlying neuronal activity (see Point 1.5 D).

Reviewer Point 1.4: *Additionally, I was wondering why BOLD responses during CS50→N were not included in the new Suppl. Fig 2E? These would potentially provide evidence related to the specificity of the fMRI findings.*

We now added the percent signal change values for CS50→N at US to Suppl. Fig. 2e. The graphs were added in the previous round of the revision (related to point 10) to illustrate the amplitude of BOLD responses to the task stimuli (odors and rewards). In light of the present discussion, however, CS50→N may provide additional insight. If we understand the Reviewer correctly, one of his concerns is that the negative BOLD response in the olfactory tubercle is due to licking. According to this concern, motion would produce BOLD deactivation. While licking drops in the US window in CS50→N trials compared to the delay period, licking intensity in CS50→R trials further increases in the US window (Fig. 1b). However, at US, the BOLD response of CS→N is more negative than the one of CS→R in the olfactory tubercle (Suppl. Fig. 2e). This supporting evidence adds to the more direct testing of the influence of motion on BOLD negativity in Point 1.5A-B.

Reviewer Point 1.5: *I appreciate that the authors emphasize fMRI as a discovery tool in the revised manuscript due to motion-related limitations, but considering the current role of fMRI in the manuscript, it is difficult to pass the fact that most of the fMRI results, which are quite many, may represent partially or even substantially motion artefacts.*

We provide here four additional lines of evidence that the fMRI results are not explained by motion artifacts:

- A. As a first validation, we took advantage of the fact that mice underwent extensive conditioning and, therefore, CS-triggered value responses were reinforced and became stable (cf. Figs. 5-8 in the manuscript and Oetl et al. 2020). Thus, the neuronal value response at CS remains

stable even if licking (or more generally motion) fluctuates on a trial-by-trial basis. Hence, we could use the variability in trial-by-trial motion for stratification of trials to assess whether the BOLD response patterns at CS are consistent during low and high motion. Contrary to CS events, such a stratification approach is not suitable for validating the BOLD correlates of RPE components (r and $-V$), because reward consumption is inherently connected to head motion. A stratification by frame-wise displacement would at the same time be a stratification of trials by vigor of reward consumption, which is related to value components. In contrast, at CS, the neuronal representation of value is reinforced and not expected to co-fluctuate with motion. (In support of our results regarding the RPE, we provide, in addition to Point 1.4, new evidence from recent tetrode recordings (see Point 1.5D) confirming the differential expression of the RPE at US in the anterior and posterior tubercle in Fig. 2h of the present manuscript.)

We tested the prediction that the BOLD deactivation at CS (in olfactory tubercle) persists in trials with low frame-wise displacement (FD). Towards this aim, we provide an analysis showing that stratifying trials (into two strata containing frames with low and high FD, respectively) does not affect the BOLD response patterns (new Suppl. Fig. 4a-d). Granjean et al. (2020) defined an FD threshold of 0.1 mm, and Gutierrez-Barragan et al. (2022) used 0.075 mm. We chose here a more restrictive threshold, such that low-motion CS trials were defined by a maximum FD of 0.05 mm in the first and second TR of the trial (corresponding to CS and delay). These image frames include the peak of the CS response as modeled by the fast-peaking HRF used in our analysis. We identified 16 out of 51 sessions with at least 10 trials for each CS type with FD smaller than 0.05 mm. Importantly, for the two FD strata a balanced number of trials for CS50 and CS100 averaged across sessions was obtained (with an average of $FD < 0.05$ and $FD \geq 0.05$ trials of 21.5 and 18.5, respectively, for CS50, and of 18.4 and 21.6 for CS100). We created separate regressors for low-motion and high-motion CS100 events, and for low-motion and high-motion CS50 events. The analysis pipeline is the same as in Supplementary Fig. 3b and 3e, where all 51 sessions were analyzed without any cut-off for frame-wise displacement. To illustrate the anatomical pattern of BOLD responses in this smaller sample, statistical threshold was set to $p < 0.05$, with minimum cluster size $k = 10$. Note that this lenient statistical threshold does not allow for inferences per se, but for comparing anatomical patterns of activation/deactivation between strata. We show here the response to CS100 for the low-motion trials (new Suppl. Fig. 4b) and the high-motion trials (new Suppl. Fig. 4c). In this trial stratification, the pattern of BOLD responses at CS was comparable for the two levels of motion and recapitulates the original result shown in Suppl. Fig. 3b of the manuscript. Finally, to illustrate that the anatomical pattern of value differentiation was preserved also in low-motion trials, we show the contrast between CS100 and CS50 where only low-motion trials were considered (new Suppl. Fig. 4d). The result validates the original finding in Suppl. Fig. 3e.

In summary, the results support that the direction of changes in BOLD signals (deactivation/activation), their pattern, and the order of CS50 vs. CS100 responses in relation to their value persist in conditions of low motion.

New figure legend to Supplementary Fig. 4a-d: “**Supplementary Figure 4. BOLD response patterns at CS are robust to motion.** Related to Figure 2.

a, Anatomical illustration of olfactory, striatal, and higher-order regions (location from Bregma indicated in mm).

b-c, The pattern of BOLD responses was similar when considering only trials with a maximum frame-wise displacement (FD) of 0.05 mm in the two image frames at and following CS onset (low-motion trials), and trials with an FD above this cut-off (high-motion trials). For this analysis, only sessions were included that had, per CS type, at least 10 low-motion trials, yielding a balanced average number of low- and high-motion trials for CS50 and CS100 in 16 sessions. The CS100 events in high- and low-motion trials were modeled by separate regressors, replacing the CS100 odor regressor in GLM 2; the same was applied to CS50. To illustrate the anatomical pattern of BOLD responses in this subsample, statistical threshold was set to $p < 0.05$ uncorrected, with minimum cluster size $k = 10$. Note that this lenient statistical threshold does not allow for inferences per se, but for comparing anatomical patterns of activation/deactivation between strata. This is shown for the response to CS100 in (b) low-motion and (c) high-motion trials.

d, Value differentiation as reflected in the contrast between CS100 and CS50 including only low-motion events ($p < 0.05$ uncorrected, cluster size $k = 10$) shows a similar pattern as in the original analysis (see Suppl. Fig. 3e). Together these results support that the BOLD response patterns associated with value coding are not an artifact due to motion.”

We added to the Methods section (l. 772): “We therefore tested whether the BOLD response patterns were affected by stratification into ‘low-motion’ and ‘high-motion’ events. We also assessed whether for the critical feature of value coding, namely the contrast of CS50 and CS100 at CS, the anatomical pattern was preserved in trials with low frame-wise displacement (FD). FD was computed for each image frame as the sum of the derivatives of the six realignment parameters (Power et al., 2012). Granjean et al. (2020) defined an FD threshold of 0.1 mm, and Gutierrez-Barragan et al. (2022) used 0.075 mm. We chose a more restrictive threshold, such that low-motion CS trials were defined by a maximum FD of 0.05 mm in the two image frames at and following the CS time-point. These image frames include the peak of the CS response as modeled by the fast-peaking mouse HRF. We selected all sessions ($n=16$) with at least 10 trials (per trial type) with low motion during CS and delay. This threshold was chosen to yield a balanced number of low- vs. high-motion CS100 events, low- vs. high-motion CS50 events, and low-motion CS50 vs. low-motion CS100 events (with an average of $FD < 0.05$ and $FD \geq 0.05$ trials of 21.5 and 18.5, respectively, for CS50, and of 18.4 and 21.6 for CS100). The low- and high-motion CS100 events were modeled by separate regressors, substituting the CS100 event regressor in GLM 2. Low- and high-motion CS50 events were modeled analogously. The BOLD response patterns to CS100 were similar between strata (Supplementary Fig 4a-c), and reflected the patterns found in the main analysis (cf. Supplementary Fig. 3b). Further, in the low-motion trials, the pattern of the contrast between CS100 and CS50 at CS was also preserved (Supplementary Fig. 4d compared to Supplementary

Fig 3e). Together, the data support that these main effects are robust against the effects of motion.”.

We also added to the Discussion (l. 415): “*we cannot assume a perfect separation between these two processes, due to the correlation between reward expectation/consumption and licking activity. However, in support of that motion does not explain the CS responses, the BOLD response patterns were preserved when only trials with low motion were included (see Methods section ‘Functional MRI denoising’ and Supplementary Fig. 4). ...*”

- B. As a complementary exploratory analysis that we would like to show the Reviewer, we tested additionally whether the negative BOLD responses to CS in the olfactory tubercle persist when the animals do not lick during CS or delay. When animals do not lick during CS and delay, the value representation remains stable, as the CS-elicited value response is a reinforced signal. We therefore tested the prediction that the BOLD deactivation at CS (in olfactory tubercle) persists even in trials where the animal does not lick.

Generally, there were relatively few CS50 and CS100 trials without licks during CS and delay. We could identify 23 out of the 51 sessions with at least 6 trials for each CS type without licking during CS and delay. This resulted in an average number of no-lick and lick trials of 19.7 and 20.3, respectively for CS50, and of 12.7 and 27.3 for CS100, per session. The analysis pipeline is the same as in Supplementary Fig. 3b, where all 51 sessions were analyzed with all trials included. In the new analysis, the regressor modeling CS100 was replaced by two regressors modeling no-lick- and lick-CS100 events. Analogously, CS50 was replaced by a no-lick- and a lick-CS50 regressor. To illustrate the anatomical pattern of BOLD responses in this smaller sample statistical threshold was set to $p < 0.05$, minimum cluster size $k = 10$. The Figure below shows the result of this analysis. The pattern of BOLD responses to CS100 at CS was similar between trials where animals did not (Panel b of the figure below) or did lick (Panel c) in non-motor regions (deactivation in the olfactory tubercle, and activation in the olfactory bulb and higher order cortices). For these non-motor regions, this analysis confirms the result reported originally in Suppl. Fig. 3b of the manuscript. One region, however, that differed between lick and no-lick trials was the dorsal striatum (Panel b vs. c). This is expected as the ventral striatum has been suggested to encode the value of CS, and the dorsal striatum to encode the value of an action given a particular CS (eg. O’Doherty et al., 2004 Science; Tang et al., 2022, Cell Rep). This finding is interesting and matches the previous literature. Considering however the relative scarcity of the no-lick trials, the power of the analyses is suited to probe whether BOLD response patterns are similar in a no-lick subset of the data, but not to introduce a self-standing biological new conclusion on the encoding of action values. For the latter goal, a different task would have been required. Also, this phenomenon has been shown previously in many papers. Therefore, we did not include this analysis in the revised manuscript, but acknowledged in the discussion (l. 450): „Finally, the dorsal striatum similarly reflected ~~all~~ aspects of RP-CS-bound value signals, but was dominated by the reward component of the prediction error at US. There is however an important difference between dorsal and ventral striatum. While the dorsal striatum is known to encode the value of particular actions given a particular CS, the ventral striatum has been suggested to encode the expected reward value of CS (O’Doherty et al., 2004 Science).“

Further, we explored whether in no-lick trials, the anatomical pattern of value differentiation was preserved. Specifically, we found that the pattern of the BOLD contrast in the ventral striatum (and other non-motor regions like olfactory bulb, prefrontal cortex, anterior insula)

between CS100 and CS50 (Panel d) reflects the results of the full sample analysis of Suppl. Fig. 3e.

Figure showing BOLD response patterns at CS with and without licking.

a, Anatomical illustration of olfactory, striatal, and higher-order regions (location from Bregma indicated in mm).

b-d, The patterns of BOLD responses were compared between trials where animals did or did not lick during CS presentation and delay period. For this analysis, only sessions were included that had, per trial type, at least 6 trials without licks during CS and delay, yielding 23 sessions. Both for CS100 and CS50 the 'no lick' and 'lick' trials were modeled as events at CS in separate regressors, replacing the CS100 and CS50 odor regressors in GLM 2. To illustrate the anatomical pattern of BOLD responses in this comparably small sample of 23 sessions, statistical threshold was set to $p < 0.05$ uncorrected, with minimum cluster size $k = 10$. Here shown the response to CS100 for (b) the no-lick trials and (c) the lick trials. (d) Contrast between CS100 and CS50 when including only no-lick trials."

[REDACTED]

[REDACTED]